# Personalized bacteriophage therapy outcomes for 100 consecutive cases: a multicentre, multinational, retrospective observational study

In contrast to the many reports of successful real-world cases of personalized bacteriophage therapy (BT), randomized controlled trials of non-personalized bacteriophage products have not produced the expected results. Here we present the outcomes of a retrospective observational analysis of the first 100 consecutive cases of personalized BT of difficult-to-treat infections facilitated by a Belgian consortium in 35 hospitals, 29 cities and 12 countries during the period from 1 January 2008 to 30 April 2022. We assessed how often personalized BT produced a positive clinical outcome (general efficacy) and performed a regression analysis to identify functional relationships. The most common indications were lower respiratory tract, skin and soft tissue, and bone infections, and involved combinations of 26 bacteriophages and 6 defined bacteriophage cocktails, individually selected and sometimes pre-adapted to target the causative bacterial pathogens. Clinical improvement and eradication of the targeted bacteria were reported for 77.2% and 61.3% of infections, respectively. In our dataset of 100 cases, eradication was 70% less probable when no concomitant antibiotics were used (odds ratio = 0.3; 95% confidence interval = 0.127–0.749). In vivo selection of bacteriophage resistance and in vitro bacteriophage–antibiotic synergy were documented in 43.8% (7/16 patients) and 90% (9/10) of evaluated patients, respectively. We observed a combination of antibiotic re-sensitization and reduced virulence in bacteriophage-resistant bacterial isolates that emerged during BT. Bacteriophage immune neutralization was observed in 38.5% (5/13) of screened patients. Fifteen adverse events were reported, including seven non-serious adverse drug reactions suspected to be linked to BT. While our analysis is limited by the uncontrolled nature of these data, it indicates that BT can be effective in combination with antibiotics and can inform the design of future controlled clinical trials. BT100 study, ClinicalTrials.gov registration: NCT05498363.

e-mail: jean-paul.pirnay@mil.be

Antimicrobial resistance (AMR) is a prominent global health threat with an estimated 1.27 million attributable deaths in 2019[1] and there is an urgent need to seek alternative antimicrobial strategies. Bacteriophage therapy (BT), the use of bacteriophages—the viruses of bacteria—to treat bacterial infections, was first applied by Félix d'Hérelle in 1919[2], and further developed and applied in the former Soviet Union.

A recent systematic review confirmed that BT can generally be considered as safe, with a low incidence of adverse events, and could be a promising strategy against AMR[3]. However, high-quality trials are required to make useful predictions on the outcome of bacteriophage treatments. A number of companies are currently attempting to develop and market defined broad-spectrum BT products in compliance with contemporary requirements, which involves good manufacturing practices (GMP) certification, preclinical research (toxicity and pharmacology) and conducting randomized controlled trials (RCTs). However, the handful of bacteriophage RCTs that have been performed so far have not brought the expected results in terms of effectiveness[4]. A commonly reported reason for these disappointing results is the use of invariable one-size-fits-all bacteriophage products[4].

In contrast, an increasing number of successful BT cases are reported in the scientific literature[3]. Irrespective of an obvious positive-result publication bias, most of these successful cases used tailored bacteriophage products. In addition, these personalized bacteriophage preparations, which were shown to target the infecting bacteria in vitro before their clinical application, were often used in combination with antibiotics. When appropriate, bacteriophage preparations were adapted to counter bacterial resistance that had emerged against the applied bacteriophages during BT[5], or bacteriophages were pre-adapted ('trained')[6] or engineered[7] to be more effective.

Here we report the retrospective, observational analysis of the first 100 consecutive BT cases of difficult-to-treat infections, enabled by a Belgian consortium. Because all BT cases were included in this study, not only successful, interesting, or challenging cases, we were able to (1) evaluate how often personalized BT produced a positive clinical outcome (general efficacy) and (2) identify functional relationships that are general in all cases.

Considering the relatively high number of combined categorical and numerical variables in the analysed data, the majority of patients were unique cases in most of the variables. As a result, on this dataset, no inferential statistics could be applied because these data were neither a random nor a representative sample of a population of BT-treated patients. As such, any data analysis can only be interpreted as information pertaining to the analysed patient population.

Nevertheless, the knowledge gained from these cases is likely to help physicians to select effective treatment protocols and design future clinical trials.

## Results

### Patients, bacterial infections and bacteriophage therapy

A Belgian BT consortium, consisting of the Queen Astrid Military Hospital (QAMH), KU Leuven and Sciensano (formerly known as the Scientific Institute of Public Health), facilitated BT in about 140 difficult-to-treat infections in patients in Belgium and abroad (as of July 2023), not taking into account the patients treated in the context of prospective clinical trials. The selection of patients was largely based on clinical need, regulatory approval and the availability of well-characterized bacteriophages targeting the infecting bacteria (Extended Data Fig. 1). Personalized bacteriophage preparations were produced at the QAMH in accordance with the rules in force in the territory at the time of their use in clinical practice. Of note, most selected cases concerned personalized BT as salvage therapy after standard antibiotic treatments had failed. Quality and safety of the bacteriophage preparations were verified by Sciensano according to the specifications of the Belgian bacteriophage active pharmaceutical ingredient (API) monograph[8], that is, the genomic analysis of the bacteriophage and its bacterial production host (with an emphasis on safety), and the determination of lytic activity (titre), pH, bioburden (total viable aerobic count), bacterial endotoxin level and genome sequence (identity and purity) of each bacteriophage API batch. The BT protocols that were suggested to the treating physicians were based on the experiences of the George Eliava Institute of Bacteriophages, Microbiology and Virology (Eliava Institute) in Tbilisi, Georgia (personal communications), and on the application instructions of the Ministries of Health and of Medical and Microbiology Industry of the former Union of Soviet Socialist Republics (USSR)[9–11].

During the study period (1 January 2008 to 30 April 2022), 1,066 BT requests were submitted to the QAMH. These requests resulted in 100 BT cases (9.4%). Two hundred and sixty BT requests addressed to the QAMH between April 2013 and April 2018 were analysed in detail[12]. Only 15 (5.8%) of these 260 requests resulted in actual BT. Two hundred and forty-five requests were rejected for diverse reasons: 70 applicants (26.9%) did not respond to requests for additional information; 124 requests (47.7%) concerned bacterial species against which no bacteriophages were available at the QAMH; for 46 requests (17.7%), other therapeutic options were considered more opportune; and in 5 cases (1.9%) the available bacteriophages did not target the patients' infecting bacterial strains. Rejected applications were usually referred to BT centres abroad. We consider these percentages as representative of the present patient cohort, minding an increase in the percentage of requests that resulted in BT (9.4% versus 5.8%), which is due to the increasing number of therapeutic bacteriophages in the QAMH collection. Time to treatment was dependent on whether suitable quality-controlled bacteriophages were available on hand (these could be provided immediately), or whether bacteriophages needed to be produced at the QAMH and quality and safety tests performed by Sciensano (this would take on average of 3 weeks in non-emergency cases).

A retrospective analysis of a de-identified BT database containing demographic, bacteriophage product and clinical data showed that personalized BT of 100 consecutive patients targeted 114 difficult-to-treat infections (as diagnosed by the treating physicians), including 14 second-site infections. Baseline characteristics of the patients are presented in Supplementary Table 1 and Extended Data Table 1, and provide an overview of these BT cases, which were performed by a total of 63 Bacteriophage Therapy Providers in 35 hospitals, 29 cities and 12 countries (Fig. 1a). Twenty-seven of the 100 BT cases/patients were previously reported[6,13–26]. Since 2008, the number of BT cases performed under the umbrella of different regulatory frameworks and facilitated by the Belgian consortium has increased steadily (Fig. 1b). The prevalence of the main infection types is shown in Fig. 1c. The most common indications for BT include lower respiratory tract infections (LRTI; 25.4% (29/114 infections)), skin and soft tissue

**Fig. 1 | Characteristics of the patient population involved in the 100 consecutive BT cases facilitated by the Belgian consortium. a**, Geographic location of the BT cases. **b**, Number of BT cases and their regulatory context, per year. SOC MP, standard-of-care with magistral bacteriophage preparations; DH, article 37 (unproven interventions in clinical practice) of the Declaration of Helsinki; SOC UM, standard-of-care with unlicensed medicines; ATU MP, 'Autorisation Temporaire d'Utilisation' of magistral preparations. **c**, Primary and secondary (concomitant) infection types. AbdI, abdominal infection;

OPI, orthopaedic prostheses infection. **d**, Patient age and gender distribution. Boxplot shows the interquartile range of the age (years) of the patients ($n = 90$): first quartile (29.5), median (53) and third quartile (62). The whiskers extend from the quartiles to the last data point within 1.5 × the interquartile range. Data points plotted outside the boundary of the whiskers are outliers. Female patients are represented by purple filled circles and male patients by blue filled circles. **e**, Targeted bacterial species. In some cases, bacteriophages targeted two or three bacterial species (connected by lines) in one patient.

infections (SSTI; 22.8% (26/114)), bone infections (BoneI; 14.0% (16/114)) and upper respiratory tract infections (URTI; 11.4% (13/114)). Fourteen patients presented with a second-site infection, more specifically a bloodstream infection (BSI; $n = 10$), a urinary tract infection (UTI; $n = 2$),

an SSTI ($n = 1$) or a URTI ($n = 1$). Age and gender distribution are shown in Fig. 1d. The median age of the patients was 53 years (1–91 years), and 56.7% of the patients were male. Of note, 5 patients were 1 year or younger. Fourteen bacterial species were targeted (Fig. 1e), with the

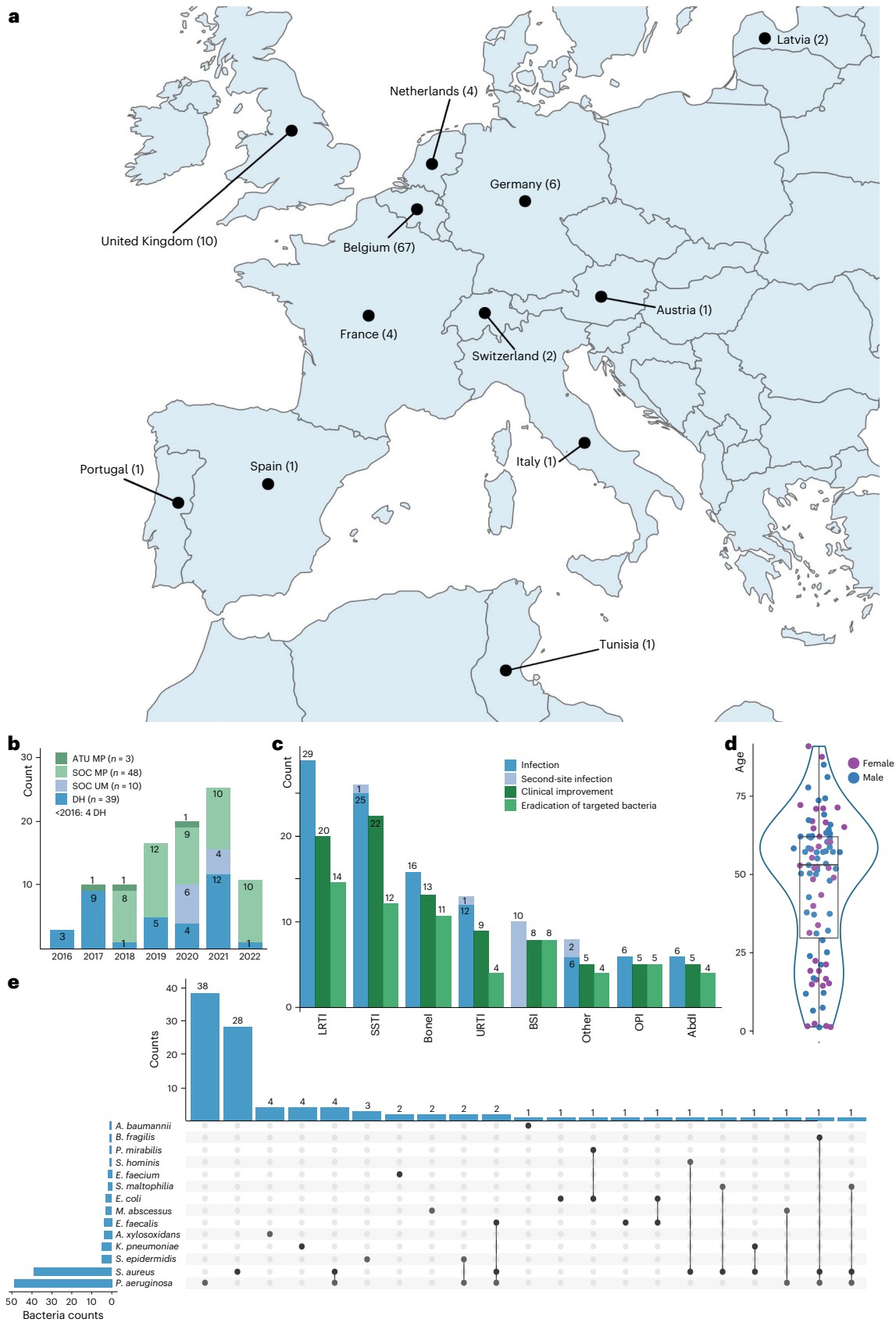

**Table 1 | General overview of bacteriophage therapy protocols according to the main infection types**

| Infection type | Application route | Bacteriophage carrier | Volume (ml) | Concentration (p.f.u.s ml⁻¹) | Dose | Duration |
|---|---|---|---|---|---|---|
| Lower respiratory tract infections | Nebulization | NaCl 0.9% | 2–4 | $10^7$–$10^8$ | q6h | 5 days–6 weeks |
| Bone and orthopaedic prostheses infections | Intralesional | NaCl 0.9% | 2–70 | $10^7$–$10^8$ | q24h | 5 days–3 weeks |
| Skin and soft tissue infections | Topical | NaCl 0.9% or Flaminal Hydro | In excess | $10^7$–$10^9$ | q24h | 5 days–3 weeks |
| Upper respiratory tract infections | Nasal spray | NaCl 0.9% | 1–15 | $10^7$ | q8h | 1–3 weeks |
| Bloodstream infections or other infection types[a] | Intravenous | NaCl 0.9% | 50–100 | $10^6$–$10^7$ | q24h | 5–10 days |

[a]When the treating physician considered it was necessary to apply bacteriophages systemically. p.f.u.s, plaque forming units; q, every.

highest prevalence for *Pseudomonas aeruginosa* (49/100 patients) and *Staphylococcus aureus* (39/100 patients).

Twenty-six individual bacteriophages (Supplementary Table 3) and six defined bacteriophage cocktails (Supplementary Table 4), including two commercially available cocktails (PyoPhage and IntestiPhage) produced by the George Eliava Institute of Bacteriophages, Microbiology and Virology (Eliava Institute) in Tbilisi (Georgia), were used. Bacteriophages were provided by the QAMH and 16 Bacteriophage Donors affiliated to 10 institutes in 7 countries.

Most BT providers adhered to BT protocols proposed by QAMH physicians, which resulted in a surprisingly small variation in BT protocols within a given indication. Table 1 provides a general overview of these protocols, while the individual protocols of the 100 cases are listed in Supplementary Table 1. Bacteriophages were administered intravenously to 20 patients (Supplementary Table 1); in 10 of them as stand-alone BT, in 10 concomitantly with intralesional ($n = 4$), nebulized ($n = 3$), topical ($n = 2$) or generalized (multiple application routes; $n = 1$) bacteriophage application. In 10 patients, intravenous bacteriophages were used to treat or prevent bloodstream infections. In 69.3% (79/114) of targeted infections, bacteriophages were administered in combination with standard-of-care antibiotics.

**Pre-adaptation of bacteriophages**

The most frequently used bacteriophages, that is, *Staphylococcus* bacteriophage ISP (33 patients) and *P. aeruginosa* bacteriophages 14-1 (22 patients), PNM (21 patients) and PT07 (18 patients) (Supplementary Table 3), were regularly (one to two times per year) adapted using a selection of three to five recent bacterial strains of concern. In addition, 13 bacteriophages were specifically pre-adapted to lyse the patient's bacteria in a therapeutically relevant manner (Methods), that is, to produce stable lysis (without emergence of bacteriophage-insensitive mutants) in liquid culture for typically 24–48 h at a multiplicity of infection (MOI) ≤ 1 (Extended Data Table 2). The genomes of the pre-adapted bacteriophages were sequenced, analysed and compared to those of their original precursors as part of the Sciensano coordinated SAPHETY project (https://www.sciensano.be/en/control-and-safety-assessment/safety-therapeutic-bacteriophage-preparations), which focuses on setting new standards for the quality and safety of therapeutic bacteriophage products. One pre-adaptation effort increased the activity of *S. aureus* bacteriophage ISP against an *S. epidermidis* clinical isolate in view of personalized BT. The pre-adaptation process (four serial passages) resulted in missense mutations in three genes, including a carbohydrate-binding domain protein and a uracil-DNA glycosylase (Extended Data Fig. 2), which are closely related (closest BLAST hits) to two previously identified receptor binding proteins[27]. However, the increased virulence and resistance suppression of the pre-adapted ISP variant was accompanied by a decreased host range. Where the original ISP clone showed a moderate activity (efficiency of plating (EOP) ≤ 0.01) against 3/16 *S. epidermidis* strains, the adapted variant

showed a therapeutically acceptable activity against the patient's strain only.

**Diagnostic tests to support bacteriophage therapy**

For 21 patients, sufficient and adequate consecutive bacterial samples and/or serum samples were provided, allowing assessment of (1) the potential in vivo emergence of resistance against the applied bacteriophages, (2) in vitro bacteriophage–antibiotic interactions and/or (3) the emergence of bacteriophage immune neutralization (Table 2).

**Selection of bacteriophage resistance**

For 16 patients, sufficient bacterial samples (isolated before, during and after BT) were available to evaluate the possible emergence of bacterial bacteriophage resistance. Whether adequate samples were available was not directly linked to the clinical indications for BT but depended mainly on the treatment centres and their bacteriological monitoring routines. For 5 patients in Table 2, no adequate sample sets were available (indicated with 'NSA, no samples available' in Table 2). We observed the in vivo selection of bacterial strains exhibiting a bacteriophage-insensitive phenotype, and the possible underlying phenotype–genotype associations in 7 of these 16 (43.8%) patients (patients 16, 20, 30, 54, 64, 82 and 91 in Table 2). Whole-genome single nucleotide polymorphism (SNP) analysis was performed for bacterial isolates from the patients where bacteriophage insensitivity emerged. In two patients (64 and 82 in Table 2), sequential bacteriophage-susceptible and bacteriophage-insensitive *P. aeruginosa* isolates were determined not to be clonal. Phylogenetic comparison showed that for patient 82, bacteriophage-susceptible strains belonged to an emerging rare sequence type (ST)235, whereas bacteriophage-resistant strains belonged to the more prevalent multidrug-resistant ST357 (Table 2 and Fig. 2a)[28,29]. For patient 64, the susceptible strain was ST1233 (same ST as the strains from patient 91), while the resistant strain was determined to be ST549 (Table 2 and Fig. 2a). In these two patients, BT probably selected for *P. aeruginosa* strains that were not a suitable host for the applied bacteriophages. Clinical improvement was reported in both patients.

SNPs or deletions in genes related to the bacteriophage receptor were assumed to be the basis of the resistance phenotype in five patients (16, 20, 30, 54 and 91 in Table 2). In three of them (patients 30, 54 and 91), the targeted *P. aeruginosa* strains were not eradicated. The selection of bacteriophage-resistant mutants in two patients (16 and 20) was previously described[16,17]. In patient 16, an isolate of the targeted *Achromobacter xylosoxidans* strain emerged to harbour a missense mutation in the gene coding for the putative colicin I receptor Cir, which was identified as a bacteriophage receptor. In patient 20, a missense mutation occurred in the *pilB* gene of the targeted *P. aeruginosa* strain, while the *pilM* gene was inactivated by the insertion of IS5 transposase. Both *pilB* and *pilM* are involved in the biosynthesis of Type IV pilus (T4P), the receptor for the applied *P. aeruginosa* bacteriophage PNM[30].

**Table 2 | Results of the supportive tests performed for 21 of the present 100 consecutive bacteriophage therapy cases**

| Patient number | Infection type | Targeted bacterial species | Applied bacterio-phage(s) | Bacteriophage administration route(s) | In vivo selection of bacteriophage resistance—possible underlying mechanism(s) | In vitro bacteriophage–antibiotic interactions | Bacteriophage immune neutralization | Clinical improve-ment | Eradication of targeted bacteria | Reference |
|---|---|---|---|---|---|---|---|---|---|---|
| 9 | Fracture-related infection | Klebsiella pneumoniae | M1 | Intralesional (catheter) | Not observed | M1 synergy with ceftazidime/avibactam and meropenem | M1 neutralization emerged between days 8 and 18 after BT initiation | Yes | Yes | Ref. [6] |
| 13 | Wound and bloodstream infection | Pseudomonas aeruginosa | 14-1, PNM and ISP (BFC 1) | Topical and intravenous | Not observed | No concomitant antibiotics | 14-1 neutralization emerged 10 days after BT initiation | Yes | Yes | Ref. [15] |
| 16 | Cystic fibrosis lung transplant infection | Achromobacter xylosoxidans | JWAlpha, JWDelta, JWT and 2:1 (APC 1.1 and APC 2.1) | Nebulization | Yes—p.Tyr601X MS Mut in colicin I receptor Cir | No concomitant antibiotics | NSA | Yes | Yes | Ref. [16] |
| 20 | Liver transplant and bloodstream infection | P. aeruginosa | 14-1, PNM and ISP (BFC 1) | Intralesional (infusions) and intravenous | Yes—p.Asp388Ala MS Mut in PilB and deactivation of PilM by insertion of IS5 transposase, both involved in Type IV pili biosynthesis, without impact on virulence | PNM synergy with colistin, aztreonam and gentamycin | ISP neutralization emerged 5 weeks after BT initiation. No neutralization of 14-1 or PNM | Yes | Yes | Ref. [17] |
| 21 | Bone allograft infection | Staphylococcus aureus | 14-1, PNM and ISP (BFC 1) | Intralesional (catheter) and intravenous | Not observed | ISP synergy with clindamycin, additive effect of ISP and ciprofloxacin, moderate ISP antagonism with rifampicin | NSA | Yes | Yes | Ref. [18] |
| 22 | Chronic osteomyelitis of the pelvis | P. aeruginosa and S. epidermidis | 14-1, PNM, ISP (BFC 1) | Intralesional (catheter) | NSA | NA | 1month after BT initiation, no bacteriophage neutralization could be detected | Yes | Yes | Ref. [19] |
| 23 | Chronic osteomyelitis of the femur | S. aureus | 14-1, PNM, ISP (BFC 1) | Intralesional (catheter) | NSA | NA | 1month after BT initiation, no bacteriophage neutralization could be detected | Yes | Yes | Ref. [19] |
| 24 | Chronic osteomyelitis of the femur | P. aeruginosa and S. epidermidis | 14-1, PNM, ISP (BFC 1) | Intralesional (catheter) | NSA | NA | 1month after BT initiation, no bacteriophage neutralization could be detected | Yes | Yes | Ref. [19] |
| 26 | Spinal infection | P. aeruginosa | 4O29, 4O32 and 4O34 | Local and intravenous | Not observed | NA | NSA | Yes | Yes | Ref. [20] |
| 27 | Orthopaedic infection | P. aeruginosa | 14-1, PNM and ISP (BFC 1) | Local | Not observed | Additive effect of the bacteriophage cocktail with ceftazidime/avibactam | NSA | Yes | Yes | Ref. [21] |
| 30 | Chronic sinusitis | P. aeruginosa and S. aureus | 14-1, PNM, ISP (BFC 1) | Nasal spray | Yes—p.Ala154 Pro MS Mut in PilC, involved in Type IV pili biosynthesis | No concomitant antibiotics | NSA | No | No | Unpublished |
| 42 | Chronic osteomyelitis of the femur | Enterococcus faecalis | PyoPhage | Intralesional (catheter) | NSA | NA | 1month after BT initiation, no bacteriophage neutralization could be detected | Yes | Yes | Ref. [19] |
| 43 | Liver transplant infection | Enterococcus faecium | EfgrKN and EfgrNG | Intravenous | Not observed | Synergy of EfgrKN with vancomycin, loss of vancomycin resistance | 49days after BT initiation, no bacteriophage neutralization could be detected | Yes | No | Ref. [23] |

**Table 2 (continued) | Results of the supportive tests performed for 21 of the present 100 consecutive bacteriophage therapy cases**

| Patient number | Infection type | Targeted bacterial species | Applied bacterio-phage(s) | Bacteriophage administration route(s) | In vivo selection of bacteriophage resistance—possible underlying mechanism(s) | In vitro bacteriophage–antibiotic interactions | Bacteriophage immune neutralization | Clinical improve-ment | Eradication of targeted bacteria | Reference |
|---|---|---|---|---|---|---|---|---|---|---|
| 54 | Ventilator-associated pneumonia | *P. aeruginosa* | 14-1, PNM and PTO7 | Nebulization | Yes—p.Thr230Proline MS Mut in PilR, involved in Type IV pili biosynthesis | No clear interaction between PNM or 14-1 and colistin | 2 months after BT initiation, no bacteriophage neutralization could be detected | Yes | No | Unpublished |
| 55 | Musculoskeletal infection | *S. epidermidis* | ISP and BEO6 | Intralesional and intravenous | Not observed | No concomitant antibiotics | NSA | No | No | Unpublished |
| 64 | Anal fistula | *P. aeruginosa* | 14-1, PNM and PTO7 | Intralesional | Yes—selection of another strain, which is not a host for bacteriophages 14-1, PNM or PTO7 | No concomitant antibiotics | 4 and 7 months after BT initiation, no bacteriophage neutralization could be detected | Yes | Yes | Unpublished |
| 66 | Cystic fibrosis lung infection | *M. abscessus* | 8UZL | Nebulization and intravenous | NSA | NA | 8UZL neutralization emerged 7 days after BT initiation | No | No | Unpublished |
| 71 | Lung infection | *P. aeruginosa* | PTO7 | Nebulization | Not observed | Additive effect of PTO7 and ceftazidime | NSA | Yes | Yes | Unpublished |
| 82 | Cystic fibrosis lung infection | *P. aeruginosa* | 4P and DP1 | Nebulization | Yes—selection of another strain, which is not a host for bacteriophages 4P and DP1 | Synergy of 4P and DP1 with levofloxacin, no clear interaction between 4P or DP1 and tobramycin | NSA | Yes | Yes | Unpublished |
| 91 | Lung infection | *P. aeruginosa* | 14-1, PNM and PTO7 | Nebulization and intravenous | Yes— **LPS biosynthesis:** (Is 2 and 3) p.Trp139X NS Mut in WapH, (Is 2 and 3) p.Gln239X NS Mut in GalU, (Is 4 and 5) p.Leu162Pro MS Mut in WapR, (Is 4 and 5) p.Leu60_Leu63del in WbpR **Type IV pili biosynthesis:** (Is 4 and 5) p.Arg120fsX in FimV, missing the first 165 aa **Other:** (Is 6) p.Gly406Ser MS Mut in CupE5 fimbrae assembly protein, (Is 2, 3, 4, 5 and 6) p.Arg994Gly MS Mut in MexB of MexAB-OprM, (Is 4 and 5) p.His87Asp MS Mut in GyrA | PTO7 synergy with colistin and meropenem | 2 weeks after BT initiation, no bacteriophage neutralization could be detected | Yes | No | Unpublished |
| 92 | Generalized necrotizing fasciitis, empyema, bacteremia | *S. aureus, P. aeruginosa* and *Stenotro-phomonas maltophilia* | ISP, 14-1, PNM, PTO7 and BUCT700 | ISP: intravenous, intrapleural, intraperitoneal and nebulization; All: topical | Not observed | ISP synergy with vancomycin, ceftarolin and clindamycin | ISP neutralization emerged 6 days after BT initiation | Yes | Yes | Unpublished |

aa, amino acids; del, deletion; fs, frameshift; Is, isolate; MS Mut, missense mutation; NA, not analysed; NS Mut, nonsense mutation; X, stop.

Bacteriophage-resistant *P. aeruginosa* mutants were also isolated from patients 30, 54 and 91 infected with *P. aeruginosa* (Table 2 and Fig. 2b–d). Among these mutations, SNPs were identified that corresponded to regions related to T4P in all three patients. In one patient (54), this mutation was in the *pilR* gene, coding for the transcriptional activator of a two-component system that regulates expression of the major pilin subunit PilA (Fig. 2b)[31]. In another patient (30) isolate, this mutation was in a gene coding for an inner membrane component, PilC, essential for T4P biogenesis (Fig. 2c)[32]. For patient 91, a premature stop codon was introduced producing a truncated gene variant of the gene *fimV*, which expresses a part of the inner membrane assembly of T4P in *P. aeruginosa* (Fig. 2d)[33]. In addition, this patient was shown to harbour bacteria that exhibited simultaneous resistance to all three unique *P. aeruginosa* bacteriophages from treatment: PNM, PT07 and 14-1 (Table 2 and Fig. 2d). Interestingly, we observed two distinct bacteriophage-resistant variants of the initially targeted *P. aeruginosa* strain, each showing resistance to the three applied bacteriophages, which all had different bacterial receptors. *P. aeruginosa* bacteriophage 14-1 infects via a lipopolysaccharide (LPS) receptor[34]. Unsurprisingly, SNPs were identified in genes in the outer core of the *P. aeruginosa* LPS membrane, that is, *wapH*, *galU*, *wbpR* (gene products truncated in these three mutant variants) and *wapR*. Although the receptor for *P. aeruginosa* bacteriophage PT07 is not known, sequence similarity to PAK-P1-like bacteriophages (98.26% identity to bacteriophage PaP1) suggests that this bacteriophage is dependent on the *P. aeruginosa* MexAB-OprM multidrug efflux pump[35]. A resistance mutant of PT07 was identified with an SNP in the gene *mexB*. Two *P. aeruginosa* isolates (Is 4 and 5 in Table 2) from patient 91 had both the *mexB* mutation and another mutation in DNA gyrase subunit A (*gyrA*), part of the bacterial DNA topoisomerase. This mutation (H87A) is within the GyrA quinolone-resistance determining region (QRDR)[36–38]. The interplay of the MexAB-OprM efflux pump and a DNA gyrase mutation has been associated with high-level fluoroquinolone resistance in *P. aeruginosa*[39]. Interestingly, the bacteriophage-insensitive *P. aeruginosa* isolates retrieved from patient 91 carrying the double mutation in *mexB* and *gyrA* showed a re-sensitization to fluoroquinolones while displaying unaltered growth kinetics, illustrated by a decrease in the minimum inhibitory concentration (MIC) from ≥4 to 0.5 µg ml⁻¹ for ciprofloxacin and from ≥8 to 1 µg ml⁻¹ for levofloxacin. Of note, patient 91 was treated concomitantly with bacteriophages and the antibiotics meropenem, colimycin and vancomycin.

### *Galleria mellonella* virulence assays

Since these mutations in isolates from patients 30, 54 and 91 are in encoded virulence factors (Type IV pili, lipopolysaccharide), we implemented a *Galleria mellonella* infection model to readily assess the virulence of bacteriophage-susceptible versus bacteriophage-resistant variants of these *P. aeruginosa* strains. Larvae infected with original, bacteriophage-susceptible isolates showed rapid and significant mortality within 48 h (100% death) (Extended Data Fig. 3). The groups infected with bacteriophage-resistant mutants from patient 91 with multiple mutations (>2) in genes encoding for different regions (LPS, MexAB-OprM and/or T4P and/or DNA gyrase) showed significantly higher survival rates (*P* < 0.0001) compared with the larvae infected with the original isolate in this model system. Significantly higher survival rates were also observed for the larvae infected with the bacteriophage-resistant variant of the *P. aeruginosa* strain isolated from patient 54 as compared with the original bacteriophage-susceptible variant (*P* = 0.01). However,

all larvae from these two groups died in 18 h. The larvae infected with the bacteriophage-resistant isolate from patient 30 showed no difference in survival compared with those infected with the original isolate. Consequently, in patient 91 we saw a combination of antibiotic re-sensitization and reduced virulence of bacteriophage-resistant isolates, which may have contributed to an eventual favourable treatment outcome.

### In vitro bacteriophage–antibiotic interactions

Bacteriophage–antibiotic–bacteria interactions were analysed for suboptimal ratios of bacteriophages to bacteria (MOI ≤ 1) and subMIC levels (0.5 × MIC) of antibiotics. These suboptimal conditions were necessary to enable the observation of these interactions. If either bacteriophages or antibiotics were applied under optimal concentrations, this would have led to the efficient killing of the bacterial strains by either antibiotics or bacteriophages, making it impossible to demonstrate possible synergistic, additive, or antagonistic interactions. In vitro bacteriophage–antibiotic–bacteria interaction experiments revealed a synergistic or additive effect of bacteriophages and concomitantly applied antibiotics in 9 out of 10 evaluated patients (9, 20, 21, 27, 43, 71, 82, 91 and 92). An overview of the test results is presented in Table 2. The results of the experiments concerning the first 5 patients (9, 20, 21, 27 and 43) were reported previously[6,17,18,23]. The detailed results (OmniLog growth curves) for the 5 most recent patients (54, 71, 82, 91 and 92) are presented in Fig. 3. In vitro synergy with bacteriophages was observed for 9 antibiotics (aztreonam (patient 20), ceftaroline (92), ceftazidime/avibactam (9), clindamycin (21 and 92), colistin (20 and 91), gentamicin (20), levofloxacin (82), meropenem (9 and 91) and vancomycin (43 and 92)), and an additive effect for three antibiotics (ceftazidime/avibactam (27), ceftazidime (71) and ciprofloxacin (21)). For one patient (54), no significant in vitro interactions between colistin and *P. aeruginosa* bacteriophages PNM (Fig. 3a) or 14-1 (Fig. 3b) were observed. Bacteriophages 4P and DP1 acted in synergy with levofloxacin (Fig. 3d,e), but showed no clear interaction with tobramycin (Fig. 3f,g), when tested in vitro against the *P. aeruginosa* strain of patient 82. Importantly, a moderate antagonism was observed for *S. aureus* bacteriophage ISP with rifampicin (patient 21 in Table 2) in one of our previously published BT cases[18]. Of note, when most of these tests were performed, BT had already started and test results did not influence patient treatment. However, today, on the basis of these results and the overall observation that pathogen eradication is more likely when phages are applied in combination with antibiotics, we strongly advise physicians to have these tests performed before treatment, if time permits.

### Bacteriophage immune neutralization

For 13 patients, sufficient serum samples (obtained before, during and after BT) were available to allow for an adequate bacteriophage immune neutralization screening. The applied serum concentration (0.9%) and incubation time (30 min) conform to the standard technique developed by M. H. Adams in 1959 to specifically detect bacteriophage neutralization activity. Bacteriophage immune neutralization was observed between 6 and 35 days after initiation of BT in 5 of 13 (38.5%) screened patients (9, 13, 20, 66 and 92 in Table 2 and Fig. 4a–d). Bacteriophage immune neutralization always involved invasive (intravenous and/or intralesional) bacteriophage administrations. In 4 of these 5 cases (patients 9, 13, 20 and 92), clinical improvement and eradication of the targeted bacterial pathogen were nevertheless observed. In a liver transplant patient (43 in Table 2), the intravenous administration of

**Fig. 2 | The in vivo emergence of bacteriophage resistance during BT.** Monitored by whole-genome analysis of sequential bacterial isolates in patients 30, 54, 64, 82 and 91 (in vivo emergence of bacteriophage resistance in patients 16 and 20 discussed in Table 2). **a**, Maximum likelihood phylogenetic tree of the genomes of the analysed sequential bacterial isolates. **b**–**d**, Circular chromosomal view (CCV) of the bacterial genomes of sequential isolates (Is) of *Pseudomonas aeruginosa* strains retrieved just before (Is 1, inner circle) and during BT (Is 2-n) from patients 54 (**b**), 30 (**c**) and 91 (**d**). Green rings display the genomes of bacteriophage-susceptible isolates, while the red rings display the genomes and relevant (for bacterial bacteriophage resistance) mutations in bacteriophage-resistant isolates. The two multicoloured outer rings display the protein annotations (categories) as present in the Clusters of Orthologous Groups of proteins (COGs) database. bp, basepairs; CDS, coding sequence; IS, insertion sequence; Mb, megabases; nt, nucleotide; PTM, post-translational modification.

bacteriophages did not elicit any immune neutralization. In another liver transplant patient (20), bacteriophage immune neutralization emerged, but only after 5 weeks, and it concerned 1 of the 3 bacteriophages that had been applied (Table 2 and Fig. 4c).

### Clinical outcomes

Clinical improvement was reported in 77.2% (88/114) of targeted infections and eradication of the targeted bacteria was observed in 61.3% (65/106) of infections for which relevant bacteriological follow-up data

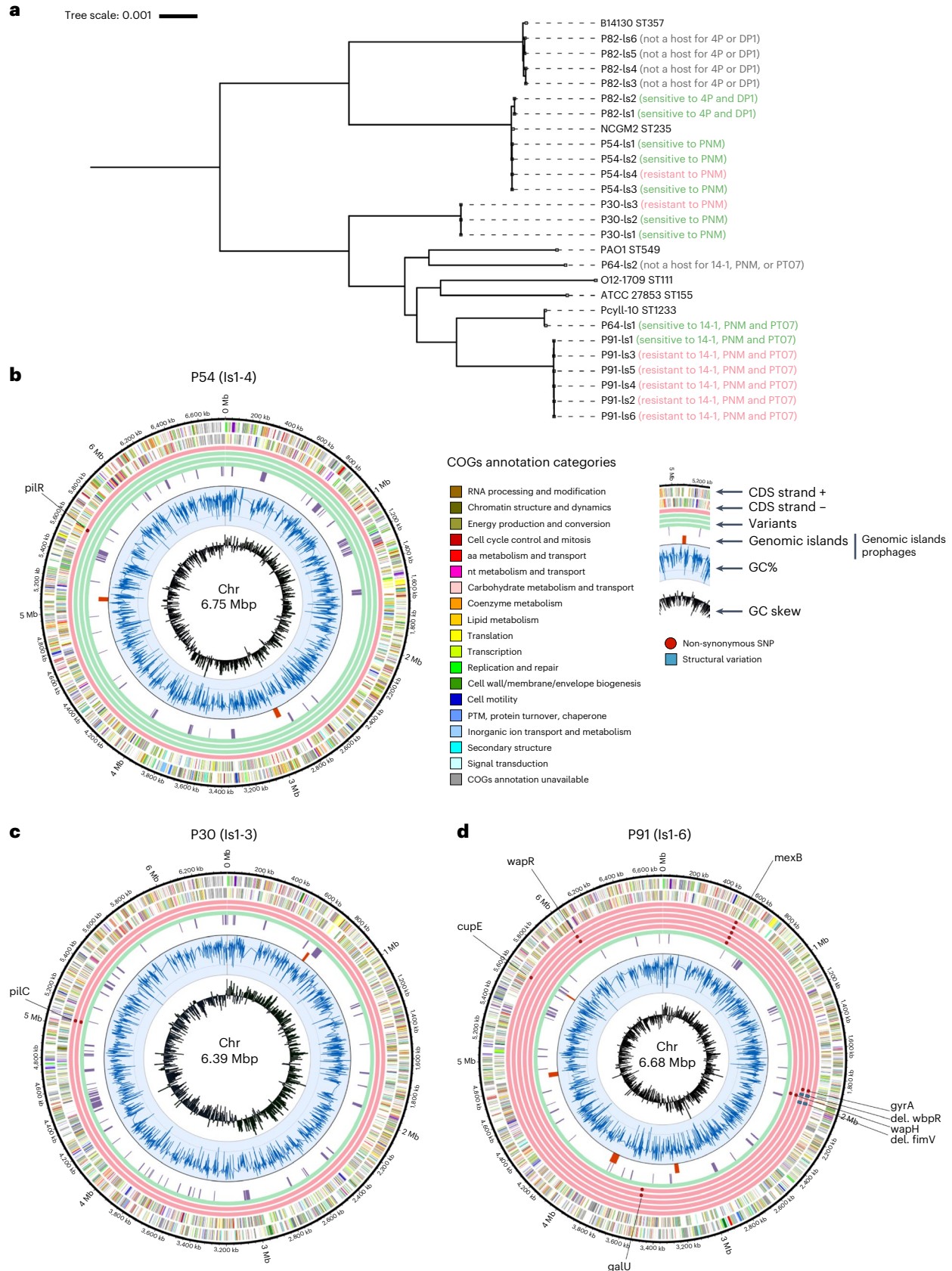

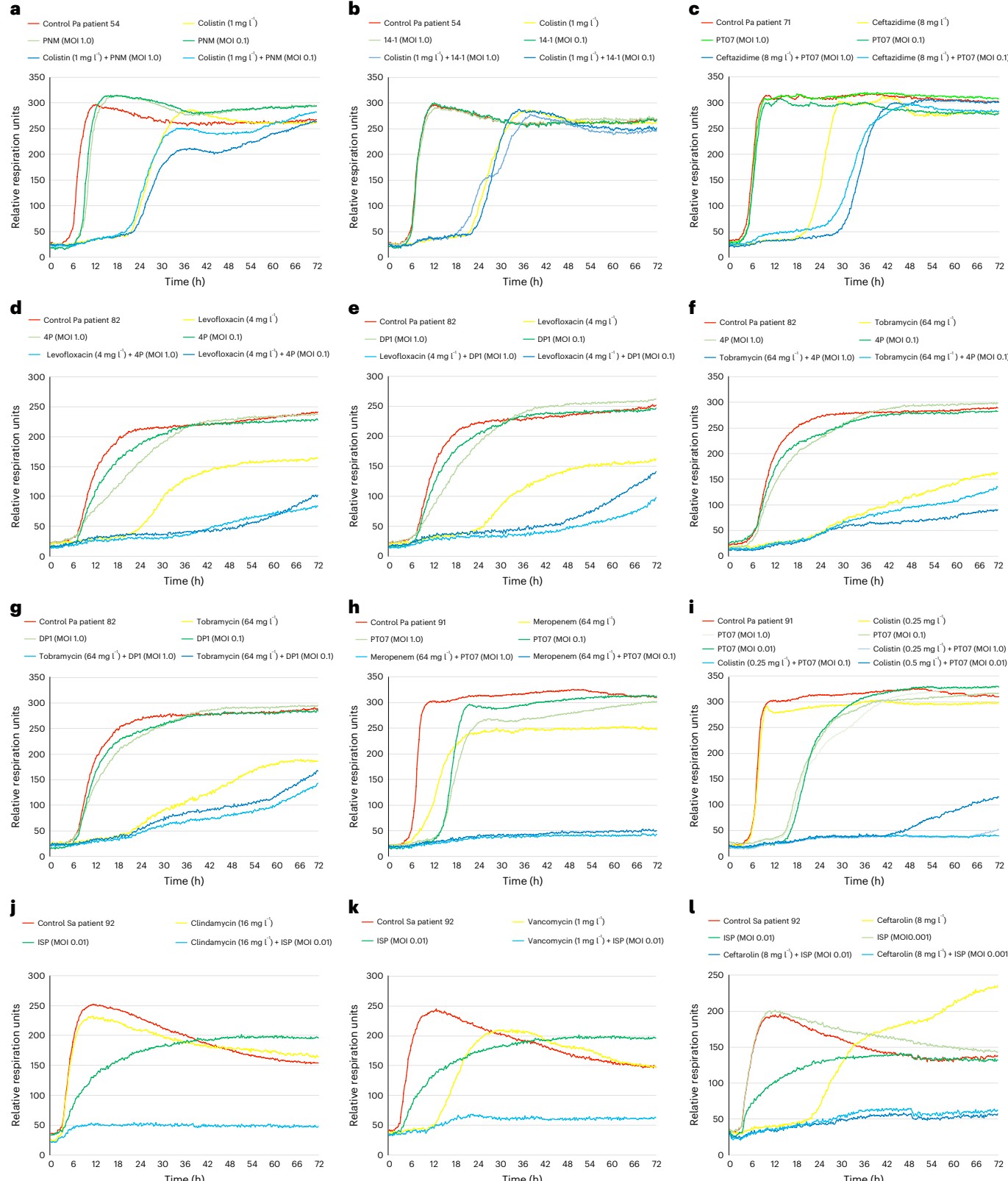

**Fig. 3 | Results of the in vitro evaluation of the combined effects of bacteriophages and concomitantly applied antibiotics on the targeted bacterial strains.** Determined by an OmniLog system for patients 54, 71, 82, 91 and 92 (those for patients 9, 20, 21, 27 and 43 are discussed in Table 2). Bacterial proliferation is presented through relative units of cellular respiration. **a,b**, No additive effect of colistin and bacteriophages PNM (**a**) and 14-1 (**b**) for patient 54. **c**, Additive effect (delayed bacterial growth) of ceftazidime and bacteriophage PT07 for patient 71. **d,e**, Synergistic effect of levofloxacin and bacteriophages 4P (**d**) and DP1 (**e**) for patient 82. **f,g**, No additive effect of tobramycin and bacteriophages 4P (**f**) and DP1 (**g**) for patient 82. **h,i**, Synergistic effect of bacteriophage PT07 and the antibiotics meropenem (**h**) and colistin (**i**) for patient 91. **j–l**, Synergistic effect of bacteriophage ISP and the antibiotics clindamycin (**j**), vancomycin (**k**) and ceftarolin (**l**). Pa, *Pseudomonas aeruginosa*; Sa, *Staphylococcus aureus*.

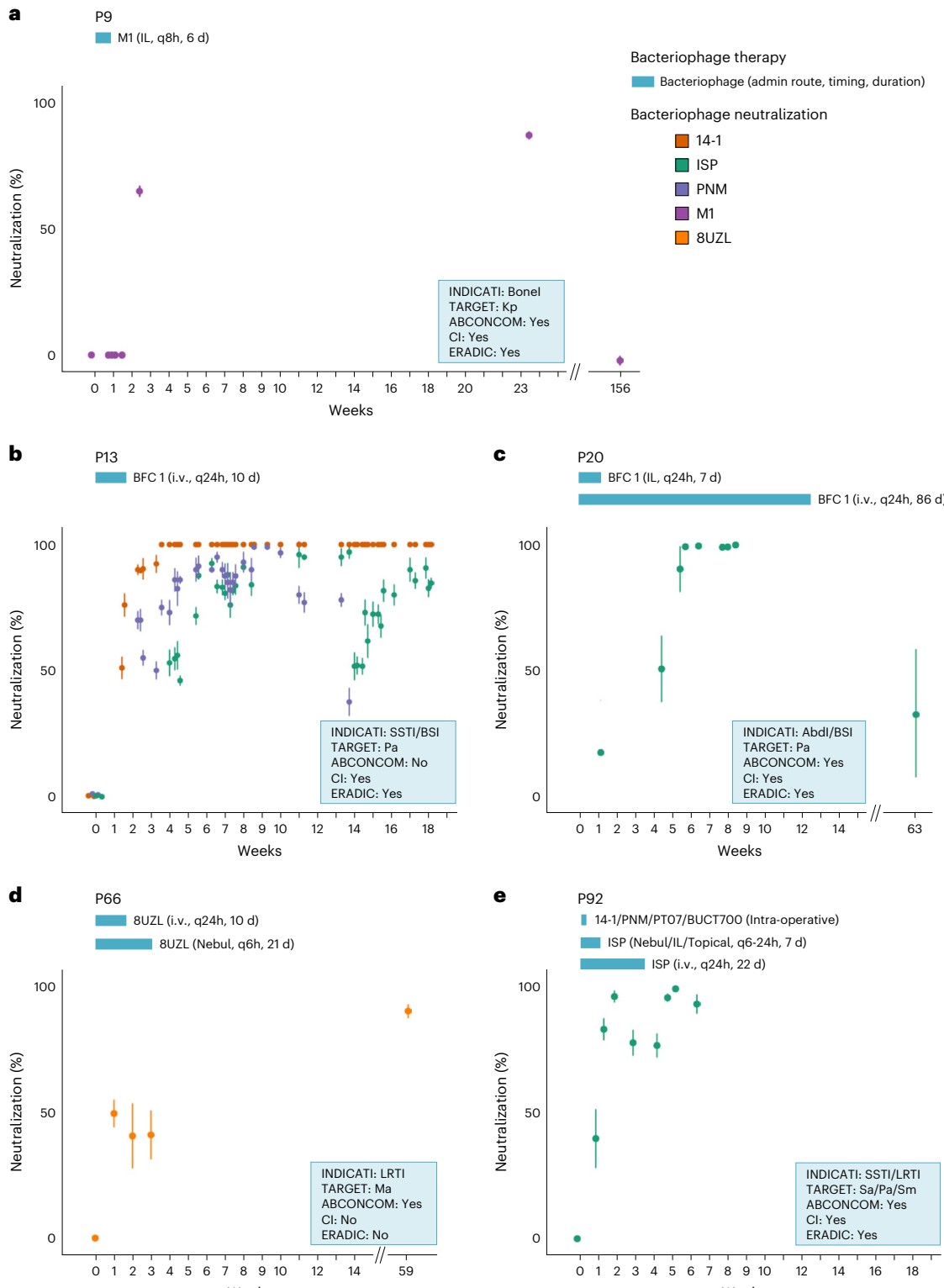

**Fig. 4 | Emergence of bacteriophage immune neutralization.**
**a**–**e**, Chronological bacteriophage immune neutralization (BIN) activity against the applied bacteriophages in sera collected before, during and after BT in patients 9 (**a**), 13 (**b**), 20 (**c**), 66 (**d**) and 92 (**e**). The evolution over time of the serum BIN activity against the applied bacteriophages is shown as % bacteriophage titre loss (compared to pre-BT control sera) after incubation of the bacteriophages with sequential serum samples for 30 min. BIN activity appeared 1–5 weeks after BT initiation. Data are presented as mean ± s.d. of three biological replicates. ABCONCOM, concomitant antibiotherapy; admin, administration; CI, clinical improvement; ERADIC, eradication; IL, intralesional; INDICATI, indication; i.v., intravenous; Kp, *Klebsiella pneumoniae*; Ma, *Mycobacterium abscessus*; Nebul, nebulization; Pa, *Pseudomonas aeruginosa*; Sa, *Staphylococcus aureus*; Sm, *Stenotrophomonas maltophilia*; TARGET, targeted bacterial species. Bacteriophage cocktail BFC 1 contains bacteriophages ISP, 14-1 and PNM.

were available (Supplementary Table 1). For 8 targeted infections, in 8 patients, no adequate post BT bacteriological data were available (Supplementary Table 1). For 7 of these cases, the treatment centre did not collect the necessary bacteriological data as part of their routine follow up of patients and was not allowed to collect the data prospectively. For the remaining case, it is not clear why the bacteriological data were not available. The treatment centre either did not collect these data, failed to extract these data from the medical files, or was not able or willing to transfer these data to the Phage Therapy Coordination Centre (PTCC).

BT resulted in clinical improvement without bacterial eradication in 18 of 106 (17.0%) targeted and bacteriologically monitored infections (Supplementary Table 1). Conversely, in 2 patients (44 and 93 in Supplementary Table 1), eradication of the targeted pathogens was observed without clinical improvement. In patient 44, an infection with an additional (non-BT-targeted) bacterial species (*Acinetobacter baumannii*) surfaced during BT, which ultimately resulted in an *A. baumannii* pulmonary septic shock and the patient's death, despite intravenous administration of tigecycline. Patient 93 succumbed to tumour progression and palliative care.

For 21 of the 92 (22.8%) patients for which bacteriological follow-up data were available, neither clinical improvement nor bacterial eradication could be observed. Five of these patients (3, 36, 40, 69 and 96 in Supplementary Table 1) died. The causes of death were septic shock ($n = 2$), cardiogenic shock ($n = 1$), multi-organ failure ($n = 1$) and COVID-19 infection ($n = 1$). In 69.3% (79/114) of targeted infections, concomitant standard-of-care antibiotics were administered (Supplementary Table 1).

Fisher's exact test for count data showed univariate significant effects on eradication for the following categorical variables: concomitant use of antibiotics (yes or no), antibiotic resistance profile of the targeted bacteria (multidrug resistant or not) and the clinical setting (ambulatory or hospitalized). No effects of patient age or gender on eradication of the targeted bacteria were observed using univariate logistic regression considering solely age or gender, respectively. A stepwise, forward selection logistic regression analysis of eradication on all independent variables determined that the concomitant use of antibiotics (variable ABCONCOM) was the most informative variable in the reduced dataset (Supplementary Table 2). In our dataset of 100 consecutive cases, eradication of the targeted bacteria (variable ERADIC) was 70% less probable when no concomitant antibiotics were used (odds ratio = 0.3; 95% confidence interval = 0.127–0.749). The *P* value for Fisher's exact test of independence between ABCONCOM and ERADIC was 0.01488. The contingency table shows that our logistic regression model is right 65% (40 + 20/92) of the time. The antibiotic resistance profile of the target bacteria (ABRPROF) and the clinical setting (CLINSETT) as well as their interactions with the concomitant use of antibiotics (ABCONCOM) were not selected in the overall logistic regression model. This could be attributed to confounding relations between these three variables within this dataset. A significant association was found between clinical improvement and bacterial eradication. Of the 23 patients with no clinical improvement, only 2 patients expressed eradication. Of the 69 patients with clinical improvement, 53 had full eradication. Intravenous BT, as stand-alone or concomitant therapy, was not shown to significantly impact clinical outcome, as also found for patient age or gender, the persistence of the bacterial infection (chronic or acute), or the use of more than one targeting bacteriophage per bacterial strain. Clinical improvement or bacterial eradication was not significantly correlated with the presence of either *P. aeruginosa* or *S. aureus*, where other species were not considered separately, as their prevalence in this study population was too low, or with any individual bacteriophage or bacteriophage cocktail.

Fifteen adverse events were reported, including seven non-serious adverse drug reactions suspected to be linked to BT (Extended Data Table 3). All suspected adverse drug reactions resolved. No correlation between adverse events and a certain bacteriophage product or administration route could be made.

## Discussion

In this overview of the first 100 consecutive real-world cases of personalized BT treatment, we show that (1) we were able to produce more than 40 batches of personalized bacteriophage APIs, some of them pre-adapted[6], which were subsequently certified for use in pharmaceutical preparations; (2) when used in the treatment of 114 difficult-to-treat infections of various types and aetiology, in combination with antibiotics in 69.3% of cases, these preparations led to clinical improvement in 77.2% and eradication of the targeted bacteria in 61.3% of cases; (3) seven non-serious suspected adverse drug reactions were reported.

The overwhelming representation of *P. aeruginosa* (49% of patients) and *S. aureus* (39% of patients) as targeted bacterial species is because these are overall major causes of severe nosocomial infections, but are also the main microorganisms causing invasive burn wound infection[40], which is historically a major focus of attention of the infectiologists of the QAMH, where the first bacteriophage treatments were carried out[41].

Of note, all bacteriophage preparations were offered free of charge. However, this endeavour—providing 43 batches of 26 bacteriophages for the treatment of 100 patients—would not have been possible if one had to comply with the large body of costly and time-consuming requirements and procedures for GMP manufacturing and licensing of biological medicinal products. Companies focusing on defined bacteriophage preparations for use in commercially viable indications might be able to deal with the demanding requirements of the conventional medicinal product (drug) licensing pathway, including GMP certification, preclinical testing and clinical trials. However, for a BT centre, these requirements form an insurmountable barrier in terms of timelines and cost. We experienced first-hand how elaborate and logistically complex personalized BT concepts are, compared with one-size-fits-all approaches, with bacterial strains and matching bacteriophages being exchanged between dozens of institutes in 12 countries. As a result, we are focusing on the development of an instant and on-site production system for bacteriophages based on artificial intelligence (AI) and synthetic biology approaches[42].

Our BT protocols prescribe relatively low bacteriophage doses, usually ~$10^7$ plaque forming units (p.f.u.s) ml$^{-1}$. In the United States, the Antibacterial Resistance Leadership Group (ARLG) Phage Taskforce suggests using the highest safe and tolerated dose of a bacteriophage product with endotoxin levels below the acceptable limits set by the Food and Drug Administration to maximize bacteriophage concentrations at the site of infection and infect as many host cells as possible with the first dose[43]. The ARLG Phage Taskforce, however, acknowledges that clinical outcomes are not always improved with higher doses, reflecting the complexity of effective bacteriophage dosing. We observed an increase in in vitro bacteriophage efficiency (lytic activity) with increasing MOI up to a certain MOI, after which regrowth can be observed more frequently and at an earlier point in time (Extended Data Fig. 4). The effective bacteriophage doses in the body are also determined by the route of bacteriophage administration. Most established BT protocols presented here are based on the principle that bacteriophages are best administered directly into the site of infection. Oral administrations were not used because no gastrointestinal infections were treated.

In 17% (18/106) of targeted infections for which bacteriological follow-up data were available, clinical improvement was reported even though the targeted bacteria were not eradicated.

In 1943, the emergence of bacteriophage-resistant bacterial mutants in liquid cultures was reported[44]. Recently, parallel evolution of bacteriophage resistance and virulence loss in *P. aeruginosa* response to bacteriophage treatment (in one patient), in vivo and in vitro was reported[45]. In vivo selected resistance was associated with

reduced growth rates, whereas in vitro isolates evolved greater biofilm production. Reference 46 showed that when bacteriophage infection risk is high, constitutive resistance mechanisms, such as a mutation of the bacteriophage receptor, are selected by the bacterial hosts, rather than inducible resistance mechanisms, such as a clustered regularly interspaced short palindromic repeats (CRISPR) system[46]. In the present study, we observed in vivo selection of a bacteriophage resistance phenotype in 43.8% (7/16) of patients for which adequate follow-up bacterial samples were available for testing. However, there is a caveat; patients for whom the possible emergence of bacterial bacteriophage resistance could be analysed were treated in only a few hospitals where routine bacterial monitoring generated sufficient suitable samples. This means that the presented bacteriophage resistance data are not generalizable. In addition, due to the limited number of patients in which resistance was demonstrated, it was not possible to statistically show its potential impact on bacterial target eradication. Regardless, failure of eradication was observed in 43% (3/7) of patients with bacteriophage resistance selection and in 22% (2/9) of patients without bacteriophage resistance. Cases where bacteriophage resistance arose were predominantly *P. aeruginosa* respiratory tract infections. It can be that bacteriophage resistance is more common in this scenario, but it may also be because respiratory tract infections and *P. aeruginosa* infections are the most represented. Non-synonymous SNPs or deletions in genes affecting the bacteriophage receptor or coding for a DNA gyrase were assumed to be at the basis of the resistance phenotype in five cases. In two patients, bacterial strains that were not hosts for the applied bacteriophages were selected. In some cases, the in vivo selected bacteriophage-resistant mutants were shown to exhibit re-sensitization to certain antibiotics and reduced virulence in a *G. mellonella* larvae model. The selection of bacteriophage-insensitive bacteria did not prevent the ultimate eradication of the targeted bacterial strains and clinical improvement in four patients.

So far, all BT RCTs have evaluated defined bacteriophage products as stand-alone therapies[4], while bacteriophage–antibiotic synergy is increasingly reported in the literature[47–50]. On the basis of BT clinical data generated in compassionate use settings, in combination with antibiotic therapy, the ARLG Phage Taskforce recently suggested that BT should be used in conjunction with conventional antibiotics[43]. Correspondingly, here we observed a statistically significant correlation between the eradication of the targeted bacteria and adjunctive standard-of-care antibiotic therapy. In addition, in several of the present 100 cases, it was assumed that the clinical resolution of multidrug-resistant infections was due to the additive or synergistic effect of various bacteriophage–antibiotic combinations[6,14,16–21,23–25]. It was hypothesized, on the basis of in vitro experiments, that the therapeutic use of bacteriophages binding to *P. aeruginosa* efflux pumps could select bacteriophage-resistant isolates with changes in the efflux pump mechanism, causing increased sensitivity to certain chemical antibiotics[51]. In the present study, we demonstrated that the therapeutic use of bacteriophage PT07, predicted to bind to the MexAB-OprM multidrug efflux pump, indeed selected (in vivo) bacteriophage-resistant mutants with changes to the efflux pump mechanism, resulting in increased sensitivity to fluoroquinolones. The use of specifically chosen bacteriophages (for example, targeting drug efflux pumps) could therefore re-sensitize bacteria towards antibiotic activity, increasing bacterial killing when used in combination with these antibiotics, and potentially decreasing selection of antibiotic- or bacteriophage-resistant clones. However, caution is warranted, as certain antibiotics can interfere with bacteriophage lytic activity[17,52]. It might thus be advisable to measure potential synergy or antagonism for the proposed combinations of bacteriophages and antibiotics before their clinical application[47].

A considerable body of experimental data has accumulated showing that bacteriophages can substantially affect immune system cells, and it has been assumed that anti-bacteriophage antibodies appearing over the course of BT could decrease the lytic activity of bacteriophages and cause therapeutic failure[53]. Consequently, the use of the same bacteriophage(s) for several weeks was discouraged in the former Soviet Union[54]. More recently, ref. 55 reported on the development of neutralizing antibodies after 2 months of intravenous BT, which led to treatment failure in an immunocompetent patient with *Mycobacterium abscessus* pulmonary infection[55]. The ARLG Phage Taskforce advised considering measurement of neutralizing antibodies during prolonged courses of BT[43]. In the present study, we observed bacteriophage immune neutralization emerging 6–35 days after initiation of invasive bacteriophage administration.

We acknowledge that our analysis, involving 100 severely ill patients for whom BT was a salvage therapy and our primary aim was to help these patients, has intrinsic limitations. No control groups, blinding or randomization were put in place and different medical specialties and infection types were involved. Evaluation of safety and efficacy was not based on pre-defined standardized tests but on the judgement of the treating physicians, and although they were all experienced, this introduces a certain subjectivity. However, we consider that this case series provides key insights that are not only valuable for the treatment of last resort patients, but also for the design of prospective clinical trials such as the PHAGEFORCE study[56]. We confirmed the safety profile of BT and the advantages of combining BT with standard-of-care antibiotic therapy. Statistical analysis showed a significantly higher probability of microbial eradication when BT was combined with standard-of-care antibiotics, and in vitro bacteriophage–antibiotic synergy was demonstrated in 9 of 10 analysed cases. Samples allowing supportive tests in 21 patients, in view of better treatment management, shed more light on some BT issues such as the in vivo selection of bacteriophage resistance, bacteriophage–antibiotics synergy and bacteriophage immune neutralization.

In conclusion, we present evidence that the use of bacteriophages in addition to standard-of-care antibiotics can significantly improve the eradication rate of targeted bacteria in this patient population. These data can be useful for designing future controlled clinical trials that are urgently needed to assist the BT field.

## Methods

### Study design and patients

We reviewed the first 100 consecutive BT cases facilitated by a Belgian consortium between 1 January 2008 and 30 April 2022. Within this consortium, the QAMH coordinated most BT cases, selecting and producing bacteriophages, and suggesting BT protocols, while KU Leuven performed supporting genomic analyses of bacteriophages under consideration and of bacterial genomes, and Sciensano controlled the quality and safety of individual bacteriophage preparations. The choice for 100 patients is arbitrary and not linked to any prospective sample size determination.

Physicians requesting BT with QAMH bacteriophage preparations for their patients submitted a BT request to the Phage Therapy Coordination Centre (PTCC) of the QAMH. The PTCC procedure for selecting patients for BT is depicted in Extended Data Fig. 1 and is largely determined by clinical need, regulatory approval and the availability of bacteriophages targeting the infecting bacteria. Clinical applications were performed by, and under the responsibility of, Bacteriophage Therapy Providers in several hospitals in Belgium and abroad. No blinding, masking or randomization were implemented, and investigators and patients were aware of the bacteriophage treatment. Demographic and clinical data were collected through the patients' treating physicians. Clinical improvement (or not), eradication of the targeted bacterium (or not), and the advent, seriousness and duration of suspected adverse drug reactions and events were assessed by the treating physicians.

Written informed consent for BT was obtained from the involved patients or their legal representatives according to local provisions.

Where warranted, local ethics committee approval for BT was obtained. According to EU Regulation No 536/2014 (Clinical Trials Regulation)[57], its transposition to Belgian Law, and following advice of the Leading Ethical Committee of the 'Universitair Ziekenhuis Antwerpen' and the 'Universiteit Antwerpen' (ID 3644), which approved the observational study protocol, the present retrospective non-interventional analysis of an existing and de-identified BT database was not considered as an experiment on the human person and did not require a dedicated informed consent. There was no patient compensation for participation in this study. The observational study protocol was registered on ClinicalTrials.gov (Study BT100, ID: NCT05498363).

## Manufacture of bacteriophage APIs

Bacteriophages were isolated and characterized by QAMH or were sourced from Bacteriophage Donors. Bacteriophage suspensions were produced in accordance with the guidelines provided by the bacteriophage API monograph[8], and the methods described in ref. 58, with some modifications. Bacteriophage stocks were prepared using the double agar overlay method with minor modifications. Three to six millilitres of bacteriophage lysate containing $10^3$–$10^5$ plaque-forming units (p.f.u.) of bacteriophages were added to a sterile 15 ml Falcon tube (Greiner Bio-One) and complemented with 0.2 ml of a bacteriophage-sensitive bacterial suspension (end concentration of $10^8$ c.f.u.s ml$^{-1}$) and luke-warm medium (Select Alternative Protein Source (APS) lysogeny broth (LB), tryptic soy broth (TSB) or TSB + 0.5% glycerol (all purchased from Becton Dickinson)) with 0.6% top agar (VWR International), to a total volume of 12 ml. This mixture was plated onto a square (12×12 cm) Petri dish (Greiner Bio-One) filled with a bottom layer of APS LB, TSB or TSB medium + 0.5% glycerol (all Becton Dickinson) and 1.5% agar (VWR International), and incubated at 32 °C (for *E. coli*, *K. pneumoniae* and *P. aeruginosa*) or 37 °C (for all the other bacterial species) for 16 h or 48 h (for *M. abscessus*). The top agar layer was scraped off using a sterile L-shaped rod (Sigma Aldrich), transferred to a sterile 50 ml sterile Falcon tube (Greiner Bio-One) and centrifuged for 20 min at 6,000 *g* using a Sorvall Legend centrifuge (Thermo Fisher). The supernatant was aspirated using a sterile 30 ml syringe (BD Plastipak, Becton Dickinson) with an 18G sterile needle (BD microlance 3, Becton Dickinson) and filtered sequentially using a 0.45 μm and a 0.22 μm polyethersulfone (PES) Millex-Gp membrane syringe filter (Merck) or using a vacuum filter system (Nalgene, Thermo Fisher). The bacteriophage suspension was centrifuged for 90 min at 35,000 *g* (40,000 *g* for podoviruses) using a Sorvall Legend centrifuge (Thermo Fisher). The resulting bacteriophage pellet was diluted in ten times less Dulbecco's phosphate buffered saline without calcium and magnesium (DPBS, Lonza) than the initial bacteriophage suspension and the pellet was left to dissolve overnight at 4 °C. The bacteriophage suspension was further diluted to a final concentration of generally $10^9$–$10^{10}$ p.f.u.s ml$^{-1}$ using DPBS (Lonza) and a volume of 150–250 ml. The diluted bacteriophage suspension was filtered using a 0.22 μm PES Millex-Gp membrane syringe filter (Merck) and subsequently purified from endotoxins using the commercially available kits EndoTrap Blue (Lonza) or EndoTrap HD (Lionex), according to manufacturer instructions. One column was utilized per 50 ml of bacteriophage suspension. Endotoxin-purified bacteriophage suspensions were filtered using medical-grade 0.22 μm polyvinylidene difluoride (PVDF) Millex-Gp syringe filters (Merck) and collected into sterile 125 or 500 ml PETG Nalgene bottles (Thermo Fisher). The final titre of each thus obtained bacteriophage API was $10^9$–$10^{10}$ p.f.u.s ml$^{-1}$.

## Quality and safety of bacteriophage APIs

Sciensano controlled the quality and safety of the bacteriophages. In accordance with the bacteriophage API monograph[8], this control was implemented on two levels (https://www.sciensano.be/en/control-and-safety-assessment/safety-therapeutic-bacteriophage-preparations). First, a genetic control was performed to check the

safety of the bacteriophage to be used in human therapy. For this purpose, genomic DNA of the bacteriophages and their bacterial hosts were isolated and purified, respectively using a MagCore Viral Nucleic Acid and an MgC Bacterial DNA kit with a 60 μl elution volume (Atrida), following manufacturer instructions. Sequencing libraries were constructed using the Illumina Nextera XT DNA sample preparation kit and sequenced on an Illumina MiSeq instrument with a 250 bp paired-end protocol (MiSeq v3 chemistry, Illumina). Trimming of short reads was performed with Trimmomatic (v.0.32)[59]. In addition, for bacterial production strains, long-read sequencing was performed using Oxford Nanopore Technologies (ONT)'s rapid barcoding kit SQK-RBK004 and a MinION flow cell (v.9.4.1), according to manufacturer instructions. Super high accuracy base calling was performed using Guppy (v.6.0.1) (ONT) and hybrid assemblies were generated using Unicycler (v.0.4.7)[60]. For bacteriophages, genome assembly was performed using SPAdes (Galaxy v.3.15.4+)[61], after which the genome was annotated using Prokka (Galaxy v.1.14.6)[62] with assistance of the PHROGS v.3 database (https://phrogs.lmge.uca.fr/). To detect undesired genes associated with antibiotic resistance or virulence, the complete bacteriophage genome was submitted to the NCBI (National Center for Biotechnology) blastn web interface (https://blast.ncbi.nlm.nih.gov/Blast.cgi) for a similarity search in different databases: ARG-ANNOT (ARG-ANNOT NT v.6 July 2019), CARD (v.3.1.4 to 3.2.5), ResFinder (https://bitbucket.org/genomicepidemiology/resfinder_db) and VFDB full (downloaded on 20 April 2022). Prophage induction was searched by mapping sequencing reads of the production batch to the bacterial production host genome using Bowtie2 (Galaxy v.2.5.0), and looking for significantly increased coverage in predicted prophage positions using PHASTER (https://phaster.ca/)[63] and Prophage Hunter (https://pro-hunter.bgi.com/)[64].

Second, Sciensano analysed various parameters of each production lot of each bacteriophage API. Bacteriophage identity and purity (scored by the percentage of bacteriophage sequence reads) was determined using DNA extraction and genome sequencing as described above. The potency of the lot was verified using classical double agar dilutions in triplicate. The bioburden (total viable aerobic count) of each bacteriophage API lot was assessed using a validated membrane filtration method based on European Pharmacopoeia (Ph. Eur.) chapter 2.6.12. Briefly, 4 ml of the 150–250 ml bacteriophage API batches (1.6–2.6%) was added to 36 ml of NaCl peptone, after which 10 ml was membrane filtered (Nalgene membrane filter, 0.45 μm). The membrane was then incubated on trypto-casein-soy (TCS) agar at 30–34 °C for at least 72 h and SCG (Sabouraud dextrose agar + chloramphenicol + gentamicin) at 20–24 °C for at least 5 days. After incubation, the number of c.f.u.s per ml of bacteriophage API was determined. Several bacterial and yeast strains were used as positive controls.

Bacterial endotoxin content of 1 ml samples (0.4–0.6%) was determined using a validated *Limulus* Amebocyte Lysate (LAL) test, according to Ph. Eur. chapter 2.6.14. Bacterial endotoxin levels were expressed in endotoxin units (EU) per ml (1 EU is equal to 1 international unit (IU) of endotoxin). The acceptance criterion (endotoxin limit) for the final bacteriophage magistral preparations (diluted bacteriophage APIs) was 5 EU kg$^{-1}$ body mass h$^{-1}$, irrespective of the administration route.

A certificate allowing the bacteriophage API to be used in pharmaceutical (magistral) preparations is provided by Sciensano upon successful completion of this two-tiered procedure.

Sciensano controlled the quality and safety of 43 batches of individual bacteriophage APIs produced by QAMH to treat the first 100 patients. These batches exhibited an average bacteriophage titre of $8.34 \times 10^9$ p.f.u.s ml$^{-1}$ (s.d. $1.16 \times 10^{10}$), a pH of 7.32 (s.d. 0.037), a bioburden of 0 colony-forming units (c.f.u.) ml$^{-1}$ (s.d. 0) and a median endotoxin level of 5 EU ml$^{-1}$ (s.d. 89.14). The bacteriophage APIs, active ingredients of magistral preparations, were diluted in, and/or combined with, the necessary excipients in a hospital pharmacy 'officina' immediately before use on a named-patient basis. The endotoxin limit

for the bacteriophage magistral preparations was defined on the basis of dosage and the patient's weight. The administered endotoxin doses were, irrespective of the administration route, always well below the threshold pyrogenic dose for intravenous administration, that is, <5.0 EU endotoxin kg$^{-1}$ body mass h$^{-1}$. Bacteriophage genomes contained no genetic determinants known to confer lysogeny, toxicity, virulence or antibiotic resistance. Host bacteria used in the manufacturing process were as safe (or least pathogenic) as possible. Some production hosts were shown to contain prophages. Bacteriophage productions with >5% of sequencing reads derived from actively replicating prophages were not used in therapy. Bacteriophage cocktails produced by the Eliava Institute (PyoPhage and IntestiPhage) were not quality-controlled by Sciensano. These products probably have higher endotoxin content and an unknown prophage content. Hence, they were never administered intravenously.

## Selection of adequate bacteriophages for therapy

The patients' infecting bacteria were sent to the PTCC and their bacteriophage susceptibility was determined. Susceptibility of bacterial strains towards the available bacteriophage cocktails or APIs was tested using the spot test as described in ref. [65]. Fresh overnight cultures of the patient's bacterial strains were added to lukewarm (46 °C) media containing 0.6% agar (top agar) and poured onto square (12 ×12 cm) Petri dishes (Greiner Bio-One) containing media with 1.5% agar (bottom agar). Different culture media were used, according to the considered bacterial species. Media were purchased from Becton Dickinson and agar from VWR International. Droplets (10 μl) of serial dilutions of each of the considered bacteriophage solutions were spotted on the top agar layer. Petri dishes were incubated overnight at 32 or 37 °C, according to the considered bacterial species. The next day, the lysis zones produced by active bacteriophages in the bacterial lawn were examined and classified as confluent lysis (4+), semi-confluent lysis (3+), opaque lysis (2+), separate plaques (+) or no activity (−). Next, for bacteriophages producing clear lysis zones, EOP was defined as previously described[65]. The EOP for the patient's bacterial strain was calculated by comparison with a highly susceptible reference host and defined as the observed number of p.f.u.s on the patient's bacterial strain (as determined by the above-described spot test) divided by the observed number of p.f.u.s on the reference bacterial strain. The EOP value obtained with the highly susceptible production host strain was considered as EOP = 1.0. In case the picture was unclear (for example, opaque lysis zones) and the results difficult or un-interpretable, the double agar overlay method was used to determine the p.f.u.s on the patient's strains and the bacteriophage production host, as described above, to define EOP more precisely. When the activity of the bacteriophages was still difficult to assess using the above-mentioned methods based on solid media, liquid broth cultures were used to assess bacteriophage activity, using the OmniLog system (Biolog). Bacterial respiration was measured without and with bacteriophages. Experiments were performed in 96-well plates (Thermo Fisher) in a final volume of 200 μl of LB or TSB medium (Becton Dickinson), supplemented with 100-fold diluted tetrazolium dye mix A or H (Biolog). Bacterial cells were inoculated at a concentration of 10$^5$ c.f.u.s per well, calculated on the basis of optical density (OD) at 600 nm and validated using a classical plate culture method. Bacteriophages were added at an MOI range of 100–0.0001, as calculated on the propagation host. Plates were incubated at a bacterial species-specific temperature (32 or 37 °C) for 72 h, and the colour change caused by reduction of the tetrazolium dye due to bacterial respiration (during growth) was recorded every 15 min by the OmniLog system. The results were analysed with Biolog Data Analysis software (v.1.7) and data were exported to Microsoft Excel files.

We considered the relative EOP as a relative measure of lysis efficiency, which, in this context, is defined as the lytic activity (titre) of the bacteriophage on the patient's bacterial strain, divided by the titre observed in a reference bacterial host known to be highly susceptible to the bacteriophage. We considered an EOP ≥ 0.1 on the patient's bacterial strain as therapeutically acceptable on the basis of the expertise from the Eliava Institute. All bacteriophage cocktails were composed of bacteriophages with compatible activities. Since April 2022, when more than one bacteriophage showed adequate in vitro activity, the overall activity of the bacteriophage combinations was analysed using the OmniLog system, as described above. When synergistic or additive effects were observed, the concerned bacteriophage combinations were recommended for clinical use.

## Pre-adaptation of bacteriophages

When the observed bacteriophage susceptibility was deemed too low for therapeutic application, and if time and resources permitted, bacteriophages were pre-adapted to increase pathogen clearance and to reduce bacteriophage resistance evolution[66–68]. According to the guidelines of the Ministry of Health of the USSR and the empirical experience of the Eliava Institute, adequate bacteriophage cocktails (not individual bacteriophages) should cause stable lysis, that is, without the emergence of bacteriophage-insensitive bacterial mutants, of the target bacteria in liquid medium for a prolonged period (typically 24–48 h), and at an MOI of 0.0001–0.00001 and bacterial concentrations of 10$^6$ c.f.u.s ml$^{-1}$ (refs. [69–72]). For individual bacteriophages, MOIs ≤ 1.0 were deemed appropriate. To obtain these bacteriophage virulence and bacterial regrowth suppression thresholds, the (modified) Appelmans method was applied for the pre-adaptation of bacteriophages on bacterial strains, as previously described[73]. To a 15 ml Falcon tube (Greiner Bio-One) were added: 4.5 ml of LB or TSB medium (Becton Dickinson), 0.5 ml of tenfold dilutions of the considered bacteriophage and a volume of either the patient's bacterial strain or a pre-production panel of collected 'problematic' bacterial strains, to obtain a final concentration of 10$^6$ c.f.u.s ml$^{-1}$. The tubes were incubated at a bacterial species-specific temperature (32 or 37 °C) for 48 h. Bacterial growth and bacteriophage activity were monitored by OD measurement at 600 nm using a Lambda 12 UV/VIS spectrometer (Perkin Elmer) after 24 and 48 h of incubation and compared to two negative controls (bacteriophage only and LB or TSB medium only) and a positive control (bacteria only). The tube with the highest bacteriophage dilution showing an OD$_{600}$ value similar to the negative controls was selected and chloroform was added to a final concentration of 2.0% (v/v). The tube was shaken and incubated for at least 2 h at 2–8 °C. After incubation, the upper phase (without chloroform) was aspirated using a sterile 30 ml syringe (BD Plastipak, Becton Dickinson) with an 18G sterile needle (BD microlance 3, Becton Dickinson) and filtered using a 0.45 μm or a 0.22 μm PES Millex-Gp membrane syringe filter (Merck). The obtained bacteriophage lysate underwent several (at least three) of the above-described passages until adequate virulence and resistance suppression levels were obtained.

The comparison between a bacteriophage and its patient-adapted version was recently published[6]. However, the genetic comparison of pre-adapted phages with their unadapted ancestors falls outside the scope of this study.

## Bacteriophage preparation stability

The stability of the bacteriophage APIs was monitored by determining their titre at 2–8 °C monthly. Bacteriophage APIs with titres of 10$^9$–10$^{10}$ p.f.u.s ml$^{-1}$ retained their activity for at least 1 year[74]. One or more bacteriophage APIs can be diluted and/or mixed with a carrier (for example, an isotonic intravenous solution or a hydrogel) into a magistral preparation under the supervision of a hospital pharmacist and according to the provisions of a medical prescription provided by the patient's treating physician. Diluting and mixing various bacteriophages are events that can compromise their stability[74,75], and experiments showed that, in general, magistral preparations are best used within 1 week after their manufacture.

## Bacteriophage therapy protocols

The PTCC suggested BT protocols on the basis of the application instructions of the Ministry of Health of the USSR[9–11] and the Eliava Institute, some of which can be found in the leaflets of their BT products. These documents (in Russian) do not mention any (published) data. One of them states that 'bacteriophage neutralization can emerge between 10 and 15 days after intravenous application'. We have not been able to determine whether these 30–40-year-old guidelines and instructions may be based on systematic studies, or if they are largely based on empirical experience. Therefore, we prefer to catalogue them as Centre for Evidence-Based Medicine (CEBM) evidence level 5, that is, recommendations formulated by experts on the basis of their own professional experiences. This evidence is probably also based on the review of data from case reports and non-systematic studies.

Bacteriophage administration intervals were largely influenced by clinical indications and administration routes. For instance, it is more straightforward to apply bacteriophages several times per day to the infected lungs of an intubated patient (nebulization) than to infected burn wounds (topical), which are generally unpacked and treated only once a day.

For nebulization of bacteriophage preparations, vibrating mesh type nebulizers were advised because they were shown to induce less titre loss due morphological damage than air-jet nebulizers[76,77]. For bone and orthopaedic prosthesis infections, we advised the use of a pigtail catheter or another draining device for rinsing the wound cavities before bacteriophage application and for the actual administration of bacteriophages[19]. For topical application, we advised mixing of the bacteriophages with an adequate hydrogel[75]. In general, our protocols prescribed relatively low bacteriophage doses, usually ~$10^7$ p.f.u.s $ml^{-1}$, and ranging from $10^6$–$10^7$ p.f.u.s $ml^{-1}$ for continuous intravenous BT to $10^9$ p.f.u.s $ml^{-1}$ for topical BT in a few SSTI cases. In contrast, some clinics prefer the administration of considerably higher doses, for instance, up to $10^{10}$–$10^{11}$ p.f.u.s $ml^{-1}$ for intravenous BT[78,79].

## Diagnostic tests in support of bacteriophage therapy

In addition to bacteriophage susceptibility testing, three BT supportive tests were offered without obligation to the Bacteriophage Therapy Providers to allow for improved BT management: (1) monitoring of the in vivo emergence of bacteriophage resistance using sequential bacterial samples isolated during BT, (2) analysis of the in vitro bacteriophage–antibiotic interactions before the start of BT and (3) evaluation of bacteriophage immune neutralization, or the ability of the patient's serum to neutralize therapeutic bacteriophages.

**In vivo selection of bacteriophage resistance.** The in vivo selection of bacteriophage resistance was monitored using sequential bacterial samples isolated during BT. Bacteriophage susceptibility was evaluated using the methods described earlier. When decreased bacteriophage sensitivity was observed, the isolate's genome was sequenced and analysed to determine the clonality of the isolate (compared with the pre-BT isolate) and to investigate the genetic background for the observed bacteriophage resistance phenotype. For genome sequencing, the method described in ref. 6 was followed with some deviations: for nanopore processing, Guppy (v.6.3.8) (ONT) (base calling, demultiplexing) and Porechop (v.0.2.4) (barcode clipping) (https://github.com/rrwick/Porechop) were used. Genomes were assembled with Unicycler (v.0.4.8)[60] and SNP variants were called using Snippy (v.4.6.0) (https://github.com/tseemann/snippy). For genome annotation and visualization, EggNOG-mapper (v.2.1.8)[80], mobileOG-db (v.1.1.2)[81], Phigaro (v.2.3.0)[82], Circos (v.0.69.8)[83] and GC-profile[84] were used. A pan-genome analysis using Roary (v.3.13.0)[85] from annotated genomes (Prokka v.1.14.6)[62] was performed to create a maximum likelihood phylogenetic tree using core alignment in fasttree (v.2.1.10) visualized with iTOL (itol. embl.de)[86]. For multilocus sequence typing (MLST), genomes were scanned against PubMLST (https://pubmlst.org/) schemes, including

ST111 (O12-1709), ST357 (B14130), ST235 (NCGM2), ST1233 (PcyII-10) PAO1 (ST549) and ATCC 27853 (ST155) as representative genomes/STs. The programs Porechop, Unicycler, Snippy, EggNOG-mapper, Roary, Prokka, Fasttree and MLST were accessed through the Galaxy server (https://usegalaxy.eu/).

***Galleria mellonella* virulence assays.** Ten *P. aeruginosa* isolates (Pa30 (Is 1), Pa30 (Is 3), Pa54 (Is 1), Pa54 (Is 4) and Pa91 (IS 1–6)) were grown in LB broth (Becton Dickinson) to an $OD_{600}$ of 0.25–0.35. One millilitre of the bacterial cultures was centrifuged and resuspended in sterile DPBS (Lonza). *G. mellonella* larvae were grouped in batches of 10 (standardized for weight) and then injected in the hindmost proleg with a 10 μl aliquot of $10^{-5}$ dilutions (±10 c.f.u.s) of the washed bacterial cultures. After infection, the larvae were incubated in the dark at 37 °C. Activity scores were monitored every 6 h and compared to DPBS-injected controls. Activity scores ranged from 0 to 9, based on activity level (with and without stimulation), melanization and survival[17].

**In vitro bacteriophage–antibiotic interactions.** Bacteriophage–antibiotic–bacteria growth kinetics were analysed upon request of the treating physicians using the bacterial and bacteriophage isolates obtained before the start of BT. For patients treated before October 2021, these evaluations were performed retrospectively on bacterial and bacteriophage isolates stored at −20 °C in LB + 20% glycerol (Becton Dickinson). Bacterial respiration was measured using the OmniLog system (Biolog). The growth kinetics of the targeted bacterial pathogens were assessed in the presence of the bacteriophages only, the relevant antibiotics (to be used concomitantly) only and bacteriophage–antibiotic combinations. Experiments were performed in triplicate (biological replicates) in 96-well plates (Thermo Fisher) in a final volume of 200 μl of LB or TSB medium (Becton Dickinson) supplemented with 100-fold diluted tetrazolium dye mix A or H (Biolog). Bacterial cells were inoculated at a concentration of $10^5$ c.f.u.s per well, calculated on the basis of $OD_{600}$ measurements and validated using a classical plate culture method. Antibiotics and bacteriophages were added at subMIC (0.5 × MIC) levels and MOIs ≤ 1.0 (calculated on the propagation host), respectively. The titres of the bacteriophages were confirmed after each experiment using the classical double agar overlay method. Plates were incubated at 37 °C for 72 h and the colour change caused by reduction of the tetrazolium dye due to bacterial respiration (during growth) was recorded every 15 min by the OmniLog system. The results were analysed with Biolog Data Analysis software (v.1.7) and data were exported to Microsoft excel files. We defined bacteriophage–antibiotic combinations as synergistic when the bacterial growth suppression period produced by the addition of both the bacteriophage and the antibiotic is clearly longer than the simple sum of the suppression periods induced by the bacteriophage and the antibiotic separately.

**Bacteriophage immune neutralization.** The possible emergence of bacteriophage immune neutralization, or the ability of the patient's serum to neutralize therapeutic bacteriophages, was evaluated according to ref. 87, with some modifications. Whole blood samples were collected before BT initiation and at various time points during and after bacteriophage application. Blood was allowed to clot for at least 30 min in a vertical position and then centrifuged in a swinging bucket rotor for 10 min at 2,000 g at room temperature. The obtained serum samples were stored at −80 °C ± 5 °C. To assess the effect of the serum samples on bacteriophage lytic activity, 0.9 ml of 1:100 diluted sera was mixed with 0.1 ml of the bacteriophage suspension at a concentration of 2 × $10^7$ p.f.u.s $ml^{-1}$ and incubated for 30 min at 37 °C. Bacteriophage lytic activity (titre) was determined before and after incubation with the patient's serum samples using the double agar overlay plaque assay (as previously described). Comparison of pre- and post-incubation lytic activity allowed for the determination of the proportion of neutralized bacteriophages. Each serum sample was tested in triplicate.

## Clinical outcome

Clinical improvement, eradication of the targeted bacterium and the advent, seriousness and duration of suspected adverse drug reactions or events were assessed by the treating physicians. Neither safety data were prospectively collected, nor were the descriptors defined in advance to clinicians.

## Data collection

Before BT, demographic and clinical data were collected through a medical form, which was completed by the Bacteriophage Therapy Providers. The medical doctor's BT prescription, information regarding the applied bacteriophage product and its administration route, dosage, duration and information regarding possible concomitant (antibiotic) treatments were also recorded. The 'phagograms' reporting on the evaluation of the bacteriophage susceptibility of the patient's bacterial isolates sampled before and sometimes during treatment were also archived. If the bacteriophage treatment was performed in a hospital, a clinical follow-up form requesting information about the clinical outcome (including suspected adverse drug reactions and events) was completed by the treating physician and the nursing team and sent to the PTCC. In case of ambulatory BT, clinical follow-up information was collected directly from the patients. All demographic, bacteriophage product and clinical data were recorded in a Research Electronic Data Capture (REDCap) designed database[88]. Data collection and analysis were not performed blind to the conditions of the experiments.

## Definitions

In accordance with the guidelines of an international expert proposal for interim standard definitions for acquired resistance, multidrug resistance (MDR) was defined as acquired non-susceptibility to at least one agent in three or more antimicrobial categories, extensive drug resistance (XDR) as non-susceptibility to at least one agent in all but two or fewer antimicrobial categories, and pandrug resistance (PDR) as non-susceptibility to all agents in all antimicrobial categories[89]. The term 'usual drug resistance' (UDR) was used to describe isolates that are not fully susceptible, but could nonetheless be readily treated (at least on the basis of the in vitro susceptibility assays) using standard therapies[90]. If an infection persisted for more than 6 months, it was considered a 'chronic infection'. Clinical improvement was defined as the improvement of at least one symptom associated with the bacterial infection, as assessed by the treating physician. No clinical metrics were applied (for example, illness severity scores). The influence of other (medical/surgical) interventions was not determined. Eradication of the targeted bacterium was defined as the absence of the originally targeted causative agent of the bacterial infection in culture, or when the patient's treating physician concluded, on the basis of a follow-up survey, that the patient was freed of the targeted bacterial pathogen. Microbiological eradication was not prospectively or systematically evaluated. The period between the start of BT and the evaluation of the clinical outcome varied according to the treating physician and the indication, and ranged from 1 month to 1 year, the latter for difficult-to-treat bone infections.

## Statistical methods

The following variables were analysed for 92 of the 100 patients (for which a complete dataset was available): eradication of the targeted bacteria, clinical improvement, concomitant use of antibiotics, antibiotic resistance profile of the target bacteria, suspected adverse drug reactions and the clinical setting (ambulatory treatment or hospitalized). All these variables were binary categorical. In addition, the 14 infection types and 21 bacterial species targeted by BT were monitored on nominal categorical scales. Age and gender were analysed on numeric scales. The statistical analysis was conducted using the statistical software environment SAS (v.9.4). We used a stepwise, forward selection procedure on a reduced dataset (Supplementary Table 2) to determine the most informative variable in the dataset, with

the variable 'Eradication (ERADIC)' as response variable for our logistic regression model. The probability modelled is ERADIC = 'Yes' (that is, successful eradication). A sketch (left) and the contingency table (right) of the logistic regression model used to analyse the reduced dataset (Supplementary Table 2) are depicted below.

Logit model: $\pi = \dfrac{1}{1+e^{-(b_0+b_1 x)}}$

$b_0$: Intercept = 0.74 (p<0.01)
$b_1$: No concommitant AB = -1.151 (p<0.01)

Odds-ratio: $\dfrac{\pi}{1-\pi}$ = 0.309

$\pi$ = probability that eradication is "Yes"
$1-\pi$ = probability of "No" eradication

Contigency table (n=92)

| | | Concomittant AB | |
|---|---|---|---|
| | | No | Yes |
| Eradication | No | 20 | 19 |
| | Yes | 13 | 40 |

Fisher's exact test was performed using R (v.4.3.0) (https://www.R-project.org/)[91], and the R Stats Package (v.4.3.2) was used to search for significant correlations between variables. The data presented in Fig. 1 (patient population characteristics) and Fig. 4 (bacteriophage immune neutralization) were analysed using R (v.4.3.0) and visualized with the following packages: tidyverse (v.2.0.0)[92], UpSetR (v.1.4.0)[93], ggmap (v.3.0.2)[94] and rnaturalearth (R package version 0.3.2.9000)[95]. The log-rank test with Bonferroni correction for multiple comparisons (GraphPad v.0.5.1) was used for *G. mellonella* survival curve comparisons.

## Reporting summary

Further information on research design is available in the Nature Portfolio Reporting Summary linked to this article.

## Data availability

Detailed clinical protocols, results and additional data are available in the paper and in Supplementary Tables 1 and 2. The protocol for the retrospective, observational study is available at https://clinicaltrials.gov/ct2/show/NCT05498363?term=NCT05498363&draw=2&rank=1. The bacteriophage genome sequences can be retrieved in the GenBank database under the accession codes listed in Supplementary Table 3. The genome data of the bacterial isolates can be accessed via NCBI BioProject PRJNA975428. All other data supporting the findings of this study are available within the paper. Readers can apply for access to data, which will be supplied in compliance with the obligations and responsibilities that the investigators hold for the patients involved in the study. Source data are provided with this paper.

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

## Acknowledgements

Most research and development and manufacturing costs were borne by the Royal Higher Institute for Defense, the QAMH, Sciensano and KU Leuven. C.C., G.S., M.M. and T.G. were supported by the Royal Higher Institute for Defence, Brussels, Belgium. C.L. was supported by a Postdoctoral Mandate from KU Leuven (PDMt2/21/038) and a junior postdoctoral fellowship from the Research Foundation - Flanders (FWO) (12D8623N). This work would not have been possible without the support of E. Kutter, M. Vaneechoutte, R. Adamia, A. Vanderkelen, P. Neirinckx and G. Laire. The funders had no role in the conceptualization, design, data collection, analysis, decision to publish or preparation of the manuscript.

## Author contributions

M.M., G.S., J.G., N.C., M.K., C.C., T.G., S.D.S. and Bacteriophage Donors isolated, characterized and made bacteriophages available for clinical use. M.M., G.S. and J.G. performed the in vitro matching (phagogram) of bacteriophages to the patients' infecting bacteria. M.M. and B.V.N. performed the bacteriophage–antibiotic interaction assays. M.M., G.S. and J.G. produced the bacteriophage APIs and a substantial part of the magistral bacteriophage preparations. M.d.J., P.-J.C., J.W., C.L., S.G., R.L. and Bacteriophage Donors sequenced and analysed the genomes of the selected bacteriophages and bacteria. S.G. designed and performed the *Galleria mellonella* experiments. M.M., B.V.N. and J.O. performed the bacteriophage immune neutralization assays. M.d.J., P.-J.C., J.-P.D. and G.V. designed the quality management system, and performed and validated the quality control of the bacteriophage preparations. S.D., J.O., B.V.N., N.C., M.K., P.S. and J.-P.P. screened the BT requests, selected and coordinated BT cases, including the suggestion of BT protocols. S.D., J.O., B.V.N., N.C., M.K., T.R., J.-P.P. and Bacteriophage Therapy Providers facilitated or performed BT cases. S.D., A.S., E.V.B., D.D.V. and J.-P.P. collected and summarized all relevant clinical and epidemiological data. E.S., C.L. and J.-P.P. performed the statistical analysis. C.L., S.G. and J.-P.P. drafted the figures and tables. J.-P.P. prepared the first draft of the paper. All authors revised the final version and agreed to the submission of the final version.

## Competing interests

R.N.-P. has been a scientific consultant for BiomX Inc., and has participated and served as a principal investigator and member of Data Safety Monitoring Boards for a clinical trial by Technophage S. A. M.D.L. and N.C. are co-founders of the startup Fagoterapia Lab s.r.l. T.F. is the principal investigator of the PhagoDAIR I study and is consultant for PHAXIAM to conceive clinical trials (no direct funding, contract signed with hospices Civils de Lyon). R.N.-.P, S.M., T.F. and J.-P.P., respectively, serve as Chair, Secretary, Education Officer/Clinical Officer and Science Officer of the European Society of Clinical Microbiology and Infectious Diseases (ESCMID) Study Group for Non-Traditional Antibacterial Therapy (ESGNTA), Basel, Switzerland.

## Additional information

**Extended data** is available for this paper at https://doi.org/10.1038/s41564-024-01705-x.

**Correspondence and requests for materials** should be addressed to Jean-Paul Pirnay.

Jean-Paul Pirnay ⓘ [1,2,70] ✉, Sarah Djebara[3,70], Griet Steurs[1], Johann Griselain[1], Christel Cochez[1], Steven De Soir[1], Tea Glonti[1], An Spiessens[3], Emily Vanden Berghe[3], Sabrina Green ⓘ [4], Jeroen Wagemans ⓘ [4], Cédric Lood ⓘ [4], Eddie Schrevens[5], Nina Chanishvili[6], Mzia Kutateladze ⓘ [6], Mathieu de Jode[7], Pieter-Jan Ceyssens[7], Jean-Pierre Draye[1], Gilbert Verbeken[1], Daniel De Vos[1], Thomas Rose[1], Jolien Onsea[8], Brieuc Van Nieuwenhuyse ⓘ [9], Bacteriophage Therapy Providers*, Bacteriophage Donors*, Patrick Soentjens[3,71], Rob Lavigne ⓘ [4,71] & Maya Merabishvili[1,71]

[1]Laboratory for Molecular and Cellular Technology, Queen Astrid Military Hospital, Brussels, Belgium. [2]European Society of Clinical Microbiology and Infectious Diseases (ESCMID) Study Group for Non-traditional Antibacterial Therapy (ESGNTA), Basel, Switzerland. [3]Center for Infectious Diseases, Queen Astrid Military Hospital, Brussels, Belgium. [4]Laboratory of Gene Technology, Department of Biosystems, KU Leuven, Leuven, Belgium. [5]Department of Biosystems, KU Leuven, Leuven, Belgium. [6]Eliava Institute of Bacteriophages, Microbiology and Virology, Tbilisi, Georgia. [7]Bacterial Diseases, Sciensano, Brussels, Belgium. [8]Department of Trauma Surgery, University Hospitals Leuven; Department of Development and Regeneration, KU Leuven, Leuven, Belgium. [9]Institute of Experimental and Clinical Research, Pediatric Department, UCLouvain, Brussels, Belgium. [70]These authors contributed equally: Jean-Paul Pirnay, Sarah Djebara. [71]These authors jointly supervised this work: Patrick Soentjens, Rob Lavigne, Maya Merabishvili. *Lists of authors and their affiliations appear at the end of the paper. ✉e-mail: jean-paul.pirnay@mil.be

## Bacteriophage Therapy Providers

**Kim Win Pang[3], Willem-Jan Metsemakers[8], Dimitri Van der Linden[10], Olga Chatzis[10], Anaïs Eskenazi[11], Angel Lopez[12], Adrien De Voeght[13], Anne Françoise Rousseau[14], Anne Tilmanne[15], Daphne Vens[16], Jean Gérain[17], Brice Layeux[18], Erika Vlieghe[19], Ingrid Baar[20], Sabrina Van Ierssel[20], Johan Van Laethem[21], Julien Guiot[22], Sophie De Roock[23], Serge Jennes[23], Saartje Uyttebroek[24], Laura Van Gerven[24], Peter W. Hellings[24], Lieven Dupont[25], Yves Debaveye[26], David Devolder[27], Isabel Spriet[28], Paul De Munter[29], Melissa Depypere[30], Michiel Vanfleteren[31], Olivier Cornu[32], Stijn Verhulst[33], Tine Boiy[33], Stoffel Lamote[34], Thibaut Van Zele[35], Grégoire Wieërs[36], Cécile Courtin[37], David Lebeaux[38], Jacques Sartre[39], Tristan Ferry[2,40], Frédéric Laurent[41], Kevin Paul[42], Mariagrazia Di Luca[43], Stefan Gottschlich[44], Tamta Tkhilaishvili[45], Novella Cesta[46], Karlis Racenis[47], Telma Barbosa[48], Luis Eduardo López-Cortés[49], Maria Tomás[50], Martin Hübner[51], Truong-Thanh Pham[52], Paul Nagtegaal[53], Jaap Ten Oever[54], Johannes Daniels[55], Maartje Loubert[56], Ghariani Iheb[57], Joshua Jones[58], Lesley Hall[59] & Matthew Young[60]**

[10]Pediatric Infectious Diseases, Pediatric Department, Cliniques Universitaires Saint-Luc, Université Catholique de Louvain-UCLouvain, Brussels, Belgium. [11]Clinic of Infectious Diseases, Erasme Hospital, Brussels, Belgium. [12]HNO-Zentrum Simmering, Vienna, Austria. [13]Department of Medicine, Division of Hematology, Centre Hospitalier Universitaire de Liège, University of Liège, Liège, Belgium. [14]Department of Intensive Care and Burn Center, University Hospital, University of Liège, Liège, Belgium. [15]Faculté de Médecine, Université Libre de Bruxelles, Brussels, Belgium. [16]Division of Pediatric Infectious Diseases and Infection Prevention and Control, Hôpital Universitaire des Enfants Reine Fabiola, Université Libre de Bruxelles (ULB), Brussels, Belgium. [17]Department of Internal Medicine, Delta Hospital (CHIREC), Brussels, Belgium. [18]Delta Hospital (CHIREC), Brussels, Belgium. [19]Department of General Internal Medicine, Infectious Diseases and Tropical Medicine, University Hospital Antwerp, Edegem, Belgium. [20]Department of Critical Care Medicine, University Hospital Antwerp, University of Antwerp, Edegem, Belgium. [21]Department of Internal Medicine and Infectious Diseases, Vrije Universiteit Brussel, Universitair Ziekenhuis Brussel, Brussels, Belgium. [22]Pneumology Department, CHU Liège, Domaine Universitaire du Sart-Tilman, Liège, Belgium. [23]Queen Astrid Military Hospital, Brussels, Belgium. [24]Department of Otorhinolaryngology, Head and Neck Surgery, University Hospitals Leuven; Experimental Otorhinolaryngology, Rhinology Research, Department of Neurosciences, KU Leuven, Leuven, Belgium. [25]Department of Pneumology, University Hospitals Leuven; Respiratory Diseases and Thoracic Surgery, Department of Chronic Diseases and Metabolism, KU Leuven, Leuven, Belgium. [26]Department of Intensive Care Medicine, University Hospitals Leuven; Department of Cellular and Molecular Medicine, KU Leuven, Leuven, Belgium. [27]Pharmacy Department, University Hospitals Leuven, Leuven, Belgium. [28]Pharmacy Department, University Hospitals Leuven; Clinical Pharmacology and Pharmacotherapy, Department of Pharmaceutical and Pharmacological Sciences, KU Leuven, Leuven, Belgium. [29]Department of General Internal Medicine, University Hospitals Leuven; Department of Microbiology, Immunology and Transplantation, KU Leuven, Leuven, Belgium. [30]Department of Laboratory Medicine, University Hospitals Leuven; Laboratory of Clinical Bacteriology and Mycology, KU Leuven, Leuven, Belgium. [31]St-Jozefskliniek Izegem, Izegem, Belgium. [32]Department of Orthopaedic Surgery, University Hospital Saint Luc, Brussels, Belgium. [33]Department of Pediatrics, Antwerp University Hospital, Edegem, Belgium. [34]Department of Intensive Care Medicine, AZ Groeninge, Kortrijk, Belgium. [35]Department of Otorhinolaryngology, UZ Ghent University Hospital, Ghent, Belgium. [36]Service de Médecine Interne Générale, Clinique Saint Pierre, Ottignies, Belgium. [37]Service de Dermatologie, Cliniques Saint-Pierre, Ottignies, Belgium. [38]Service de Microbiologie, Unité Mobile d'Infectiologie, AP-HP, Hôpital Européen Georges Pompidou, Paris, France. [39]Laboratoire de biologie médicale, Centre Hospitalier de Valence, Valence, France. [40]Centre de Référence des Infections Ostéo-Articulaires Complexes de Lyon (CRIOAc Lyon), Hospices Civils de France, Lyon, France. [41]Institut des agents Infectieux - Hospices Civils de Lyon, Centre International de Recherche en Infectiologie, Lyon, France. [42]University Children's Hospital, University Medical Center Hamburg-Eppendorf, Hamburg, Germany. [43]Department of Biology, University of Pisa, Pisa, Italy. [44]Praxis Cordes and Gottschlich and Heß and Scherl, Rendsburg, Germany. [45]Department of Cardiothoracic and Vascular Surgery, Charité German Heart Center, Berlin, Germany. [46]PhD Course in Microbiology, Immunology, Infectious Diseases, and Transplants (MIMIT), University of Rome Tor Vergata, Rome, Italy. [47]Department of Biology and Microbiology, Riga Stradins University, Riga, Latvia. [48]Department of Pediatrics, Maternal Child Center of the North (CMIN), University Hospital Center of Porto (CHUP), Porto, Portugal. [49]Enfermedades Infecciosas y Microbiología Clínica, Hospital Universitario Virgen Macarena, Sevilla, Spain. [50]Translational and Multidisciplinary Microbiology Group (MicroTM) - Institute of Biomedical Research A Coruña, Microbiology Department of Hospital of A Coruña (CHUAC), University of A Coruña (UDC), A Coruña, Spain. [51]Department of Visceral Surgery, Lausanne University Hospital CHUV, University of Lausanne (UNIL), Lausanne, Switzerland. [52]Division of Infectious Diseases, Department of Medicine, Geneva University Hospitals, Geneva, Switzerland. [53]Department of Otorhinolaryngology and Head and Neck Surgery, Erasmus MC, Rotterdam, the Netherlands. [54]Department of Internal Medicine and Radboud Centre for Infectious Diseases, Radboud University Medical Center, Nijmegen, the Netherlands. [55]Department of Pulmonary Medicine, Amsterdam UMC Locatie VUmc, Amsterdam, the Netherlands. [56]Meander Medisch Centrum, Amersfoort, the Netherlands. [57]Clinique Saint Augustin, Tunis, Tunisia. [58]Edinburgh Medical School: Biomedical Sciences, University of Edinburgh, Edinburgh, UK. [59]Diabetes and Endocrinology, Queen Elizabeth University Hospital, Glasgow, UK. [60]Diabetes Foot Clinic, Royal Infirmary, Edinburgh, UK.

## Bacteriophage Donors

**Nana Balarjishvili[6], Marina Tediashvili[6], Yigang Tong[61], Christine Rohde[62], Johannes Wittmann[62], Ronen Hazan[63], Ran Nir-Paz[2,63], Joana Azeredo[64], Victor Krylov[65], David Cameron[66], Melissa Pitton[66], Yok-Ai Que[66], Gregory Resch[67], Shawna McCallin[2,68], Matthew Dunne[68] & Samuel Kilcher[69]**

[61]College of Life Science and Technology, Beijing University of Chemical Technology, Beijing, China. [62]Leibniz Institute DSMZ, German Collection of Microorganisms and Cell Cultures GmbH, Braunschweig, Germany. [63]Israeli Phage Therapy Center (IPTC) of Hadassah Medical Center and the Hebrew University, Jerusalem, Israel. [64]Department of Biological Engineering, University of Minho, Braga, Portugal. [65]Mechnikov Research Institute of Vaccines and Sera, Moscow, Russia. [66]Department of Intensive Care Medicine, Bern University Hospital, University of Bern, Bern, Switzerland. [67]Laboratory of Bacteriophages, Lausanne University Hospital, Lausanne, Switzerland. [68]Department of Neuro-Urology, Balgrist University Hospital, University of Zürich, Zürich, Switzerland. [69]Institute of Food, Nutrition and Health, ETH Zurich, Zurich, Switzerland.

**Extended Data Table 1 | Baseline characteristics of the first 100 consecutive patients treated with bacteriophages**

| Baseline characteristics of the 100 analysed patients | |
| --- | --- |
| Number of cases | 100 |
| Sex (female), % (n)* | 43.3% (39) |
| Age group, % (n)* | |
| 0 to < 24 months | 5.6% (5) |
| 2 to < 20 years | 13.3% (12) |
| 20 to < 40 years | 14.4% (13) |
| 40 to < 60 years | 33.3% (30) |
| 60 to < 80 years | 28.9% (26) |
| 80 to < 100 years | 4.4% (4) |
| Care setting, % (n) | |
| Hospitalized | 77% (77) |
| Ambulatory care | 21% (21) |
| Hospitalized & ambulatory care | 2% (2) |
| Regulatory context, % (n) | |
| Standard-of-care with magistral bacteriophage preparations | 48% (48) |
| Article 37 of the Declaration of Helsinki | 39% (39) |
| Standard-of-care with unlicensed medicines | 10% (10) |
| 'Autorisation Temporaire d'Utilisation' of magistral preparations | 3% (3) |
| Infection types, % (n)** | |
| Lower respiratory tract infection | 25.4% (29) |
| Skin & soft tissue infection | 22.8% (26) |
| Bone infection | 14.0% (16) |
| Upper respiratory tract infection | 11.4% (13) |
| Bloodstream infection | 8.8% (10) |
| Abdominal infection | 5.3% (6) |
| Orthopaedic prostheses infection | 5.3% (6) |
| Urinary tract infection | 1.8% (2) |
| Other | 5.3% (6) |
| Antibiotic resistance profile of targeted infections, % (n)** | |
| Usual drug resistance | 47.4% (54) |
| Multidrug resistance | 25.4% (29) |
| Extensive drug resistance | 20.2% (23) |
| Pandrug resistance | 5.3% (6) |
| Extensive drug resistance & multidrug resistance | 0.9% (1) |
| Extensive drug resistance & usual resistance | 0.9% (1) |
| Concomitant standard-of-care antibiotic treatment, % (n)** | 69.3% (79) |

*n = 90 (for 10 patients, age and gender were not disclosed) **n = 114 (including 14 second-site infections).

**Extended Data Table 2 | Characteristics of the bacteriophage therapy cases that necessitated pre-adaptation of bacteriophages**

| Patient number | Infection type | Targeted bacterial species | Pre-adapted bacteriophage(s) | # serial passages used for pre-adaptation | Propagation strain used in production | Clinical improvement | Eradication of the targeted bacteria |
|---|---|---|---|---|---|---|---|
| 9 | Fracture-related infection | *Klebsiella pneumoniae* | M1 | 15 | Patient strain | Yes | Yes |
| 16 | Cystic fibrosis lung transplant infection | *Achromobacter xylosoxidans* | JWAlpha, JWDelta, JWT, and 2-1 (APC 1.1 and APC 2.1) | 3 | Patient strain | Yes | Yes |
| 40 | Chronic osteomyelitis | *Bacteroides fragilis* | UZM3 | 4 | Patient strain | No | No |
| 43 | Lung transplant infection | *Enterococcus faecium* | EfgrKN and EfgrNG | 2 | Patient strain | Yes | No |
| 46 | Disseminated bronchiectasis | *Staphylococcus aureus* *Stenotrophomonas maltophilia* | ISP BUCT700 | 6 2 | Patient strain | Yes | No |
| 55 | Prosthetic knee infection | *Staphylococcus epidermidis* | ISP* | 4 | ATCC6538 | No | No |
| 66 | Cystic fibrosis lung infection | *Mycobacterium abscessus* | 8UZL | 5 | Patient strain | No | No |
| 82 | Cystic fibrosis lung infection | *Pseudomonas aeruginosa* | 4P and DP1 | 3 | 573 | Yes | Yes |

*Staphylococcus aureus* bacteriophage ISP was pre-adapted (using 4 serial passages), on five strains, from five different patients to better target *Staphylococcus epidermidis* strains.

**Extended Data Table 3 | Suspected adverse drug reactions and events in the 100 consecutive bacteriophage therapy cases, reported using EudraVigilance terminology**

| Patient number | Drug (Bacteriophage product) | Route of administration | Duration of administration (days) | Indication | MedDRA LLT | Duration (days) | Relatedness of drug to reaction/event | Action taken with drug | Outcome | Seriousness |
|---|---|---|---|---|---|---|---|---|---|---|
| 3 | BFC 1 | Respiratory (inhalation) | 4 | Lower respiratory tract infection | Septic shock | 2 | Not suspected | Drug withdrawn | Fatal | Death |
| 11 | BFC 2 | Respiratory (inhalation) and oral | 10 | Lower respiratory tract infection | Coughing after drug inhalation* | 6 | Suspected | Dose not changed | Recovered/ resolved | Not serious |
| 20 | BFC 1 | Intralesional and intravenous | 7 (intralesional) 86 (intravenous) | Abdominal and bloodstream infection | Abdominal-discomfort* | 2 | Suspected | Drug withdrawn | Recovered/ resolved | Not serious |
| 31 | ISP | Nasal | 21 | Ear, nose and throat infection | Rash lips* | 1 | Suspected | Drug withdrawn | Recovered/ resolved | Not serious |
| 39 | ISP | Intralesional | 10 | Bone infection | Fever* | 1 | Suspected | Dose not changed | Recovered/ resolved | Not serious |
| 42 | PyoPhage (Eliava) | Intralesional | 7 | Bone infection | Application site redness and pain* | 1 | Suspected | Dose not changed | Recovered/ resolved | Not serious |
| 44 | M1 | Respiratory (inhalation) and intravesical | 14 (respiratory) 10 (intravesical) | Lower respiratory tract and urinary tract infection | Septic shock | 11 | Not suspected | Drug withdrawn | Fatal | Death |
| 58 | ISP | Topical | 6 | Diabetic foot infection | Heart failure | 2 | Not suspected | Drug withdrawn | Recovered/ resolved | Life threatening |
| 69 | M1 | Intralesional | 18 | Abdominal infection | Cardiogenic shock | Unknown | Not suspected | Drug withdrawn | Fatal | Death |
| 79 | PNM and PT07 | Intravenous | 4 | Chronic spondylodiscitis | Postoperative ileus | 1 | Not suspected | Drug withdrawn | Fatal | Death |
| 88 | E4 and Efs7 | Intra-articular and intravenous | 3 (intra-articular) 15 (intravenous) | Bone infection | Body temperature increased* | 1 | Suspected | Dose not changed | Recovered/ resolved | Not serious |
| 93 | 14-1 | Intralesional and respiratory (inhalation) | 7 (intralesional) 14 (respiratory) | Empyema and spinocellular carcinoma | Tumour progression and palliative care | 10** | Not suspected | Drug withdrawn | Fatal | Death |
| 96 | 14-1, PNM and PT07 | Topical and intravenous | 1 (topical) 5 (intravenous) | Burninfection and bloodstream infection | Septic shock | 4 | Not suspected | Drug withdrawn | Fatal | Death |
| 99 | ISP | Nasal | 21 | Chronic sinusitis | Diarrhoea and abdominal pain* | 25 | Suspected | Dose not changed | Recovered/ resolved | Not serious |
| 100 | ISP | Intralesional | 7 | Surgical wound infection with fistula | Nausea | 1 | Not suspected | Dose not changed | Recovered/ resolved | Not serious |

BT, bacteriophage therapy; LLT, Lowest Level Term; MedDRA, Medical Dictionary for Regulatory Activities; NA, not applicable. *Considered to be a suspected adverse drug reaction, as a causal relationship between BT and the event was suspected and reported. **The patients died 10 days after the start of palliative care and the discontinuation of BT.

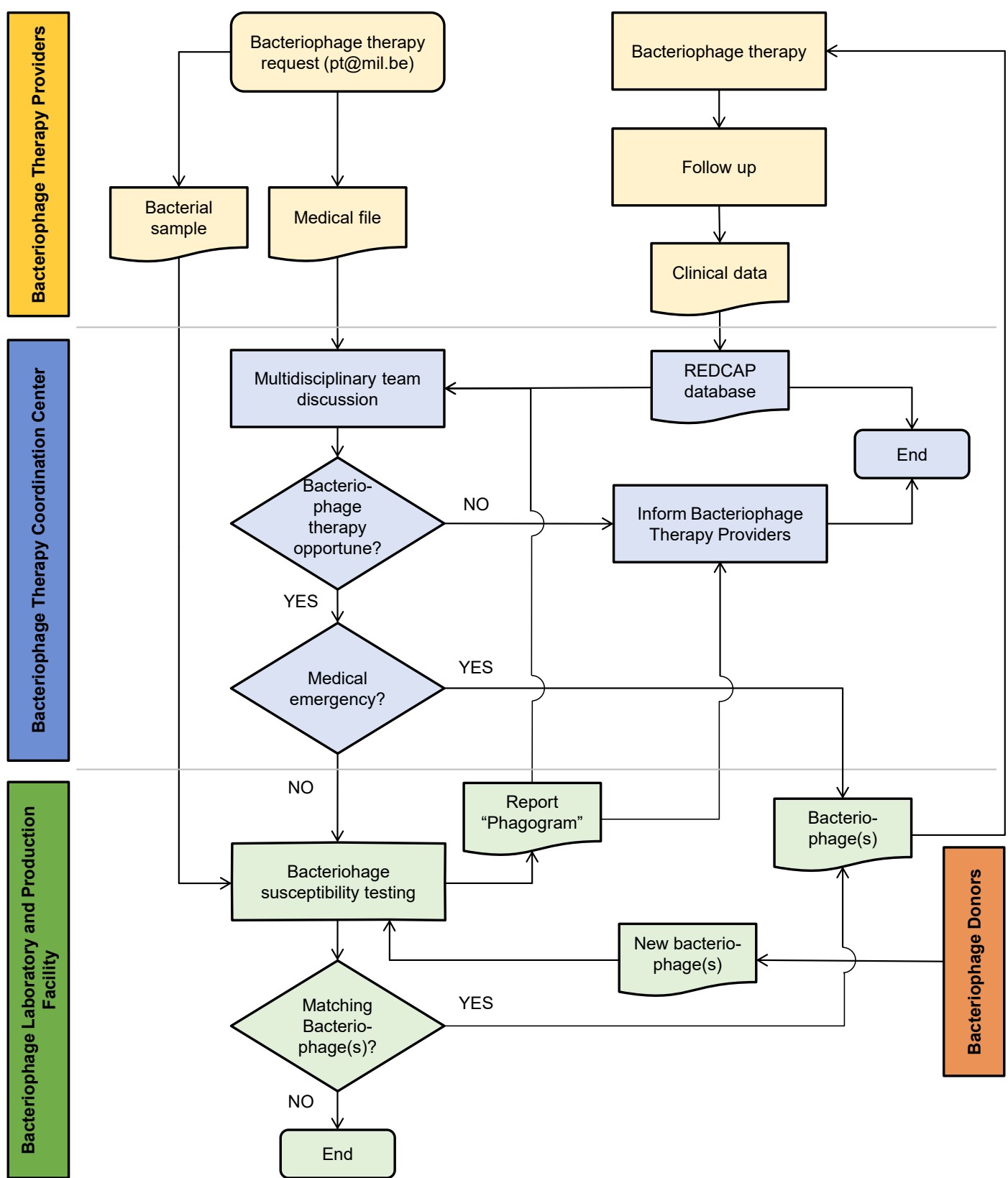

**Extended Data Fig. 1 | The Phage Therapy Coordination Centre's patient selection process for bacteriophage therapy.**

| POSITION | EFFECT | LOCUS_TAG | ANNOTATED PRODUCT | PREDICTION HHPRED/HMMER/Phyre |
|---|---|---|---|---|
| 41566 | missense_variant c.1745A>C p.Lys582Thr | HOQ69_gp039 | Structural protein (ESI-MS) | Putative carbohydrate-binding domain protein (Staphylococcus virus K) |
| 43215 | missense_variant c.1068G>T p.Lys356Asn | HOQ69_gp041 | Structural protein (ESI-MS) | Putative receptor binding protein (Staphylococcus virus K) |
| 60502 | missense_variant c.91T>C p.Ser31Pro | HOQ69_gp060 | Putative DNA polymerase | Uracil-DNA glycosylase |
| 82393 | missense_variant c.170G>A p.Ser57Asn | HOQ69_gp095 | Hypothetical protein | / |

**Extended Data Fig. 2 | Missense mutations in the pre-adapted variant of bacteriophage ISP, as compared to the original clone (before adaptation).** HHpred (https://www.sciencedirect.com/science/article/pii/ S0022283617305879), HMMR (https://nar.oxfordjournals.org/content/46/W1/ W200), and Phyre (https://www.nature.com/articles/nprot.2009.2) were used for functional prediction. ESI-MS, electrospray ionization mass spectrometry.

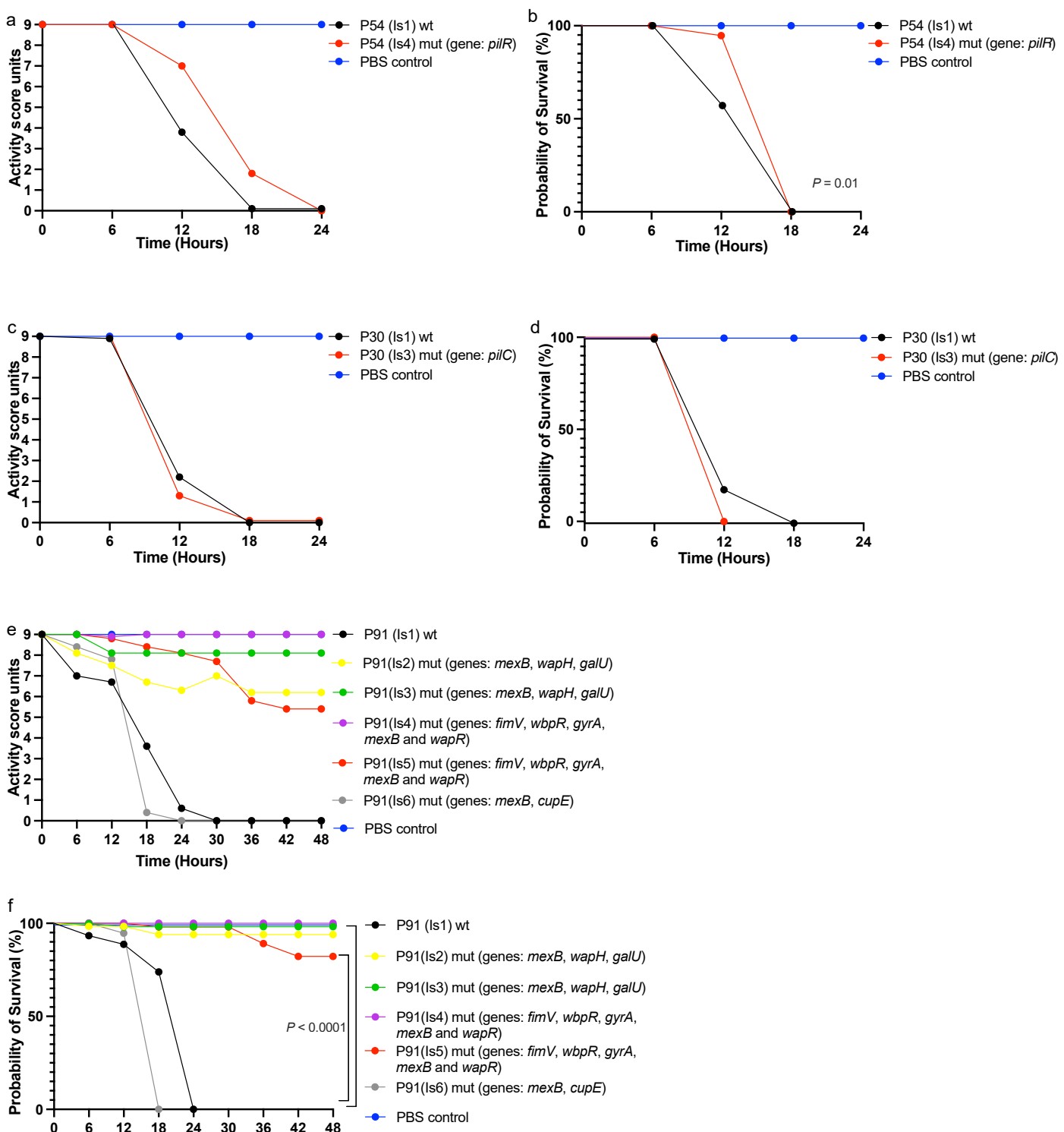

**Extended Data Fig. 3 | Kaplan-Meier plots and activity scores of *Galleria mellonella* larvae post-infection.** Ten larvae in each group were either inoculated with phosphate buffered saline (PBS, control), with the initial bacteriophage-susceptible isolates (wild type, wt), or with the in vivo selected bacteriophage-insensitive mutants of the *Pseudomonas aeruginosa* strains isolated from patient (P) 91, 54, and 30. **a-b**, P54 (Is1 and 4). **c-d**, P30 (Is1 and 3). **e-f**, P91 (Is1 to 6). Mean values of activity scores are represented by a dot symbol. *P* values were calculated using the log-rank test with Bonferroni correction for multiple comparisons. Is, isolate; mut, mutation.

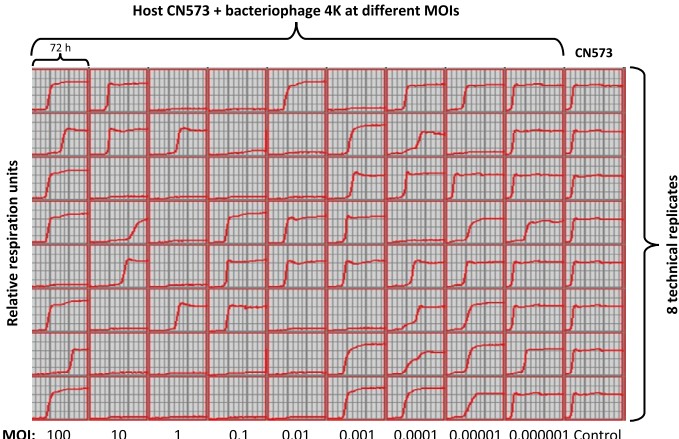

**Extended Data Fig. 4 | Results of the in vitro evaluation of the influence of serial multiplicities of infection (MOIs) on the virulence and on the resistance suppression of *Pseudomonas aeruginosa* bacteriophage 4 K on** the bacterial host strain CN573, as determined in liquid culture, using an OmniLog® system. Bacterial proliferation is presented through relative units of cellular respiration over time (72 h).

# Reporting Summary

## Statistics

For all statistical analyses, confirm that the following items are present in the figure legend, table legend, main text, or Methods section.

| n/a | Confirmed | |
|---|---|---|
| ☐ | ☒ | The exact sample size (*n*) for each experimental group/condition, given as a discrete number and unit of measurement |
| ☐ | ☒ | A statement on whether measurements were taken from distinct samples or whether the same sample was measured repeatedly |
| ☐ | ☒ | The statistical test(s) used AND whether they are one- or two-sided<br>*Only common tests should be described solely by name; describe more complex techniques in the Methods section.* |
| ☐ | ☒ | A description of all covariates tested |
| ☐ | ☒ | A description of any assumptions or corrections, such as tests of normality and adjustment for multiple comparisons |
| ☐ | ☒ | A full description of the statistical parameters including central tendency (e.g. means) or other basic estimates (e.g. regression coefficient) AND variation (e.g. standard deviation) or associated estimates of uncertainty (e.g. confidence intervals) |
| ☐ | ☒ | For null hypothesis testing, the test statistic (e.g. *F*, *t*, *r*) with confidence intervals, effect sizes, degrees of freedom and *P* value noted<br>*Give P values as exact values whenever suitable.* |
| ☒ | ☐ | For Bayesian analysis, information on the choice of priors and Markov chain Monte Carlo settings |
| ☒ | ☐ | For hierarchical and complex designs, identification of the appropriate level for tests and full reporting of outcomes |
| ☒ | ☐ | Estimates of effect sizes (e.g. Cohen's *d*, Pearson's *r*), indicating how they were calculated |

*Our web collection on statistics for biologists contains articles on many of the points above.*

## Software and code

Policy information about availability of computer code

| Data collection | REDCap (Research Electronic Data Capture) and Microsoft Excel (version 16.77) |
|---|---|
| Data analysis | Trimmomatic (version 0.32), Guppy (versions 6.0.1 and 6.3.8), Unicycler (versions 0.4.7 and 0.4.8), SPAdes (Galaxy Version 3.15.4+), Prokka (version Galaxy 1.14.6), PHASTER (https://phaster.ca/), Prophage Hunter (https://pro-hunter.bgi.com/), Biolog Data Analysis software (version 1.7), Porechop (version 0.2.4), Snippy (version 4.6.0), EggNOG-mapper (version 2.1.8), mobileOG-db (version 1.1.2), Phigaro (version 2.3.0), circos (version 0.69.8), and GC-profile, Roary (version 3.13.0), fasttree (version 2.1.10), iTOL (itol.embl.de), SAS (version 9.4), R (version 4.3.0), tidyverse (version 2.0.0), UpSetR (version 1.4.0), ggmap (version 3.0.2), rnaturalearth (R package version 0.3.2.9000), and GraphPad V (version 0.5.1), the PHROGS v3 database (https://phrogs.lmge.uca.fr/)ARG-ANNOT (ARG-ANNOT NT V6 July 2019), CARD (Versions 3.1.4 to 3.2.5), ResFinder (https://bitbucket.org/genomicepidemiology/resfinder_db), blastn web interface (https://blast.ncbi.nlm.nih.gov/Blast.cgi), VFDB full (downloaded on 20 April 2022), Bowtie2 (Galaxy Version 2.5.0) |

For manuscripts utilizing custom algorithms or software that are central to the research but not yet described in published literature, software must be made available to editors and reviewers. We strongly encourage code deposition in a community repository (e.g. GitHub). See the Nature Portfolio guidelines for submitting code & software for further information.

# Data

Policy information about availability of data

All manuscripts must include a data availability statement. This statement should provide the following information, where applicable:

- Accession codes, unique identifiers, or web links for publicly available datasets
- A description of any restrictions on data availability
- For clinical datasets or third party data, please ensure that the statement adheres to our policy

Detailed clinical protocols, results, and additional data are available in the manuscript and in Supplementary Tables 1 and 2. The protocol for the retrospective, observational study is available at: https://clinicaltrials.gov/ct2/show/NCT05498363?term=NCT05498363&draw=2&rank=l. The bacteriophage genome sequences can be retrieved in the GenBank database under the accession codes listed in Extended Data Table 1. The genome data of the bacterial isolates can be accessed via the NCBI BioProject PRJNA975428. PHROGS v3 database available at https://phrogs.lmge.uca.fr/. The authors declare that all other data supporting the findings of this study are available within the article.

# Research involving human participants, their data, or biological material

Policy information about studies with human participants or human data. See also policy information about sex, gender (identity/presentation), and sexual orientation and race, ethnicity and racism.

| | |
|---|---|
| Reporting on sex and gender | The median age of the patients was 53 years (1–91 years), and 56.7% of the patients were male. No effects of age or gender on eradication were found using a logistic regression analysis. |
| Reporting on race, ethnicity, or other socially relevant groupings | Race, ethnicity, or other socially relevant groupings were not considered or reported. |
| Population characteristics | The median age of the patients was 53 years (1–91 years), and 56.7% of the patients were male. No effects of age or gender on eradication were found using a logistic regression analysis. |
| Recruitment | This study concerns the first 100 consecutive bacteriophage therapy (BT) cases facilitated by a Belgian consortium (single group, open label). Physicians requesting BT with bacteriophage preparations for their patients submitted a BT request to the Phage Therapy Coordination Centre (PTCC) of the QAMH. The PTCC procedure for selecting patients for BT is depicted in Extended Data Figure 1 and is largely determined by clinical need, regulatory approval, and the availability of bacteriophages targeting the infecting bacteria. |
| Ethics oversight | According to EU Regulation No 536/2014 (Clinical Trials Regulation), its transposition to Belgian Law, and per advice of the Leading Ethical Committee of the "Universitair Ziekenhuis Antwerpen" and the "Universiteit Antwerpen" (ID 3644), which approved the observational study protocol. The present retrospective non-interventional analysis of an existing and de-identified BT database was not considered as an experiment on the human person and did not require a dedicated informed consent. This information is also provided in the manuscript. |

Note that full information on the approval of the study protocol must also be provided in the manuscript.

# Field-specific reporting

Please select the one below that is the best fit for your research. If you are not sure, read the appropriate sections before making your selection.

☒ Life sciences ☐ Behavioural & social sciences ☐ Ecological, evolutionary & environmental sciences

For a reference copy of the document with all sections, see nature.com/documents/nr-reporting-summary-flat.pdf

# Life sciences study design

All studies must disclose on these points even when the disclosure is negative.

| | |
|---|---|
| Sample size | 100 patients (retrospective report no sample predetermined). For *Galleria mellonella* sample size was based on previous publication Nieuwenhuyse et al., 2022) |
| Data exclusions | For Galleria model one experiment was excluded from the analysis as different time points and measurements were taken. |
| Replication | Bacteriophage - antibiotic synergy testing was performed only once, in an emergency routine setting, prior to the application of bacteriophages. For Galleria model this experiment was repeated once. All other experiments were repeated at least 3 times. |
| Randomization | Patients were not randomised. The study has a single group design. For *G. mellonella,* larvae were randomly assigned. |
| Blinding | There was no blinding. The study has an open label design. For *G. mellonella* the assessment was not blinded. |

# Reporting for specific materials, systems and methods

We require information from authors about some types of materials, experimental systems and methods used in many studies. Here, indicate whether each material, system or method listed is relevant to your study. If you are not sure if a list item applies to your research, read the appropriate section before selecting a response.

## Materials & experimental systems

| n/a | Involved in the study |
|-----|----------------------|
| ☒ | ☐ Antibodies |
| ☒ | ☐ Eukaryotic cell lines |
| ☒ | ☐ Palaeontology and archaeology |
| ☐ | ☒ Animals and other organisms |
| ☐ | ☒ Clinical data |
| ☒ | ☐ Dual use research of concern |
| ☒ | ☐ Plants |

## Methods

| n/a | Involved in the study |
|-----|----------------------|
| ☒ | ☐ ChIP-seq |
| ☒ | ☐ Flow cytometry |
| ☒ | ☐ MRI-based neuroimaging |

## Animals and other research organisms

Policy information about studies involving animals; ARRIVE guidelines recommended for reporting animal research, and Sex and Gender in Research

| | |
|---|---|
| Laboratory animals | Species is Galleria mellonella; strain is unspecified; sex indifferent; stage larvae; age about 3 weeks |
| Wild animals | No wild animals used in this study |
| Reporting on sex | Sex was not considered |
| Field-collected samples | The study did not involve samples collected from the field |
| Ethics oversight | It was considered that use of this invertebrate model did not require specific guidance. |

Note that full information on the approval of the study protocol must also be provided in the manuscript.

## Clinical data

Policy information about clinical studies

All manuscripts should comply with the ICMJE guidelines for publication of clinical research and a completed CONSORT checklist must be included with all submissions.

| | |
|---|---|
| Clinical trial registration | Study BT100, ID: NCT05498363 |
| Study protocol | The protocol for the retrospective, observational study is available at: https://clinicaltrials.gov/ct2/show/NCT0S498363?term=NCT0S498363&draw=2&rank=l. |
| Data collection | Prior to BT, demographic and clinical data were collected through a medical form, which was completed by the Bacteriophage Therapy Providers. The medical doctor's BT prescription, information regarding the applied bacteriophage product and its administration route, dosage, duration, and information with regard to possible concomitant (antibiotic) treatments were also recorded. The "phagograms", reporting on the evaluation of the bacteriophage susceptibility of the patient's bacterial isolates sampled before and sometimes during treatment were also archived. If the bacteriophage treatment was performed in a hospital, a clinical follow-up form, requesting information about the clinical outcome (incl. possible adverse events and reactions), was completed by the treating physician and the nursing team and sent to the PTCC. In case of ambulatory BT, clinical follow-up information was collected directly from the patients. All demographic, bacteriophage product, and clinical data were recorded in a REDCap (Research Electronic Data Capture) designed database. Details on settings, places where the data were collected and periods when collected and data collection are included in the manuscript. |
| Outcomes | Clinical improvement, eradication of the targeted bacterium, and the advent, severity and duration of adverse events or reactions were assessed by the treating physician. |

