## [Peer Review File · Nature Microbiology]

Peer Review Information

Journal: Nature Microbiology

Manuscript Title: Personalized bacteriophage therapy outcomes for 100 consecutive cases: a multi-centre, multi-national, retrospective, observational study

Corresponding author name(s): Dr Jean-Paul Pirnay

Reviewer Comments & Decisions:Decision Letter, initial version:**Message** 17th November 2023

Dear Jean-Paul,

Thank you for your patience while your manuscript "Retrospective, observational analysis of the first one hundred cases of personalized bacteriophage therapy facilitated by a Belgian consortium" was under peer-review at Nature Microbiology. It has now been seen by 4 referees, whose expertise and comments you will find at the end of this email. Although they find your work of some potential interest, they have raised a number of concerns that will need to be addressed before we can consider publication of the work in Nature Microbiology.

Should further experimental data allow you to address these criticisms, we would be happy to look at a revised manuscript.

Please include a data availability statement as a separate section after Methods but before references, under the heading "Data Availability". This section should inform readers about the availability of the data used to support the conclusions of your study. This information includes accession codes to public repositories (data banks for protein, DNA or RNA sequences, microarray, proteomics data etc...), references to source data published alongside the paper, unique identifiers such as URLs to data repository entries, or data set DOIs, and any other statement about data availability. At a minimum, you should include the following statement: "The data that support the findings of this study are available from the corresponding author upon request", mentioning any restrictions on availability. If DOIs are provided, we also strongly encourage including these in the Reference list (authors, title, publisher (repository name), identifier, year). For more guidance on how to write this section please see: <http://www.nature.com/authors/policies/data/data-availability-statements-data-citations.pdf>

* Include a "Response to referees" document detailing, point-by-point, how you addressed

3each referee comment. If no action was taken to address a point, you must provide a compelling argument. This response will be sent back to the referees along with the revised manuscript.

* If you have not done so already we suggest that you begin to revise your manuscript so that it conforms to our Article format instructions at <http://www.nature.com/nmicrobiol/info/final-submission>. Refer also to any guidelines provided in this letter.

When submitting the revised version of your manuscript, please pay close attention to our [href="https://www.nature.com/nature-portfolio/editorial-policies/image-integrity">Digital Image Integrity Guidelines](https://www.nature.com/nature-portfolio/editorial-policies/image-integrity). and to the following points below:

Note: This url links to your confidential homepage and associated information about manuscripts you may have submitted or be reviewing for us. If you wish to forward this e-mail to co-authors, please delete this link to your homepage first.

Nature Microbiology is committed to improving transparency in authorship. As part of our efforts in this direction, we are now requesting that all authors identified as 'corresponding author' on published papers create and link their Open Researcher and Contributor Identifier (ORCID) with their account on the Manuscript Tracking System (MTS), prior to acceptance. This applies to primary research papers only. ORCID helps the scientific community achieve unambiguous attribution of all scholarly contributions. You can create and link your ORCID from the home page of the MTS by clicking on 'Modify my Springer Nature account'. For more information please visit please visit www.springernature.com/orcid.

If you wish to submit a suitably revised manuscript we would hope to receive it within 6 months. If you cannot send it within this time, please let us know. We will be happy to consider your revision, even if a similar study has been accepted for publication at Nature Microbiology or published elsewhere (up to a maximum of 6 months).

Yours sincerely,

Reviewer Expertise:

Referee #1: Clinical phage therapy
 Referee #2: Clinical phage therapy
 Referee #3: Bacteriophage biology
 Referee #4: Biostatistics

Reviewer Comments:

Reviewer #1 (Remarks to the Author):

Pirnay et al describe 100 consecutive cases of bacteriophage therapy. The need for descriptions of real-world applications comes from the burgeoning interest in therapeutic application of bacteriophages to 'rescue' antibiotic failure. This is the largest recent 'real-world' case series in which 'compassionate use' indications, necessarily diverse and heterogeneous in nature, are catalogued in detail.

The authors are to be congratulated on their leadership in this area, their careful description of the information they have to hand and their thoughtful analysis of the data. What the manuscript does well is provide a simple and easy to read catalogue of the use of phages administered for (mostly) *Pseudomonas aeruginosa* and *Staphylococcus aureus* infection, in cases of (mostly) respiratory and orthopaedic infections.

It also provides an unadulterated record of all the available data from a leading phage therapy site that is trying to advance the systematic use of phage therapy, applying as much standardisation of data collection as feasible.

Standardisation and reporting

Dosing schedules are semi-standardised, with IV dosing at 10^7 - 10^9 pfu once daily for 5-10 days, and longer schedules using slightly more given topically (or into a lesion) daily and 3 or 4 times daily, sometimes for weeks, into airways by spray (nasal) or nebuliser. All preparations appear to be 'clean' and phages were shown to be potent in vitro before usage.

Several important clinical application issues are addressed but the data are difficult to evaluate as they could not be prospectively or systematically collected. The authors are very clear about the nature of the data so this is simply an acknowledgment of the 'state of play' in phage therapy by the team.

Outcomes

Clinical improvement or otherwise was subjectively defined by requesting clinicians (e.g. see Methods from lines 1022-37). There are no clinical metrics applied (e.g. illness severity scores in various systems) nor can we determine the influence of other (medical/surgical) interventions. The entry criteria are not clear – that is, there is no 'gateway' into the system at which data are gathered (although an advisory panel of physicians sets the general prescribing protocols in conjunction with the requestor/s) – but it is generally implied that these are all cases in which therapy was already optimal but inadequate. Microbiological eradication is presented as a record of the normal clinical process of testing (in some) for the persistence of the putative pathogen and therefore is limited as an outcome marker as it is not prospectively or systematically evaluated.

Safety

Safety data terminology/ reporting is not standardised (e.g. as per the EudraVigilance system), perhaps simply because the data are not prospectively collected nor the

5descriptors defined in advance to clinicians. These data cannot be used as a safety report but safety is not regarded as a particular risk area and the data are consistent with the general signals around safety which are already reported.

Neutralising antibody development

Neutralising antibody development is an important issue in regard to therapeutic applications. These were assessed by mixing serum (apparently not complement depleted) with phages at $\sim 10^7$ pfu/mL (10% serum final v/v) for a dichotomous result (rather than a semi-quantitative Plaque Reduction Neutralisation Test). Of great interest is the apparent variability of immunogenicity within phage cocktails, with some phages apparently more immunogenic than others in the same individual. Unfortunately, there are no data on whether there was amplification of some phages (so-called 'autodosing') that may have influenced exposure and therefore immune response. It therefore remains unclear whether phage therapy inevitably leads to neutralising antibody development, and the route of administration does not seem to be a determinant here. It seems also that only 13 of the 100 cases were tested and neutralising antibodies were detected in 5. The authors state that it did not prevent clinical improvement and bacterial eradication in four of these but we do not know even whether the phages amplified in host bacterial populations (i.e. hit their target). The inference in the Discussion (e.g. line 540) therefore really does need a caveat along these lines.

Resistance evolution

A few instances of resistance evolution in response to phage pressure are described in the 21 cases which were able to be evaluated. 7 cases experienced emergence of phage resistant bacteria, and most of these were in settings in which there was high burden of disease (respiratory tract or immunocompromised infection) or in which a phage may have been directed against a target that was not the principal pathogen. Resistance evolution was phage receptor-specific or through emergence of an alternative to the primary target pathogen. It is not clear whether those cases in which no resistance emerged simply had no post-treatment samples available or whether they were true cures and there was no clear correlation with microbiological eradication or failure thereof. A statistical model was applied to predict this but is hampered by the integrity of the data making up the variables. It is also not clear whether the more problematic or refractory cases were those in which evaluation was more extensive and therefore whether it is appropriate to generalise a percentage (the reader might assume that $\sim 40\%$ of all phage therapy leads to R evolution in vivo – that may be so but these data do not show that). It would be sensible to add a small caveat in that regard into the discussion. If the authors can demonstrate that these 21 cases are representative of the 100 and therefore these data generalisable, that would be valuable.

Q: was growth kinetics performed on every isolate-phage comparison or were they all subjected to EOPs but only a few to growth kinetics? It seems as though many of the later ones went through the Omnilog system – is that the subset that was reported and, if so, could this be more clearly reiterated?

In addition, a few specific suggestions and queries (Q), by line:

Line 83: ...e.g. by 'compounding pharmacies' in countries like the US.

Line 86: delete brackets; add 'for' before '...urgent cases'

Line 121: 'a total of'

Line 150: Q - 8×10^9 , SD 1×10^{10} – please confirm

Line 177: comparisons

Line 194: Q – define 'moderate' activity?

Line 227: delete 'at'

Line 256: Q - also on quinolones? (pt 91?)

Line 260: Q - resensitised or generally unfit? Did you do growth curves?

Line 283: omnilog growth curves fig 3/ table 2 showed synergy in 9/10: q – please define how synergy was defined? Duration of suppression? Relative final OD? etc

Line 296: 5/13 developed immune neutralisation and some were partial; Q – was this predicted by phage exposure (do we know or can we calculate intensity of phage viraemia after administration?)

Line 388: Q – confirm 24-28 hour suppression of growth as this seems to vary in the figures displayed. Did you mean plaque persistence as clear?

Line 481: Q – or simply continue the antibiotic/ surgical Rx etc?

Line 488: It might be avoid the overgeneralisation – where there is a simple trade-off (e.g. selecting against a pump or porin) that is well tolerated in optimal growth media then the ab susceptibility in vitro evidently changes – it is not a general property of phages but rather a happy coincidence that can be predicted to some extent by the receptor. It might be good to be more clear about this in lines 489-90.

Line 540: implies successful phage therapy – can the authors provide evidence that the phage at least amplified in the host bacteria and therefore might have contributed to outcome or do the authors think this needs a caveat e.g. since this is the Discussion section, it might be worth pointing this out here.

Line 557: did the authors demonstrate the advantages of combining BT with standard of care? They certainly showed some in vitro evidence of synergy and can reasonably extrapolate this to increased antimicrobial effectiveness – perhaps rephrase this? Its probably important to keep this caveat to a very simple statement however in order to avoid difficult arguments about antibiotic effects which cannot be addressed from these data – e.g. if aminoglycoside efficacy is mostly related to the post antibiotic effect while beta-lactam killing is mostly related to maintaining time above MIC, how do these relate to phage effects?

Line 904: Q - so 48h suppression only required for cocktails?

Line 906: Q – greater than or equal to 1?

Line 936: Are there guidelines in regard to dosing etc in the instructions (refs 22-24) and are they based on published data in regard to kinetics? It would be good to make clear to the reader whether this is “level IX” (expert opinion) or higher quality evidence and, if that is the case, to provide references to those data.

Line 976: Q - “ab MIC” ? MOIs less than or equal to 1 or MIC 1.0? Does sub-MIC concentrations means 0.25 x MIC or something else (was this standardised?)

Line 989: Q – please confirm: 0.9 mL of 1: 100 diluted sera is ~10% serum v/v, without complement depletion

Table 1: Q - dosing heterogeneity (MNOI and as intervals) is striking –q 6 to q24 h – can we confirm this to be arbitrary clinician choice and identify as such – are there any kinetics data to inform this? Were these doses based on any published data eg Schooley et al 2017 or was this about total dose - are these doses discretionary and are there any concerns about consistency within and/or between protocols?? See Q above re Russian MoH protocols.

Q - pts 30,54,64,82 and 91 – are these the only R? or are the others simply not known as no specimens were available or because it did not occur? is the group that was singled out for testing representative of the whole 100 (i.e. can we generalise these data at all?) In case 66 – no benefit – were these M abscessus uniformly susceptible? (e.g the common occurrence of ‘rough’ and ‘smooth’ types in the population is often difficult to recognise and may relate to therapeutic failure, esp if unrecognised early on). M abscessus is a common and important indication for phage therapy and this population variation is common so therefore may be worthy of a comment if you have data.

Q - cases 22,23,24,42 – is it possible to identify the probable causative pathogens? E.g. Staph aureus in this context is rarely an irrelevant coloniser etc – complex mix of pathogens in orthopaedic cases – all got better but is phage therapy always targeting the main pathogen?

Nb. cocktails are al broad spectrum (multi organisms) rather than multi target so no phage synergies in these – can we identify those in which true synergies were sought and

proven in vitro and if so, did they have better outcomes?

Q - of the cases in which resistance arose, these were dominated by Pseud infections and the resp tract. Is this because respiratory samples were easiest to obtain or are you inferring that this is a scenario in which R is common?

Nb. Ext Table 4 – is it possible to use standard EudraVigilance terminology and definitions here?

Fig 1a – could be deleted

Fig 1b – regulatory pathway is EU-specific and could also be deleted and simply mentioned in text

Fig 1c – ‘primary’ or ‘secondary’ infection – please define

Fig 1d – nice diagram although also inessential

Summary

This is an important contribution from the field leaders because it describes 100 phage therapy cases and is the largest set described in recent times. The authors are clear about the limitations of the data, and I have raised a few extra questions that may be able to be addressed. The authors are pointing us in the direction we need to take.

The data are difficult to make much from in regard to outcomes, immunity, kinetics or even safety. An interesting variation in the development of neutralising antibodies is described on but we can only speculate on the cause and implications. Whether the ‘negative’ reports in this case series are really phage therapy failures is as uncertain as whether the ‘positive’ outcomes are really phage therapy successes. The authors are completely transparent about this but nevertheless put forward one of the largest case series ever in phage therapy and should be congratulated for making such a thorough analysis of a difficult data set.

With regard to the authors closing remarks, while there are multiples sites doing this now, this collaboration is the international lead and have shown that it is indeed feasible to set up a flexible and safe personalised BT system.

It is not clear that 77% of these cases were successful because of the (phage) intervention as other important contributing (confounding) variables are not controlled in retrospective series. However, we can accept that the referral was made because improvement was not occurring nor thought likely. It is therefore accurate to say that treating physicians regarded BT to have been associated with clinical improvement in 77% and I think that is how it is best expressed.

In vivo development of resistance was evaluated in only a small subset in vivo and it is not clear that this can be generalised, nor is it clear how to interpret the interesting variation in neutralising ab development because the data needed to do so are simply not available.

Finally, future RCT design needs clarity around both dosing (that is, how much phage and how often) and the expected effect or benefit size (to understand the statistical power). The bioavailability/ dosing question is not addressed here.

Nevertheless, these data do indicate that the effect size may be quite significant and that doses of 109pfu or even less for a couple of weeks or less may suffice.

Reviewer #2 (Remarks to the Author):

General comments

This manuscript provides an immense amount of data that will be of interest to those engaged in clinical and translational research related to bacteriophage therapeutics. As the authors note, the main shortcoming of the analysis is that there was not a uniform,

prospectively specified definition of clinical treatment success. This is one of the same challenges facing those wishing to interpret similar retrospective case series published from Poland, Georgia and other former Soviet bloc countries.

Having said this, the manuscript meticulous in detail and vast in scope. Unlike many of the aforementioned retrospective reports from the Eastern bloc, this analysis presents data that more directly touch on important issues facing contemporary phage therapeutics: How often do patients raise an antibody response to phages that are administered therapeutically? Does this affect treatment outcomes? Does this occur more often when phages are administered parenterally as opposed to locally? How often do bacteria with reduced phage susceptibility arise during phage therapy? Are these bacterial variants as "fit" or invasive as pre-treatment strains? If possible, should phages be administered with antibiotics that are active against the bacteria being targeted by bacteriophage therapy? The manuscript does not provide definitive answers to these questions but in many cases it presents data of sufficient quality and rigor to further shape the research agenda.

Providing data about 100 patients in chronological order is highly meritorious and helps better define success rates since this approach begins to address issues of positive selection bias from the case report literature. As best one can tell, however, the authors are presenting what one might call an "as treated" analysis. Two critical pieces of information that are missing from the manuscript are the denominator from which these 100 "as treated" patients were drawn and how often therapy was recommended but no phage could be identified and produced in time to treat the patient (e.g., had this been a modified intent to treat analysis what would the success rates have been?). Can the authors provide data about how many patients were proposed to the Bacteriophage Therapy Coordinating Center in the time period covered by this report (e.g., between January 1, 2008 and April 30, 2022)? Likewise, during this period, how many patients were recommended for phage therapy but not treated because phages could not be identified?

One of the issues that is a bit unclear about the triage process is whether the 100 patients who were ultimately selected for therapy were ones for whom an active phage was already in hand at the time of the request (as opposed to situations in which screening of environmental phages was required). If candidates were not recommended for treatment or did not receive treatment, what were the reasons for this? Did they have inappropriate clinical conditions? Did they have bacterial infections for which lytic phages were not available in a clinically relevant time frame? Etc. Please clarify.

What was the mean time from a treatment request to the initiation of therapy? If you did need to perform environmental searches and prepare phages for administration from scratch, how long did this take on average?

More information about the specific objective criteria used by the Bacteriophage Therapy Coordinating Center to recommend or discourage phage use would benefit others trying to adapt this experience to their own circumstances.

Comments in the manuscript about specific regulatory aspects of phage therapeutics are interesting and important in the European context but they might not be generalizable to other settings. The authors are correct on line 401 about the challenges posed by strict GMP manufacturing standards for individualized therapeutic application of phages. While it is true that the "magistral" phage production approach has been quite workable in the hands of the investigators, the approach described in the section beginning on line 790 is

not strictly identical to the US compounding laboratory procedures to which they compare magistral production in Europe. The authors are to be commended for the extent to which they describe and adhere to manufacturing and quality control aspects of their work in the sections beginning on lines 790 and 826. The use of a specific organization (Sciensano) to monitor quality and safety would not be required in the compounding pharmacy “industry” in the US. This has led to a number of major fiascos such as the introduction of aspergillus into methylprednisolone that was injected into the CSF by unscrupulous neurosurgeons in Nashville, TN. (1) Individual patient phage in the US use analogous to that described by the authors is governed by the US Food and Drug Administration’s individual patient IND process. In this situation the FDA provides guidance about the level of production quality required for individual patients on the basis of a dialog with the physician about the urgency of treatment and the practicality of additional phage characterization and purification steps. The process works well and assures a level of oversight that has avoided mishaps that might occur if clinical phage preparation devolved to individual laboratories that were “overseen” by so-called expert committees at the hospitals in which the phages were to be administered. The QAMH experience is a bit different from an unregulated compounding approach in that a highly experienced group of phage researchers have prepared 40 batches of phages using rigorous pre-specified procedures that were QC’ed by a single experienced organization. Thus, while it is important (and indeed necessary) to describe the approach taken by QAMH, it should be stated that other oversight and regulatory approaches may be preferable in other settings.

1. Chiller TM, Roy M, Nguyen D, Guh A, Malani AN, Latham R, Peglow S, Kerkering T, Kaufman D, McFadden J, Collins J, Kainer M, Duwve J, Trump D, Blackmore C, Tan C, Cleveland AA, MacCannell T, Muehlenbachs A, Zaki SR, Brandt ME, Jernigan JA; Multistate Fungal Infection Clinical Investigation Team. Clinical findings for fungal infections caused by methylprednisolone injections. *N Engl J Med.* 2013 Oct 24;369(17):1610-9. doi: 10.1056/NEJMoa1304879. PMID: 24152260.

Minor comments

Line

45 Small point but I’d suggest inserting the word “perceived” since commercial sustainability is in the eyes of the beholder. Big Pharma is not interested in drugs that are merely “commercially sustainable” but rather blockbusters that yield blockbuster returns. Thus, I hate to let them off the hook by saying that antibiotics are not commercially sustainable and would suggest qualifying this comment.

46 The AMR Action fund hopes to bring 2 – 4 antibiotics to the market by 2030. The way the sentence is worded (“is expected”) is imprecise. Those donating the money may expect for this fund to do just this but it is not a foregone conclusion that this would happen. It would be more accurate to say that the fund was created with the hope that 2 – 4 new antibiotics could be brought to market.

171 Can the authors identify experimental data that define an EOP>0.1 as being “therapeutically acceptable”?

337 and Figure 1c Recognizing all of the complexities of retrospectively analyzing a case series in which patients were selected for bacteriophage therapy based on the perception of a multidisciplinary team that phages might be of benefit clinically and in which clinical outcomes were determined by the treating physician without a formal definition, can the authors provide treatment “success” rates for infections of the various types depicted in Figure 1c? Such data might help others make more informed decisions about selecting

candidates for phage therapy.

Reviewer #3 (Remarks to the Author):

The manuscript by Pirnay et al., describes a retrospective study of the first one hundred personalised phage therapy cases treated by the Belgium consortium, which included groups across Europe, and targeted a broad range of difficult-to-treat infections with antimicrobial-resistance determinants. This is an incredibly important study and I want to upfront thank the authors for their work, progress, and impact they have made on the field. I especially enjoyed the juxtaposition between phage therapy approaches, mainly the disappointing results that have been seen from RCT using pre-defined phage cocktails compared with the personalised, phage-for-service model proposed here.

With regards to the manuscript, I believe this will be a seminal paper in the field that will likely define much of what the field prescribes for personalised phage therapy moving forward. It is for this reason that I have been especially picky and nuanced in my comments provided here. I have no major concerns with regards to the results, statistics and data presented in the paper, and much of what is presented should be considered state-of-the-art with respect to personalised phage therapy. However, in my opinion, the authors have not done a good job of presenting their data in the most impactful way, nor have they effectively communicated the work in an accessible way, particularly for a broader readership in Nature Microbiology. As such, much of my comments and suggestions will be focused on these two criticisms.

The impact of this paper, in my opinion, is on the retrospective treatment of 100 patients with personalised phage therapy. I encourage the authors to keep emphasis this front and centre when revising the manuscript in order to maintain focus, while significantly shortening their writing. I have two major summary points for the authors to address. The first is revising the overall structure of the manuscript, which is not traditional, which is fine, but as a result there are many sections and paragraphs that are not well placed, breaking the flow and narrative of the paper. Secondly, the paper is unnecessarily long and at times unfocused. An example of this is found in the discussion, which is nearly five solid pages of text, it has little organisation in its writing, and has many sections that are better suited to a literature review rather than being included within a retrospective analysis that is presented here. In brief, the authors are doing too much, and in my opinion, this has negatively impacted the paper and its message.

Please see below line-by-line comments and suggestions:

Lines 92-96: 'Quality & safety were verified by Sciensano' - I think it would be of benefit to the field (and future regulators) to briefly state how these preparations were verified. i.e., sterility testing? Endotoxin? Sequencing? In vitro screens?

Lines 101-102: As noted in the abstract, not all cases employed these assays - could the authors include a short note on this point?

Lines 101-108: Is this best suited for the introduction? This writing reads more like a method and I personally think it breaks the flow between impact of Eliava and the summary of this article.

Line 151: Can you provide further details on the methodology used for your bioburden assays? Were these USP71 sterility tests? If so, how much volume or fraction of the batch

11was tested? Or was this a plating and CFU count performed by Sciansano? This is important to define as a 0 CFU count can be a false positive depending on your volume tested compared to batch production. This detail also needs to be clearly defined in the methods, but a brief summary statement in main text is also justified due to the importance of these end-point metrics. I also note the associated methods section was very limited on these important points (lines 852-854). Readers should not have to go to other documentation/papers to find these key criteria.

Line 152: Please clarify - were the median EU measurements of your phages 5 EU/mL of product? Or median of 5 EU/mL per patient kg? I believe it was the former, but this should be very clear in the writing. Again, associated methods were poorly described.

Lines 154-157: Do these requirements hold for the Eliava phage cocktails? I note these were administered via topical, nebulised, rectal, and intralesional. From my understanding these products are not produced within safe/FDA limits for endotoxins

Lines 159-160: Same comment as above for Eliava phage cocktails. From my understanding these cocktails are not genetically defined. In your cases using these products, were these sequenced? If so, how did you confirm <5% bacterial reads if host genomes were not available/known?

Lines 176-199: In my opinion, this section is an example of where the authors have tried to include too much detail and depth in the manuscript, and I think it detracts from its focus.

You state that a detailed genetic comparison of adapted phages falls out of scope of this manuscript, and I agree. But you then go on and provide a detailed genomic comparison of two published adapted phages. The writing should be tightened in aspects of the manuscript, I would suggest removing this content and keeping your focus on the therapy cases. You cannot report everything in this paper, so I would suggest focusing on the clinical impact.

Figure 1: This is a suggestion, but I feel like Figure 1 is missing an opportunity to present a simple yet impactful visualisation of the success and eradication rates of the 100 treated patients - this is likely the impactful point most readers and citations will be looking for - yet it is not graphically represented.

Lines 231-232: How was colicin I identified as a phage receptor? In this study or a previous one? If no knock out work was completed to validate this receptor then this should be listed as a putative receptor. Same comment with T4P in subsequent lines, either provide citation for mechanistic work, or use putative.

Lines 238-243: When discussing patients 54, 30, and 91 please be clear that these patients were all infected with P.a.

Lines 264-276: Please add a concluding statement for the Galleria model section and define the initial goals of this assay. I would also ask the authors to consider its relevance within this article. The impact here is patients treated, and I'm not certain the animal work fits well, and in some ways, it detracts from the study.

Figure 3: I'd encourage authors to revise their data presentation in Figure 3 - this looks more like a supplemental figure. There is a lot of data here and it is not obvious the point you are trying to make with this figure. Further the axis fonts are almost unreadable, there are no error bars presented, and I doubt non-phage experts would be able to easily

decipher this figure. Why not represent this as AUC? Or another condensed metric to emphasize the phage vs phage+Abx comparison? Then include these curves as supplemental figures.

Lines 282-287: It is worth to briefly mention how synergy and additive effects were calculated/distinguished.

Lines 278-293: It was unclear from this section, was phage-antibiotic synergy only tested in vitro? Did these results impact or influence the patient treatment? Were these synergistic/additive combinations used clinically? How did this data inform the clinical treatment of the 100 patients?

Figure 4: This figure can be improved by overlaying additional treatment information, such as the duration of phage therapy, clinical improvement and bacterial eradication (if these occurred), and potentially co-administration of antibiotic treatment.

Lines 318-321: Please rephrase as it currently reads as if five patients died of septic shock, when this was only two.

Discussion: The discussion was close to five pages of solid text. It at times, lost focus and read more like a literature review or thesis, rather than a critical discussion on the 100-cases treated. Further, the structure of the discussion seemed erratic, jumping from topic to topic without much flow or coherency.

After finishing this section, I found I had missed/forgotten the impact of this work, which in my opinion is the treatment of 100 BT cases, and was more distracted by the writing, historical musings, and summary of prior studies.

I strongly encourage the authors to review this discussion with the goal to focus their writing on critical points, to reduce the text by over half, and to maintain a logical flow in the discussion that summarises the major findings across their study.

Line 356: Please remove the dashes here, it breaks sentence flow

Lines 386-388: This definition is an idyllic goal. More specifically, I note that most (arguably all) of your biologic growth curves (Fig3) show the emergence of phage resistance at low MOIs within as short as seven hours and stable lysis for 24-48 hours was not evident. This is to be expected for in vitro growth curve assays. Yet by your own definitions, these phages do not constitute an efficient therapeutic phage, yet they were used with promising success rates. So how do you reconcile these discrepancies and adhere to your own definition?

Lines 418-430: This section reads more like a literature review than a discussion. What approach for dosing did you take in this study? As far as I could tell, dosing information from your own study was completely absent here. Keep the description of the prior work focus on your study, particularly within the discussion.

Lines 438-443: Recommend this text be removed.

Lines 451-464: Again, a literature review style of writing that inflates your writing and retracts for this papers message. Recommend this text be removed.

511-527: Recommend this text be removed.

Reviewer #4 (Remarks to the Author):

Remarks to manuscript from the viewpoint of statistics

=====

The authors correctly claim in lines 345f that “the majority of patients were unique cases” and that “no inferential statistics could be applied”. In this sense, the reporting of figures like odds ratios can be seen as hints to possible connections without fully well-grounded statistical evidence. However, reporting confidence intervals (CIs) is part of inferential statistics. I do not say that CIs should not be mentioned. But it should be noted that they must be interpreted with caution. Fisher’s exact test (line 322) is also part of inferential statistics.

From line 119 on can conclude that some patients suffered from coinfections. This should be explicitly mentioned. It seems that a coinfection was treated as two independent infections concerning the bacteriophage therapy. My medical is not sufficient to judge whether this is justified.

In line 309 it is mentioned that “in eight patients, no adequate post BT bacteriological data was available”. What is the reason for the missing data? If an adverse event like death was the reason, this should be taken into account in the analysis.

Maybe this question is naïve. But is the antibiotic resistance profile (line 323) a dichotomous variable? Otherwise it is not clear to me how Fisher’s exact test was applied.

In lines 324f it is mentioned that “No effects of age or gender on eradication were found using a logistic regression analysis.” This was univariate logistic regression considering solely age and gender, respectively?

Does “stepwise logistic regression analysis” in line 325 refer to forward selection (or backward elimination)?

Logistic regression can easily show overfitting effects when it is tested on the same data with which the variables were selected and the coefficients were determined. The confusion matrix mentioned in line 329 should be computed based on leave-one-out cross-validation and not based on the training data.

Line 330 states that the logistic regression model was “right in 65% ... of the time”. What is the proportion of the majority class? If it is 65% or more, 65% would be very poor.

I do not understand the connection between the confusion matrix mentioned in line 329 and the logistic regression. It seems that the confusion matrix provides the correlation between eradication and AB treatment. Where do the predictions of the logistic regression enter the confusion matrix?

Author Rebuttal to Initial comments

REVIEWER #1

Pirnay et al describe 100 consecutive cases of bacteriophage therapy. The need for descriptions of real-world applications comes from the burgeoning interest in therapeutic application of bacteriophages to 'rescue' antibiotic failure. This is the largest recent 'real-world' case series in which 'compassionate use' indications, necessarily diverse and heterogeneous in nature, are catalogued in detail.

The authors are to be congratulated on their leadership in this area, their careful description of the information they have to hand and their thoughtful analysis of the data. What the manuscript does well is provide a simple and easy to read catalogue of the use of phages administered for (mostly) *Pseudomonas aeruginosa* and *Staphylococcus aureus* infection, in cases of (mostly) respiratory and orthopaedic infections.

It also provides an unadulterated record of all the available data from a leading phage therapy site that is trying to advance the systematic use of phage therapy, applying as much standardisation of data collection as feasible.

RESPONSE TO REVIEWER 1:

Thank you for the constructive comments. Adapting the manuscript accordingly, makes it much clearer and enhances its quality. Please find below the point-by-point response to the questions and points raised.

Standardisation and reporting

Dosing schedules are semi-standardised, with IV dosing at 10^7 - 10^9 pfu once daily for 5-10 days, and longer schedules using slightly more given topically (or into a lesion) daily and 3 or 4 times daily, sometimes for weeks, into airways by spray (nasal) or nebuliser. All preparations appear to be 'clean' and phages were shown to be potent in vitro before usage.

Several important clinical application issues are addressed but the data are difficult to evaluate as they could not be prospectively or systematically collected. The authors are very clear about the nature of the data so this is simply an acknowledgment of the 'state of play' in phage therapy by the team.

Outcomes

Clinical improvement or otherwise was subjectively defined by requesting clinicians (e.g. see Methods from lines 1022-37). ***There are no clinical metrics applied (e.g. illness severity scores in various systems) nor can we determine the influence of other (medical/ surgical) interventions.***

Response: We agree. This is indeed a limitation of this study and is highlighted in the revised manuscript.

The entry criteria are not clear – that is, there is no ‘gateway’ into the system at which data are gathered (although an advisory panel of physicians sets the general prescribing protocols in conjunction with the requestor/s) – but it is generally implied that these are all cases in which therapy was already optimal but inadequate. Microbiological eradication is presented as a record of the normal clinical process of testing (in some) for the persistence of the putative pathogen andtherefore is limited as an outcome marker as it is not prospectively or systematically evaluated.

Response: Indeed, this was the case. This is highlighted in the revised manuscript.

Safety

Safety data terminology/ reporting is not standardised (e.g. as per the EudraVigilance system), perhaps simply because ***the data are not prospectively collected nor the descriptors defined in advance to clinicians***. These data cannot be used as a safety report but safety is not regarded as a particular risk area and the data are consistent with the general signals around safety which are already reported.

Response: We agree. This limitation is highlighted in the revised manuscript.

Neutralising antibody development

Neutralising antibody development is an important issue in regard to therapeutic applications. These were assessed by mixing serum (apparently not complement depleted) with phages at $\sim 10^7$ pfu/mL (10% serum final v/v) for a dichotomous result (rather than a semi-quantitative Plaque Reduction Neutralisation Test). Of great interest is the apparent variability of immunogenicity within phage cocktails, with some phages apparently more immunogenic than others in the same individual. Unfortunately, there are no data on whether there was amplification of some phages (so-called 'autodosing') that may have influenced exposure and therefore immune response. It therefore remains unclear whether phage therapy inevitably leads to neutralising antibody development, and the route of administration does not seem to be a determinant here.

It seems also that only 13 of the 100 cases were tested and neutralising antibodies were detected in 5.

Response: Indeed, according to our limited data set (13 cases for which sufficient, adequate samples were available), phage therapy does not always lead to serum neutralization. In only 5/13 cases, all involving invasive phage therapy, phages were neutralized by the patient's serum. This was emphasized in the revised manuscript.

The authors state that it did not prevent clinical improvement and bacterial eradication in four of these but ***we do not know even whether the phages amplified in host bacterial populations (i.e. hit their target)***. ***The inference in the Discussion (e.g. line 540) therefore really does need a caveat along these lines.***

Response: Indeed, we added this caveat to the discussion.

Resistance evolution

A few instances of resistance evolution in response to phage pressure are described in the 21 cases which were able to be evaluated. 7 cases experienced emergence of phage resistant bacteria, and most of these were in settings in which there was high burden of disease (respiratory tract or immunocompromised infection) or in which a phage may have been directed against a target that was not the principal pathogen. Resistance evolution was phage receptor-specific or through emergence of an alternative to the primary target pathogen. ***It is not clear whether those cases in***

17which no resistance emerged simply had no post-treatment samples available or whether they were true cures and there was no clear correlation with microbiological eradication or failure thereof.

Response: Possible *in vivo* phage resistance emergence could only be adequately investigated in 16 patients, for which sufficient pre-, inter-, and post-treatment sampleswere available. In seven of these 16 patients (43.8%), phage-resistant strains were shown to emerge. In nine patients, bacterial phage-resistance was not observed.

In the absence of adequate samples, it is not possible to discuss bacterial phage resistance in the remaining 84 patients. Due to the low number of patients in which the selection of bacterial phage resistance could be evaluated, it is impossible to correlate phage resistance with failure of eradication. For what it's worth, failure of eradication was observed in two out of the nine patients (22%) for whom no phage resistance was observed, while failure of eradication was observed in three out of seven patients (43%) in whom phage-resistant strains emerged.

A statistical model was applied to predict this but is hampered by the integrity of the data making up the variables. ***It is also not clear whether the more problematic or refractory cases were those in which evaluation was more extensive and therefore whether it is appropriate to generalise a percentage (the reader might assume that***

~40% of all phage therapy leads to R evolution in vivo – that may be so but these data do not show that). It would be sensible to add a small caveat in that regard into the discussion. If the authors can demonstrate that these 21 cases are representative of the 100 and therefore these data generalisable, that would be valuable.

Response: In seven of 16 (43.8%) analyzed cases, the selection of phage-resistant strains was documented. These 16 analyzed cases were those for which sufficient samples were available, which depended on the treatment centers, rather than on the complexity or severity of the cases. We are not allowed to link data to treatment facilities or physicians. This caveat is added to the discussion, stressing that this this percentage applies to these 16 cases and cannot be generalized.

Q: was growth kinetics performed on every isolate-phage comparison or were they all subjected to EOPs but only a few to growth kinetics? It seems as though many of the later ones went through the Omnilog system – is that the subset that was reported and, if so, could this be more clearly reiterated?

Answer: OmniLog growth kinetics can be determined since 2021, upon request of the treating physicians. Some growth kinetics were determined retrospectively using bacterial strains and phages that were stored at -20 °C. In this manuscript we only show some relevant/interesting growth kinetics (bacteriophage-antibiotic interactions) for the most recent patients as the older ones had been published previously. This is clarified in the revised manuscript.

In addition, a few specific suggestions and queries (Q), by line:

Line 83: ...e.g. by 'compounding pharmacies' in countries like the US.

Response: We adapted this in the revised manuscript. Line 86:

delete brackets; add 'for' before '...urgent cases' **Response:** We

adapted this in the revised manuscript. Line 121: 'a total of'

Response: We adapted in the revised manuscript. Line

150: Q – 8×10^9 , SD 1×10^{10} – please confirm

Response: We confirm. Phage titers can vary significantly (from 10^5 to almost 10^{11} PFU/mL), depending on the phage and its production host.Line 177: comparisons

Response: We adapted this in the revised manuscript. Line

194: Q – define ‘moderate’ activity?

Response: Moderate activity means that the phage exhibits an efficiency of plating (EOP) ≤ 0.01 on the patient’s strain. This definition is added to the revised manuscript.

Line 227: delete ‘at’

Response: We adapted this in the revised manuscript. Line

256: Q - also on quinolones? (pt 91?)

Response: We do not know. This was not tested, as nearly all quinolone antibiotics in use today are fluoroquinolones.

Line 260: Q – resensitised or generally unfit? Did you do growth curves?

Response: Growth curves were determined and the re-sensitized mutant showed no impaired fitness. This information was added to the revised manuscript.

Line 283: omnilog growth curves fig 3/ table 2 showed synergy in 9/10: q – please define how synergy was defined? Duration of suppression? Relative final OD? Etc

Response: Our synergy definition is indeed based on the duration of suppression. We speak of synergy when the growth suppression period produced by the addition of both phage and antibiotic is clearly longer than the sum of the suppression periods induced by the antibiotic and the phage separately. The definition was added to the revised manuscript (Methods).

Line 296: 5/13 developed immune neutralisation and some were partial; Q – was this predicted by phage exposure (do we know or can we calculate intensity of phage viraemia after administration?)

Response: Unfortunately, we have no idea of the intensity of phage viremia after administration as we had only access to samples harvested in function of standard of care treatment (e.g. bacterial culture to monitor the infection), and no prospective sampling. We can thus only mention that phage neutralization only occurred upon invasive phage applications in the 13 patients for which we had an adequate sample set.

Line 388: Q – confirm 24-28 hour suppression of growth as this seems to vary in the figures displayed. Did you mean plaque persistence as clear?

Response: We confirm. It varies indeed, from phage to phage, and also depends on the targeted bacterial species. Nevertheless, 24-48 h suppression of growth in liquid medium is indeed the goal. We admit that in some rare occasions this goal was not fully achieved or necessitated the pre-adaptation of the phages to the patient’s strains. The growth curves displayed in Figure 3 relate to the evaluation of bacteriophage-antibiotic- bacteria interactions and were obtained using a sub-optimal ratio of phages to bacteria (= multiplicity of infection (MOI)), and antibiotics were added a sub-MIC concentrations. These sub-optimal conditions are necessary to enable the analysis of possible synergy or antagonism. If either phages or antibiotics would be applied in optimal concentrations, leading to the efficient killing of the bacterial strain by either antibiotics or phages, it would be impossible to demonstrate synergy or antagonism. This information was

21added to the revised manuscript.Line 481: Q – or simply continue the antibiotic/ surgical Rx etc?

Response: That is a possibility, but treating physicians and authors assumed that the clinical resolution was due to the application of bacteriophage-antibiotic combination.

Line 488: It might be avoid the overgeneralisation – where there is a simple trade-off (e.g. selecting against a pump or porin) that is well tolerated in optimal growth media then the ab susceptibility in vitro evidently changes – it is not a general property of phages but rather a happy coincidence that can be predicted to some extent by the receptor. It might be good to be more clear about this in lines 489-90.

Response: As the selections against a pump or a porin occurred in the patients, not in optimal growth media, it seems that the mutations we observed were also sufficiently tolerated *in vivo*. Beneficial (for the patient) trade-offs are indeed not a general property of phages, but the accumulation of knowledge about phage receptors and their mutations (e.g. their consequences for bacterial fitness and antibiotic resistance) can direct the choice of phages and phage-antibiotic combinations, making these trade-offs less coincidental. This is clarified in the revised manuscript.

Line 540: implies successful phage therapy – can the authors provide evidence that the phage at least amplified in the host bacteria and therefore might have contributed to outcome or do the authors think this needs a caveat e.g. since this is the Discussion section, it might be worth pointing this out here.

Response: We have no evidence of phage amplification and we added this caveat to the discussion.

Line 557: did the authors demonstrate the advantages of combining BT with standard of care? They certainly showed some in vitro evidence of synergy and can reasonably extrapolate this to increased antimicrobial effectiveness – perhaps rephrase this?

Response: Yes, the *in vitro* synergy data points into that direction, but also the overall statistical analysis showed a higher probability of eradication when BT was combined with standard of care antibiotics. We rephrased the sentence to clarify the statement.

Its probably important to keep this caveat to a very simple statement however in order to avoid difficult arguments about antibiotic effects which cannot be addressed from these data – e.g. if aminoglycoside efficacy is mostly related to the post antibiotic effect while beta-lactam killing is mostly related to maintaining time above MIC, how do these relate to phage effects?

Response: The presented data does indeed not allow for a more specific analysis of phage-antibiotic interactions.

Line 904: Q - so 48h suppression only required for cocktails?

Response: Yes, for cocktails the requirements are stricter in terms of suppression time and MOI. This is clarified in the revised manuscript.

Line 906: Q – greater than or equal to 1?

Response: No, smaller than or equal to 1.

Line 936: Are there guidelines in regard to dosing etc in the instructions (refs 22-24) and are

they based on published data in regard to kinetics? It would be good to make clear to the reader whether this is “level IX” (expert opinion) or higher quality evidence and, if that is the case, to provide references to those data.Response: The instructions (in Russian) do not mention any kinetics data and do not mention any references to published data. One of them mentions phage neutralization data (“neutralization can emerge between days 10-15 after intravenous application”). We have not been able to determine whether these 30 to 40-year-old guidelines and instructions may be based on systematic studies, or if they are largely based on empirical experience. Therefore, we prefer to catalog them as Centre for Evidence-Based Medicine (CEBM) evidence level 5, i.e. recommendations formulated by experts, on their own professional experiences. This evidence can also include literature reviews of relevant studies, case reports, and institutional reviews of data based on non-systematic studies.

Line 976: Q - “ab MIC” ? MOIs less than or equal to 1 or MIC 1.0? Does sub-MIC concentrations means 0.25 x MIC or something else (was this standardised?)

Response: Phages were added at a sub-optimal multiplicity of infection (MOI), i.e. less than or equal to 1.0, and antibiotics were added a sub-MIC concentration. Sub-MIC concentration means 0.5 x MIC. These sub-optimal conditions are necessary to enable the analysis of possible synergy or antagonism. If either phages or antibiotics would be applied in optimal concentrations, leading to the efficient killing of the bacterial strain by either antibiotics or phages, it would be impossible to demonstrate synergy or antagonism. This information was added to the revised manuscript.

Line 989: Q – please confirm: 0.9 mL of 1: 100 diluted sera is ~10% serum v/v, without complement depletion

Response: The 1:100 dilution of the serum samples results in 1% serum (v/v), and the addition of 0.1 mL of bacteriophage suspension to 0.9 mL of 1% serum further reduces the serum concentration to 0.9 %, indeed without complement depletion.

The applied concentrations and incubation times conform to the standard technique developed by M. H. Adams in 1959, and which was developed to only detect specific bacteriophage neutralization activity. This information was added to the revised manuscript.

Table 1: Q - dosing heterogeneity (MNOI and as intervals) is striking –q 6 to q24 h – can we confirm this to be arbitrary clinician choice and identify as such – are there any kinetics data to inform this? Were these doses based on any published data eg Schooley et al 2017 or was this about total dose - are these doses discretionary and are there any concerns about consistency within and/or between protocols?? See Q above re Russian MoH protocols.

Answer: Dosing is based on the Georgian/USSR (mostly empirical) experiences. Of course, the indication and application routes also influence the intervals. For instance, it is much easier to apply phages four times per day to the infected lungs of an intubated patient (nebulization) than to infected burn wounds (topical), which are generally unpacked and treated once a day. This was clarified in the revised manuscript.

Q - pts 30,54,64,82 and 91 – are these the only R? or are the others simply not known as no specimens were available or because it did not occur? is the group that was singled out for testing representative of the whole 100 (i.e. can we generalise these data at all?)

Answer: The *in vivo* selection of phage R was analyzed in 16 patients for which sufficient pre-, inter-, and post-BT bacteriological samples were available (i.e. the 21 patients in Table 2, minus

five patients for which no samples were available (indicatedwith “NSA” in Table 2)). The selection of patients is based on the availability of sufficient adequate samples, which depends on the routine sampling regimens of the treatment centers. It is not a prospective study, so we were not allowed to suggest direct (additional) sampling. The bacteriophage resistance data cannot be generalized and this is stressed in the revised manuscript.

In case 66 – no benefit – were these M abscessus uniformly susceptible? (e.g. the common occurrence of ‘rough’ and ‘smooth’ types in the population is often difficult to recognise and may relate to therapeutic failure, esp if unrecognised early on). M abscessus is a common and important indication for phage therapy and this population variation is common so therefore may be worthy of a comment if you have data.

Response: The *Mycobacterium abscessus* strain of patient 66 clearly and consistently exhibited the rough colony morphotype. This information was added to the revised manuscript.

Q - cases 22,23,24,42 – is it possible to identify the probable causative pathogens? E.g. Staph aureus in this context is rarely an irrelevant coloniser etc – complex mix of pathogens in orthopaedic cases – all got better but is phage therapy always targeting the main pathogen?

Nb. cocktails are all broad spectrum (multi organisms) rather than multi target so no phage synergies in these – can we identify those in which true synergies were sought and proven in vitro and if so, did they have better outcomes?

Response: The “cases 22-24 and 42” line in Table 2 summarizes four patient cases. For the sake of clarity (e.g. which pathogen was targeted by which phage), we assigned one line to each of the four patients in the revised Table 2. Note that all cocktails were composed of phages with compatible activities. We did not observe statistically significant differences in outcomes between phage cocktails (or single phages). This was specified in the revised manuscript.

Q - of the cases in which resistance arose, these were dominated by Pseud infections and the resp tract. Is this because respiratory samples were easiest to obtain or are you inferring that this is a scenario in which R is common?

Response: We also observed the prevalence of *P. aeruginosa* and respiratory tract infections, and we feel that especially the *P. aeruginosa* overrepresentation might be significant. However, *P. aeruginosa* and respiratory tract infections are by far the most prevalent pathogen-indication combination in the considered patient population, and the number of patients for which we obtained adequate samples (and subsequent resistance data) is too small to make strong conclusions. We added this consideration to the revised manuscript.

Nb. Ext Table 4 – is it possible to use standard EudraVigilance terminology and definitions here?

Response: Yes, we used standard EudraVigilance terminology and definitions in the revised Table 4, and in the main text.

Fig 1a – could be deleted

Response: Yes, Fig. 1a could be deleted, but it provides a quick visual overview of the geographical spread of the considered bacteriophage therapy cases. We leave the choice to the editorial team.

Fig 1b – regulatory pathway is EU-specific and could also be deleted and simply mentioned in text

Response: Yes, it could be deleted, but it illustrates the variety of regulatory pathways, and the increase of standard of care treatments using magistral phage preparations. We leave the choice to the editorial team.

Fig 1c – ‘primary’ or ‘secondary’ infection – please define

We decided to no longer speak of primary, secondary or co-infections, but of infections and second-site infections, terms that are less likely to cause confusion. The manuscript was adapted accordingly.

Fig 1d – nice diagram although also inessential

Response: It is indeed not essential. It shows the patient age and gender distribution, data that is not provided in any table or text (we were not allowed to link demographic data to results). It shows, for instance, that five children younger than one year were treated. We leave the choice to the editorial team.

Summary

This is an important contribution from the field leaders because it describes 100 phage therapy cases and is the largest set described in recent times. The authors are clear about the limitations of the data, and I have raised a few extra questions that may be able to be addressed. The authors are pointing us in the direction we need to take.

The data are difficult to make much from in regard to outcomes, immunity, kinetics or even safety. An interesting variation in the development of neutralising antibodies is described on but we can only speculate on the cause and implications. Whether the ‘negative’ reports in this case series are really phage therapy failures is as uncertain as whether the ‘positive’ outcomes are really phage therapy successes. The authors are completely transparent about this but nevertheless put forward one of the largest case series ever in phage therapy and should be congratulated for making such a thorough analysis of a difficult data set.

With regard to the authors closing remarks, while there are multiples sites doing this now, this collaboration is the international lead and have shown that it is indeed feasible to set up a flexible and safe personalised BT system.

It is not clear that 77% of these cases were successful because of the (phage) intervention as other important contributing (confounding) variables are not controlled in retrospective series. However, we can accept that the referral was made because improvement was not occurring nor thought likely. It is therefore accurate to say that treating physicians regarded BT to have been associated with clinical improvement in 77% and I think that is how it is best expressed.

In vivo development of resistance was evaluated in only a small subset in vivo and it is not clear that this can be generalised, nor is it clear how to interpret the interesting variation in neutralising ab development because the data needed to do so are simply not available.

Finally, future RCT design needs clarity around both dosing (that is, how much phage and how often) and the expected effect or benefit size (to understand the statistical power). The bioavailability/ dosing question is not addressed here.

Nevertheless, these data do indicate that the effect size may be quite significant and that doses of 109pfu or even less for a couple of weeks or less may suffice.

We thank reviewer 1 for the comments that will help us improve this manuscript.REVIEWER #2

General comments

This manuscript provides an immense amount of data that will be of interest to those engaged in clinical and translational research related to bacteriophage therapeutics. As the authors note, the main shortcoming of the analysis is that there was not a uniform, prospectively specified definition of clinical treatment success. This is one of the same challenges facing those wishing to interpret similar retrospective case series published from Poland, Georgia and other former Soviet bloc countries.

Having said this, the manuscript meticulous in detail and vast in scope. Unlike many of the aforementioned retrospective reports from the Eastern bloc, this analysis presents data that more directly touch on important issues facing contemporary phage therapeutics: How often do patients raise an antibody response to phages that are administered therapeutically? Does this affect treatment outcomes? Does this occur more often when phages are administered parenterally as opposed to locally? How often to bacteria with reduced phage susceptibility arise during phage therapy? Are these bacterial variants as “fit” or invasive as pre-treatment strains? If possible, should phages be administered with antibiotics that are active against the bacteria being targeted by bacteriophage therapy? The manuscript does not provide definitive answers to these questions but in many cases it presents data of sufficient quality and rigor to further shape the research agenda.

Providing data about 100 patients in chronological order is highly meritorious and helps better define success rates since this approach begins to address issues of positive selection bias from the case report literature. As best one can tell, however, the authors are presenting what one might call an “as treated” analysis. Two critical pieces of information that are missing from the manuscript are the denominator from which these 100 “as treated” patients were drawn and how often therapy was recommended but no phage could be identified and produced in time to treat the patient (e.g., had this been a modified intent to treat analysis what would the success rates have been?).

RESPONSE TO REVIEWER 2:

Thank you for the constructive comments. We also believe this manuscript will be great interest to the clinical and translational field. Please see our point-by-point response to the questions and points raised.

Can the authors provide data about **how many patients were proposed** to the Bacteriophage Therapy Coordinating Center in the time period covered by this report (e.g., between January 1, 2008 and April 30, 2022)? Likewise, during this period, **how many patients were recommended for phage therapy but not treated because phages could not be identified?**

Response: Between January 1, 2008 and April 30, 2022, 1,066 BT requests were

30submitted to the phage therapy coordination center of the Queen Astrid military hospital (QAMH). These requests resulted in 100 BT cases (9.4%).

For a detailed assessment of requests made to the QAMH, please see our previous publication by Djebara et al., which assessed the period between April 2013 – April2018. In this publication it is stated that, “Only 15 (5.8%) of the 260 phage therapy requests resulted in actual phage therapy. Two hundred and forty-five requests were rejected for diverse reasons.” and “5 (25%) out of the 20 infecting bacterial strains for which a “phagogram” was performed were found to be non-susceptible to the available phages.”

Considering that the manuscript concerns patients who were treated during a period of more than a decade, this “denominator” will be rather large, so we did not do a detailed analysis of this type of information at this time. We consider these percentages as representative of the present patient cohort, minding an increase of the percentage of requests that resulted in BT (9.4% vs. 5.8%), which is due to the increasing number of therapeutic bacteriophages in the QAMH collection. We included a statement about this in this manuscript for emphasis.

Reference: Djebara, S., *et al.* Processing Phage Therapy Requests in a Brussels Military Hospital: Lessons Identified. *Viruses* **11**(2019).

One of the issues that is a bit unclear about the triage process is whether the 100 patients who were ultimately selected for therapy were ones for whom an active phage was already in hand at the time of the request (as opposed to situations in which screening of environmental phages was required).

If candidates were not recommended for treatment or did not receive treatment, what were the reasons for this? Did they have inappropriate clinical conditions? Did they have bacterial infections for which lytic phages were not available in a clinically relevant time frame? Etc. Please clarify.

Response: For reference, the criteria for selection are listed in pg. 3 lines 91-92 “The selection of patients was largely based on clinical need, regulatory approval, and the availability of well-characterized bacteriophages targeting the infecting bacteria.” The selection process is detailed in Extended Data Figure 1.

As answered previously, a detailed assessment was not performed here. However, in Djebara et al., the major reasons for lack of recommendation are listed below. As stated above we believe the percentages highlighted in Djebara et al. are representative of this patient cohort, minding an increase of the percentage of requests that resulted in BT (9.4% vs. 5.8%), which is due to the increasing number of therapeutic bacteriophages in the QAMH collection. In Djebara et al. it is stated that,

Only 15 (5.8%) of the 260 phage therapy requests resulted in actual phage therapy. Two hundred and forty-five requests were rejected for diverse reasons (Figure 1):

- *70 applicants (26.9%) did not respond to the email request for more information;*
- *124 requests (47.7%) concerned bacterial pathogens against which the QAMH had no potent phages available;*
- *46 applications (17.7%) did not meet the other two eligibility criteria (antibiotic treatment failure and/or absence of other therapeutic options);*
- *5 (25%) out of the 20 infecting bacterial strains for which a “phagogram” was performed were found to be non-susceptible to the available phages.*

We included a statement about this in this manuscript for emphasis.

What was the mean time from a treatment request to the initiation of therapy? If you did need to perform environmental searches and prepare phages for administration from scratch, how long did this take on average?

Response: As this manuscript spans more than a decade of work it is difficult to assess the mean time to treatment for these cases. Mean time to treatment is dependent on the availability of phages produced and quality controlled on hand (these can be provided immediately) as well as time required to produce the phages and to perform quality and safety tests performed by Sciensano. We estimate that the time required would be three weeks in non-emergency cases. We added this information to the revised manuscript.

More information about the specific objective criteria used by the Bacteriophage Therapy Coordinating Center to recommend or discourage phage use would benefit others trying to adapt this experience to their own circumstances.

Response: Thank you. We added the eligibility/rejection criteria (see also previous questions).

Comments in the manuscript about specific regulatory aspects of phage therapeutics are interesting and important in the European context but they might not be generalizable to other settings. The authors are correct on line 401 about the challenges posed by strict GMP manufacturing standards for individualized therapeutic application of phages. While it is true that the “magistral” phage production approach has been quite workable in the hands of the investigators, the approach described in the section beginning on line 790 is not strictly identical to the US compounding laboratory procedures to which they compare magistral production in Europe. The authors are to be commended for the extent to which they describe and adhere to manufacturing and quality control aspects of their work in the sections beginning on lines 790 and 826. The use of a specific organization (Sciensano) to monitor quality and safety would not be required in the compounding pharmacy “industry” in the US. This has led to a number of major fiascos such as the introduction of aspergillus into methylprednisolone that was injected into the CSF by unscrupulous neurosurgeons in Nashville, TN. (1) Individual patient phage in the US use analogous to that described by the authors is governed by the US Food and Drug Administration’s individual patient IND process. In this situation the FDA provides guidance about the level of production quality required for individual patients on the basis of a dialog with the physician about the urgency of treatment and the practicality of additional phage characterization and purification steps. The process works well and assures a level of oversight that has avoided mishaps that might occur if clinical phage preparation devolved to individual laboratories that were “overseen” by so-called expert committees at the hospitals in which the phages were to be administered. The QAMH experience is a bit different from an unregulated compounding approach in that a highly experienced group of phage researchers have prepared 40 batches of phages using rigorous pre-specified procedures that were QC’ed by a single experienced organization. Thus, while it is important (and indeed necessary) to describe the approach taken by QAMH, it should be stated that other oversight and regulatory approaches may be preferable in other settings.

1. Chiller TM, Roy M, Nguyen D, Guh A, Malani AN, Latham R, Peglow S, Kerkering T, Kaufman D, McFadden J, Collins J, Kainer M, Duwve J, Trump D, Blackmore C, Tan C, Cleveland AA, MacCannell T, Muehlenbachs A, Zaki SR, Brandt ME, Jernigan JA; Multistate Fungal Infection

Clinical Investigation Team. Clinical findings for fungalinfections caused by methylprednisolone injections. N Engl J Med. 2013 Oct 24;369(17):1610-9. doi: 10.1056/NEJMoa1304879. PMID: 24152260.

Response: Thank for your comments and insight, which we included in the revised manuscript.

We were not aware of the significant differences in pharmacy preparations between the EU and the US.

Other oversight and approaches may indeed be preferable in other settings.

In the US, the FDA IND individual case-by-case assessment can be a time-consuming process and will not be ideal for this type of treatment especially in cases of dire emergency. Even though there is an emergency IND that can speed up this process, the FDA does not encourage single patient INDs. As stated in their website, the FDA wants to see phages put through classical clinical trials.

Minor comments

Line 45 Small point but I'd suggest inserting the word "perceived" since commercial sustainability is in the eyes of the beholder. Big Pharma is not interested in drugs that are merely "commercially sustainable" but rather blockbusters that yield blockbuster returns. Thus, I hate to let them off the hook by saying that antibiotics are not commercially sustainable and would suggest qualifying this comment.

Response: Thank you. We made this addition.

46 The AMR Action fund hopes to bring 2 – 4 antibiotics to the market by 2030. The way the sentence is worded ("is expected") is imprecise. Those donating the money may expect for this fund to do just this but it is not a foregone conclusion that this would happen. It would be more accurate to say that the fund was created with the hope that 2 – 4 new antibiotics could be brought to market.

Response: Thank you. We made this addition.

171 Can the authors identify experimental data that define an EOP>0.1 as being "therapeutically acceptable"?

Response: We base our assessment criteria on the expertise from the George Eliava Institute of Bacteriophages as stated in the manuscript. For clarification we changed this statement to say "We considered an EOP \geq 0.1 on the patient's bacterial strain as therapeutically acceptable based on the expertise from the Eliava Institute".

337 and Figure 1c Recognizing all of the complexities of retrospectively analyzing a case series in which patients were selected for bacteriophage therapy based on the perception of a multidisciplinary team that phages might be of benefit clinically and in which clinical outcomes were determined by the treating physician without a formal definition, ***can the authors provide treatment "success" rates for infections of the various types depicted in Figure 1c?*** Such data might help others make more informed decisions about selecting candidates for phage therapy.

Response: Thank you, we included information of clinical improvement and eradication of the targeted bacteria in Supplementary tables 1 and 2. We agree that some of this data should also be at the forefront. To address reviewer 3 as well, we have included treatment outcomes in

Figure 1c.

We thank reviewer 2 for the comments that will help us improve this manuscript.REVIEWER #3

The manuscript by Pirnay et al., describes a retrospective study of the first one hundred personalised phage therapy cases treated by the Belgium consortium, which included groups across Europe, and targeted a broad range of difficult-to-treat infections with antimicrobial-resistance determinants. This is an incredibly important study and I want to upfront thank the authors for their work, progress, and impact they have made on the field. I especially enjoyed the juxtaposition between phage therapy approaches, mainly the disappointing results that have been seen from RCT using pre-defined phage cocktails compared with the personalised, phage-for-service model proposed here.

With regards to the manuscript, I believe this will be a seminal paper in the field that will likely define much of what the field prescribes for personalised phage therapy moving forward. It is for this reason that I have been especially picky and nuanced in my comments provided here. I have no major concerns with regards to the results, statistics and data presented in the paper, and much of what is presented should be considered state-of-the-art with respect to personalised phage therapy. However, in my opinion, the authors have not done a good job of presenting their data in the most impactful way, nor have they effectively communicated the work in an accessible way, particularly for a broader readership in Nature Microbiology. As such, much of my comments and suggestions will be focused on these two criticisms.

The impact of this paper, in my opinion, is on the retrospective treatment of 100 patients with personalised phage therapy. I encourage the authors to keep emphasis this front and centre when revising the manuscript in order to maintain focus, while significantly shortening their writing. I have two major summary points for the authors to address. The first is revising the overall structure of the manuscript, which is not traditional, which is fine, but as a result there are many sections and paragraphs that are not well placed, breaking the flow and narrative of the paper. Secondly, the paper is unnecessarily long and at times unfocused. An example of this is found in the discussion, which is nearly five solid pages of text, it has little organisation in its writing, and has many sections that are better suited to a literature review rather than being included within a retrospective analysis that is presented here. In brief, the authors are doing too much, and in my opinion, this has negatively impacted the paper and its message.

Please see below line-by-line comments and suggestions:

RESPONSE TO REVIEWER 3:

Thank you for your kind words. We believe this manuscript to be important to the field as well and we hope we have improved it in order to effectively communicate its importance. Please see below for a response to comments and questions raised.

Lines 92-96: 'Quality & safety were verified by Sciensano' - I think it would be of benefit to the field (and future regulators) to briefly state ***how these preparations were verified. i.e., sterility testing? Endotoxin? Sequencing? In vitro screens?***

37Response: Thank you. The details of quality and safety tests are explained in the manuscript pg. 18 lines 826-857. However, to address this question and the nextquestions raised, we included a short description in the Introduction and more specifics in the Methods section.

Lines 101-102: As noted in the abstract, not all cases employed these assays - could the authors include a short note on this point?

Response: Yes, the assays were offered to the physicians, but were only reported when sufficient and adequate consecutive bacterial samples and/or serum samples were provided to allow for a meaningful analysis. We included a note on this (in the Results section).

Lines 101-108: Is this best suited for the introduction? This writing reads more like a method and I personally think it breaks the flow between impact of Eliava and the summary of this article.

Response: Yes, this was moved to Methods.

Line 151: Can you provide *further details on the methodology used for your bioburden assays? Were these USP71 sterility tests? If so, how much volume or fraction of the batch was tested? Or was this a plating and CFU count performed by Sciensano?* This is important to define as a 0 CFU count can be a false positive depending on your volume tested compared to batch production. This detail also needs to be clearly defined in the methods, but a brief summary statement in main text is also justified due to the importance of these end-point metrics. I also note the associated methods section was very limited on these important points (lines 852-854). Readers should not have to go to other documentation/papers to find these key criteria.

Response: As stated in lines 853-854, the bioburden (total viable aerobic count) of each lot was assessed as described in European Pharmacopoeia (Ph. Eur.) chapter 2.6.12. The European pharmacopoeia provides guidance on test methods for microbial contamination of pharmaceutical preparations, and these are followed by Sciensano. The volume assessed was 4 mL (1.6-2.6% of 150-250 mL production batches).

We agree that the usage of less than 10% of the produced batch can result in false negatives. However, similar amounts are also used to assess sterility for antibiotic liquids. Moreover, although indirectly, the presence of any bacterial or fungal contaminant is also assessed by verifying the occurrence and amount of residual DNA present in the sample (purity scored by the % of phage reads) and the presence of endotoxins. We understand that more clarity is needed on these assays, so we have included more clarification in the manuscript.

Line 152: *Please clarify - were the median EU measurements of your phages 5 EU/mL of product? Or median of 5 EU/mL per patient kg?* I believe it was the former, but this should be very clear in the writing. Again, associated methods were poorly described.

Response: Correct, this was the product concentration. Additional clarifications were introduced into the revised manuscript.

Lines 154-157: *Do these requirements hold for the Eliava phage cocktails?* I note these were administered via topical, nebulised, rectal, and intralesional. From my understanding these products are not produced within safe/FDA limits for endotoxins

Response: The Eliava bacteriophage cocktails were not quality-controlled by Sciensano. Due to

their likely higher endotoxin content, these products were never administered intravenously. This information was added to the revised manuscript.Lines 159-160: Same comment as above for Eliava phage cocktails. From my understanding these cocktails are not genetically defined. In your cases using these products, were these sequenced? If so, how did you confirm <5% bacterial reads if host genomes were not available/known?

Response: Indeed, these cocktails were not genetically identified by Sciensano. We could thus not confirm < 5% bacterial reads. We included a statement about these exceptions in the manuscript.

Lines 176-199: In my opinion, this section is an example of where the authors have tried to include too much detail and depth in the manuscript, and I think it detracts from its focus.

You state that a detailed genetic comparison of adapted phages falls out of scope of this manuscript, and I agree. But you then go on and provide a detailed genomic comparison of two published adapted phages. The writing should be tightened in aspects of the manuscript, I would suggest removing this content and keeping your focus on the therapy cases. You cannot report everything in this paper, so I would suggest focusing on the clinical impact.

Response: Thank you for the suggestion. We have significantly shortened this section. We include some new information on ISP (previously unpublished) thus we would like to preserve some of the text.

Figure 1: This is a suggestion, but I feel like Figure 1 is missing an opportunity to present a simple yet impactful visualisation of the success and eradication rates of the 100 treated patients - this is likely the impactful point most readers and citations will be looking for - yet it is not graphically represented.

Response: We agree and, also based on reviewer 2 comments, we included a graph to show clinical improvement and bacterial eradication in Figure 1.

Lines 231-232: How was colicin I identified as a phage receptor? In this study or a previous one? If no knock out work was completed to validate this receptor then this should be listed as a putative receptor. Same comment with T4P in subsequent lines, either provide citation for mechanistic work, or use putative.

Response: Thank you. A citation was added for T4P that describes further mechanistic work (Ceyssens et al., 2011). For colicin I, this was corrected to say "putative receptor".

Lines 238-243: When discussing patients 54, 30, and 91 please be clear that these patients were all infected with P.a.

Response: We clarified this.

Lines 264-276: Please add a concluding statement for the Galleria model section and define the initial goals of this assay. I would also ask the authors to consider its relevance within this article. The impact here is patients treated, and I'm not certain the animal work fits well, and in some ways, it detracts from the study.

Response: A sentence was added to the manuscript to define goals. Also, a concluding sentence was added.

We believe this is an important experiment needed to show that virulence trade-offs are seen in

phage-resistant isolates. The manuscript not only focuses on patients treated, but goes above and beyond to show follow-up data to guide future phage therapy development.Figure 3: I'd encourage authors to revise their data presentation in Figure 3 - this looks more like a supplemental figure. There is a lot of data here and it is not obvious the point you are trying to make with this figure. Further the axis fonts are almost unreadable, there are no error bars presented, and I doubt non-phage experts would be able to easily decipher this figure. Why not represent this as AUC? Or another condensed metric to emphasize the phage vs phage+Abx comparison? Then include these curves as supplemental figures.

Response: Thank you for the suggestion. While AUCs are great to quantify the growth of a bacterial strain over a certain time period, some information is lost (e.g. after how many hours does the growth of resistant mutants occur). Representing the data as AUCs is surely warranted for large data sets, for statistical analysis, to compare groups, etc.

When limited to only 12 experiments, we feel that the growth curves are visible representations that we think readers of Nature Microbiology could readily understand and interpret. This is why we left this analysis as a growth curve. It is also impactful to show that these studies were performed in 72-hour assays which is not common in the field. We do agree that the axis fonts are unreadable and revised the figure to make the axis more readable.

Lines 282-287: It is worth to briefly mention how synergy and additive effects were calculated/distinguished.

Response: Our synergy definition is indeed based on the duration of suppression. We speak of synergy when the growth suppression period produced by the addition of both phage and antibiotic is clearly longer than the sum of the suppression periods induced by the antibiotic and the phage separately. The definition was added to the revised manuscript.

Lines 278-293: It was unclear from this section, was phage-antibiotic synergy only tested in vitro? Did these results impact or influence the patient treatment? Were these synergistic/additive combinations used clinically? How did this data inform the clinical treatment of the 100 patients?

Response: These tests are performed *in vitro* as stated in the main text and in the Methods section. When most of these tests were performed, bacteriophage therapy had already started and test results did not influence patient treatment. However, today, based on these test results, and the overall observation that pathogen eradication is more likely when phages are applied in combination with antibiotics, we strongly advise physicians to perform these tests before treatment, when time permits. This information was added to the revised manuscript.

Figure 4: This figure can be improved by overlaying additional treatment information, such as the duration of phage therapy, clinical improvement and bacterial eradication (if these occurred), and potentially co-administration of antibiotic treatment.

Response: We added the suggested additional information to Figure 4.

Lines 318-321: Please rephrase as it currently reads as if five patients died of septic shock, when this was only two.

Response: This was clarified in the revised manuscript.

Discussion: The discussion was close to five pages of solid text. It at times, lost focus and read more like a literature review or thesis, rather than a critical discussion on the 100-

cases treated. Further, the structure of the discussion seemed erratic, jumping from topic to topic without much flow or coherency.

After finishing this section, I found I had missed/forgotten the impact of this work, which in my opinion is the treatment of 100 BT cases, and was more distracted by the writing, historical musings, and summary of prior studies.

I strongly encourage the authors to review this discussion with the goal to focus their writing on critical points, to reduce the text by over half, and to maintain a logical flow in the discussion that summarises the major findings across their study.

Response: Thank you for the suggestions. Since this manuscript is an accumulation of data from 100 patients, it is important to have a longer discussion to make several points that have been raised in this manuscript. For brevity we have tightened the discussion.

Line 356: Please remove the dashes here, it breaks sentence flow

Response: We removed the dashes.

Lines 386-388: This definition is an idyllic goal. More specifically, I note that most (arguably all) of your biolog growth curves (Fig3) show the emergence of phage resistance at low MOIs within as short as seven hours and stable lysis for 24-48 hours was not evident. This is to be expected for in vitro growth curve assays. Yet by your own definitions, these phages do not constitute an efficient therapeutic phage, yet they were used with promising success rates. So how do you reconcile these discrepancies and adhere to your own definition?

Response: These standards that we have described are not an idyllic goal but a standard when adapting phages for treatment as described in lines 164-168. A 24-48 h suppression of growth in liquid medium is indeed the goal. We admit that in some rare occasions this goal was not fully achieved or necessitated the pre-adaptation of the phages to the patient's strains. The growth curves (Fig. 3), however, were obtained using a sub-optimal ratio of phages to bacteria (= multiplicity of infection (MOI)), and antibiotics were added a sub-MIC concentrations. These sub-optimal conditions are necessary to enable the analysis of possible synergy or antagonism. If either phages or antibiotics would be applied in optimal concentrations, leading to the efficient killing of the bacterial strain by either antibiotics or phages, it would be impossible to demonstrate synergy or antagonism. This information was added to the revised manuscript.

Lines 418-430: This section reads more like a literature review than a discussion. What approach for dosing did you take in this study? As far as I could tell, dosing information from your own study was completely absent here. Keep the description of the prior work focus on your study, particularly within the discussion.

Response: We report dosing with available information in Supplementary Table 1. Many of these cases have been extensively reported with dosing details as well (see reference in Table 2).

Lines 438-443: Recommend this text be removed.

Response: The text has been shortened.

Lines 451-464: Again, a literature review style of writing that inflates your writing and retracts for

this papers message. Recommend this text be removed.Response: We think this is an important point to make about bacterial resistance and fitness costs. This text has been shortened to be reader friendly.

511-527: Recommend this text be removed.

Response: This has been significantly shortened.

We thank review 3 for the constructive comments which improve our manuscript.REVIEWER #4

Remarks to manuscript from the viewpoint of statistics

The authors correctly claim in lines 345f that “the majority of patients were unique cases” and that “no inferential statistics could be applied”. In this sense, the reporting of figures like odds ratios can be seen as hints to possible connections without fully well- grounded statistical evidence. However, reporting confidence intervals (CIs) is part of inferential statistics. I do not say that CIs should not be mentioned. But it should be noted that they must be interpreted with caution. Fisher’s exact test (line 322) is also part of inferential statistics.

Response: We would like to thank the reviewer for the comments. We were cautious to not make inferences, hence our specific wording as rightly pointed out in line 345. However, we have decided to make this further clear by prefacing the lines 37 (abstract) and line 327 (results) with “In our dataset of 100 consecutive cases, ...” to emphasize that we are not inferring about the general population of patients.

From line 119 on can conclude that some patients suffered from coinfections. This should be explicitly mentioned. It seems that a coinfection was treated as two independent infections concerning the bacteriophage therapy. My medical is not sufficient to judge whether this is justified.

Response: Thank you. We mention the fact that patients had coinfections (we decided to call them second-site infections) in line 120 “including 14 second-site infections”. Each patient case being unique, we considered clinical improvement and eradication for patients with second-site infections in a similar way as cases with single infections only. Infections and second-site infections were not analyzed as independent infections, as they are most often linked (e.g. sepsis as a consequence of a bone infection).

In line 309 it is mentioned that “in eight patients, no adequate post BT bacteriological data was available”. What is the reason for the missing data? If an adverse event like death was the reason, this is should be taken into account in the analysis.

Response: The reason for seven out of eight of these cases is that the treatment center did not collect this bacteriological data as part of their routine follow up of patients, and was not allowed to collect the data prospectively.

For the remaining case, it is not clear why the bacteriological data was not available. The treatment center either did not collect this data, failed to extract this data from the medical files, or was not able or willing to transfer this data to us.

Adverse events were not the reason for the missing data. In two cases adverse events were indeed observed, but they had no impact on the collection of bacteriological data. The reason was that the healthcare personnel was not allowed to collect bacteriological data prospectively.

More explanations were added to the revised manuscript.

Maybe this question is naïve. But is the antibiotic resistance profile (line 323) a dichotomous variable? Otherwise it is not clear to me how Fisher’s exact test was applied.

47Response: That is correct, we considered whether the bacterial strain was MDR or not and this was encoded as a dichotomous variable. We have made this statement clearer in the manuscript where lines 323-325 now mention: Fisher exact test for count datashowed univariate significant effects on eradication for the following categorical variables: concomitant use of antibiotics (yes or no), antibiotic resistance profile of the targeted bacteria (MDR or not) and the clinical setting (Ambulatory or Hospitalized).

In lines 324f it is mentioned that “No effects of age or gender on eradication were found using a logistic regression analysis.” This was univariate logistic regression considering solely age and gender, respectively?

Response: Yes. This was specified in the revised manuscript.

Does “stepwise logistic regression analysis” in line 325 refer to forward selection (or backward elimination)?

Response: Thank you, the stepwise logistic regression analysis refers to forward selection. This has now been clarified in line 325 of the results (A stepwise, forward selection, logistic regression analysis...) and in the methods line 1047 (We used a stepwise, forward selection procedure...).

Logistic regression can easily show overfitting effects when it is tested on the same data with which the variables were selected and the coefficients were determined. The confusion matrix mentioned in line 329 should be computed based on leave-one-out cross-validation and not based on the training data.

Response: Thank you for this suggestion. We used a classical statistics approach here, with the logistic regression that assumes a binomial distribution of the errors. However, given the nature of the dataset, with a hundred unique cases, we are not in a position to test this assumption. Using leave-one-out cross-validation on the dataset is tempting, but we do not think that we could have overfit in this case as we only have one degree of freedom and do not have a dataset where a specific point would have a lot of leverage and pull the data in a certain direction (The variable “ABCONCOM” is binary, with 67 “yes” values and 33 “no” values).

Line 330 states that the logistic regression model was “right in 65% ... of the time”. What is the proportion of the majority class? If it is 65% or more, 65% would be very poor. I do not understand the connection between the confusion matrix mentioned in line 329 and the logistic regression. It seems that the confusion matrix provides the correlation between eradication and AB treatment. Where do the predictions of the logistic regression enter the confusion matrix?

Response: We thank the reviewer for raising this point. To answer the first part of the question, the majority class of the variable ERADICATION is “Yes”, observed 53 times. Importantly, this remark from the reviewer helped us realize that the confusion matrix presented in the Extended figure 4 was incorrectly labelled with the variable names instead of the predicted classes vs. true classes. This caused confusion regarding the interpretation of the matrix. We trust the corrected version of this figure clarifies the points raised here.

$$\text{Logit model: } \pi = \frac{1}{1 + e^{-(b_0 + b_1 x)}}$$

$$b_0: \text{Intercept} = 0.74 \text{ (} p < 0.01 \text{)}$$

$$b_1: \text{No concomitant AB} = -1.151 \text{ (} p < 0.01 \text{)}$$

$$\text{Odds-ratio: } \frac{\pi}{1 - \pi} = 0.309$$

π = probability that eradication is "Yes"

$1 - \pi$ = probability of "No" eradication

Confusion matrix (n=92)

		True class	
		No	Yes
Predicted class	No	20	13
	Yes	19	40

We thank review 4 for the constructive comments, which improve our manuscript.

Decision Letter, first revision:

Message: Our ref: NMICROBIOL-23092372A

7th March 2024

Dear Jean-Paul,

Thank you for your patience as we've prepared the guidelines for final submission of your Nature Microbiology manuscript, "Retrospective, observational analysis of the first one hundred cases of personalized bacteriophage therapy facilitated by a Belgian consortium" (NMICROBIOL-23092372A). Please carefully follow the step-by-step instructions provided in the attached file, and add a response in each row of the table to indicate the changes that you have made. Please also check and comment on any additional marked-up edits we have proposed within the text. Ensuring that each point is addressed will help to ensure that your revised manuscript can be swiftly handed over to our production team.

In recognition of the time and expertise our reviewers provide to Nature Microbiology's editorial process, we would like to formally acknowledge their contribution to the external peer review of your manuscript entitled "Retrospective, observational analysis of the first

50one hundred cases of personalized bacteriophage therapy facilitated by a Belgian consortium". For those reviewers who give their assent, we will be publishing their names alongside the published article.

Nature Microbiology offers a Transparent Peer Review option for new original research manuscripts submitted after December 1st, 2019. As part of this initiative, we encourage our authors to support increased transparency into the peer review process by agreeing to have the reviewer comments, author rebuttal letters, and editorial decision letters published as a Supplementary item. When you submit your final files please clearly state in your cover letter whether or not you would like to participate in this initiative. Please note that failure to state your preference will result in delays in accepting your manuscript for publication.

Cover suggestions

COVER ARTWORK: We welcome submissions of artwork for consideration for our cover. For more information, please see our guide for cover artwork.

Nature Microbiology has now transitioned to a unified Rights Collection system which will allow our Author Services team to quickly and easily collect the rights and permissions required to publish your work. Approximately 10 days after your paper is formally accepted, you will receive an email in providing you with a link to complete the grant of rights. If your paper is eligible for Open Access, our Author Services team will also be in touch regarding any additional information that may be required to arrange payment for your article.

Please note that *Nature Microbiology* is a Transformative Journal (TJ). Authors may publish their research with us through the traditional subscription access route or make their paper immediately open access through payment of an article-processing charge (APC). Authors will not be required to make a final decision about access to their article until it has been accepted. Find out more about Transformative Journals

Best regards,

Reviewer #1:

Remarks to the Author:

The authors have responded to all the suggestions of the reviewers. The original issues that limit the inferences that can be drawn and the generaliseability of the data overall are acknowledged. The revised MS will likely be of broad interest.

Reviewer #2:

Remarks to the Author:

I have no additional comments or suggestions to the authors and commend them on a conscientious effort to respond to four very detailed reviews.

Robert T. Schooley

Reviewer #3:

Remarks to the Author:

I'd like to thank the authors for their detailed responses. I have no further suggestions or changes for the manuscript.

Reviewer #4:

Remarks to the Author:

The authors have answered most of my questions. I am still not fully convinced by the logistic regression. It makes sense to use logistic regression for variable selection and to report the coefficient or the OR for concomitant antibiotics including confidence interval and p-value. However, instead of showing the confusion matrix for the logistic regression, a simple contingency table (concomitant antibiotics vs. eradication) might be more informative including a p-value for the Fisher-test.

This is just a hint because I somehow overlooked this fact in my first review. The Kaplan-Meier curves are based on a very coarse time scale (i.e. days where the considered period is only two days). This might have an influence on the p-values of the log-rank test. (Without proof, I suspect the p-values would be even smaller when the measurements were taken hourly.) At least, the authors should mention the possible limitations of the coarse time scale.

Author Rebuttal, first revision:

REVIEWER #4

The authors have answered most of my questions. I am still not fully convinced by the logistic regression. It makes sense to use logistic regression for variable selection and to report the coefficient or the OR for concomitant antibiotics including confidence interval and p-value. However, instead of showing the confusion matrix for the logistic regression, a simple contingency table (concomitant antibiotics vs. eradication) might be more informative including a p-value for the Fisher-test.

We agree. In the revised manuscript we show the contingency table instead of the confusion matrix. We also report the *P* value for the Fisher exact test.

This is just a hint because I somehow overlooked this fact in my first review. The Kaplan-Meier curves are based on a very coarse time scale (i.e. days where the considered period is only two days). This might have an influence on the p-values of the log-rank test. (Without proof, I suspect the p-values would be even smaller when the measurements were taken hourly.) At least, the authors should mention the possible limitations of the coarse time scale.

We agree. We have revised the assessment by increasing the frequency of data collection to every 6 hours, allowing us to evaluate both survival and activity scores more comprehensively. The new data is presented in a revised Extended Data Figure 3. In general, we observed similar *P* values as previously obtained (one *P* value indeed dropped from 0.004 to 0.0001). However, this time we observed a significant difference in the survival rate between the wild type and mutant group for the patient isolates from P54, but not from P30.

We sincerely thank Reviewer 4 for the comments that helped us improve our manuscript.

REVIEWER #2

General comments

This manuscript provides an immense amount of data that will be of interest to those engaged in clinical and translational research related to bacteriophage therapeutics. As the authors note, the main shortcoming of the analysis is that there was not a uniform, prospectively specified definition of clinical treatment success. This is one of the same challenges facing those wishing to interpret similar retrospective case series published from Poland, Georgia and other former Soviet bloc countries.

Having said this, the manuscript meticulous in detail and vast in scope. Unlike many of the aforementioned retrospective reports from the Eastern bloc, this analysis presents data that more directly touch on important issues facing contemporary phage therapeutics: How often do patients raise an antibody response to phages that are administered therapeutically? Does this affect treatment outcomes? Does this occur more often when phages are administered parenterally as opposed to locally? How often to bacteria with reduced phage susceptibility arise during phage therapy? Are these bacterial variants as “fit” or invasive as pre-treatment strains? If possible, should phages be administered with antibiotics that are active against the bacteria being targeted by bacteriophage therapy? The manuscript does not provide definitive answers to these questions but in many cases it presents data of sufficient quality and rigor to further shape the research agenda.

Providing data about 100 patients in chronological order is highly meritorious and helps better define success rates since this approach begins to address issues of positive selection bias from the case report literature. As best one can tell, however, the authors are presenting what one might call an “as treated” analysis. Two critical pieces of information that are missing from the manuscript are the denominator from which these 100 “as treated” patients were drawn and how often therapy was recommended but no phage could be identified and produced in time to treat the patient (e.g., had this been a modified intent to treat analysis what would the success rates have been?).

RESPONSE TO REVIEWER 2:

Thank you for the constructive comments. We also believe this manuscript will be great interest to the clinical and translational field. Please see our point-by-point response to the questions and points raised.

Can the authors provide data about **how many patients were proposed** to the Bacteriophage Therapy Coordinating Center in the time period covered by this report (e.g., between January 1, 2008 and April 30, 2022)? Likewise, during this period, **how many patients were recommended for phage therapy but not treated because phages could not be identified?**

Response: Between January 1, 2008 and April 30, 2022, 1,066 BT requests were

54submitted to the phage therapy coordination center of the Queen Astrid military hospital (QAMH). These requests resulted in 100 BT cases (9.4%).

For a detailed assessment of requests made to the QAMH, please see our previous publication by Djebara et al., which assessed the period between April 2013 – April2018. In this publication it is stated that, “Only 15 (5.8%) of the 260 phage therapy requests resulted in actual phage therapy. Two hundred and forty-five requests were rejected for diverse reasons.” and “5 (25%) out of the 20 infecting bacterial strains for which a “phagogram” was performed were found to be non-susceptible to the available phages.”

Considering that the manuscript concerns patients who were treated during a period of more than a decade, this “denominator” will be rather large, so we did not do a detailed analysis of this type of information at this time. We consider these percentages as representative of the present patient cohort, minding an increase of the percentage of requests that resulted in BT (9.4% vs. 5.8%), which is due to the increasing number of therapeutic bacteriophages in the QAMH collection. We included a statement about this in this manuscript for emphasis.

Reference: Djebara, S., *et al.* Processing Phage Therapy Requests in a Brussels Military Hospital: Lessons Identified. *Viruses* **11**(2019).

One of the issues that is a bit unclear about the triage process is whether the 100 patients who were ultimately selected for therapy were ones for whom an active phage was already in hand at the time of the request (as opposed to situations in which screening of environmental phages was required).

If candidates were not recommended for treatment or did not receive treatment, what were the reasons for this? Did they have inappropriate clinical conditions? Did they have bacterial infections for which lytic phages were not available in a clinically relevant time frame? Etc. Please clarify.

Response: For reference, the criteria for selection are listed in pg. 3 lines 91-92 “The selection of patients was largely based on clinical need, regulatory approval, and the availability of well-characterized bacteriophages targeting the infecting bacteria.” The selection process is detailed in Extended Data Figure 1.

As answered previously, a detailed assessment was not performed here. However, in Djebara et al., the major reasons for lack of recommendation are listed below. As stated above we believe the percentages highlighted in Djebara et al. are representative of this patient cohort, minding an increase of the percentage of requests that resulted in BT (9.4% vs. 5.8%), which is due to the increasing number of therapeutic bacteriophages in the QAMH collection. In Djebara et al. it is stated that,

Only 15 (5.8%) of the 260 phage therapy requests resulted in actual phage therapy. Two hundred and forty-five requests were rejected for diverse reasons (Figure 1):

- *70 applicants (26.9%) did not respond to the email request for more information;*
- *124 requests (47.7%) concerned bacterial pathogens against which the QAMH had no potent phages available;*
- *46 applications (17.7%) did not meet the other two eligibility criteria (antibiotic treatment failure and/or absence of other therapeutic options);*
- *5 (25%) out of the 20 infecting bacterial strains for which a “phagogram” was performed were found to be non-susceptible to the available phages.*

We included a statement about this in this manuscript for emphasis.

What was the mean time from a treatment request to the initiation of therapy? If you did need to perform environmental searches and prepare phages for administration from scratch, how long did this take on average?

Response: As this manuscript spans more than a decade of work it is difficult to assess the mean time to treatment for these cases. Mean time to treatment is dependent on the availability of phages produced and quality controlled on hand (these can be provided immediately) as well as time required to produce the phages and to perform quality and safety tests performed by Sciensano. We estimate that the time required would be three weeks in non-emergency cases. We added this information to the revised manuscript.

More information about the specific objective criteria used by the Bacteriophage Therapy Coordinating Center to recommend or discourage phage use would benefit others trying to adapt this experience to their own circumstances.

Response: Thank you. We added the eligibility/rejection criteria (see also previous questions).

Comments in the manuscript about specific regulatory aspects of phage therapeutics are interesting and important in the European context but they might not be generalizable to other settings. The authors are correct on line 401 about the challenges posed by strict GMP manufacturing standards for individualized therapeutic application of phages. While it is true that the “magistral” phage production approach has been quite workable in the hands of the investigators, the approach described in the section beginning on line 790 is not strictly identical to the US compounding laboratory procedures to which they compare magistral production in Europe. The authors are to be commended for the extent to which they describe and adhere to manufacturing and quality control aspects of their work in the sections beginning on lines 790 and 826. The use of a specific organization (Sciensano) to monitor quality and safety would not be required in the compounding pharmacy “industry” in the US. This has led to a number of major fiascos such as the introduction of aspergillus into methylprednisolone that was injected into the CSF by unscrupulous neurosurgeons in Nashville, TN. (1) Individual patient phage in the US use analogous to that described by the authors is governed by the US Food and Drug Administration’s individual patient IND process. In this situation the FDA provides guidance about the level of production quality required for individual patients on the basis of a dialog with the physician about the urgency of treatment and the practicality of additional phage characterization and purification steps. The process works well and assures a level of oversight that has avoided mishaps that might occur if clinical phage preparation devolved to individual laboratories that were “overseen” by so-called expert committees at the hospitals in which the phages were to be administered. The QAMH experience is a bit different from an unregulated compounding approach in that a highly experienced group of phage researchers have prepared 40 batches of phages using rigorous pre-specified procedures that were QC’ed by a single experienced organization. Thus, while it is important (and indeed necessary) to describe the approach taken by QAMH, it should be stated that other oversight and regulatory approaches may be preferable in other settings.

1. Chiller TM, Roy M, Nguyen D, Guh A, Malani AN, Latham R, Peglow S, Kerkering T, Kaufman D, McFadden J, Collins J, Kainer M, Duwve J, Trump D, Blackmore C, Tan C, Cleveland AA, MacCannell T, Muehlenbachs A, Zaki SR, Brandt ME, Jernigan JA; Multistate Fungal Infection

Clinical Investigation Team. Clinical findings for fungalinfections caused by methylprednisolone injections. N Engl J Med. 2013 Oct 24;369(17):1610-9. doi: 10.1056/NEJMoa1304879. PMID: 24152260.

Response: Thank for your comments and insight, which we included in the revised manuscript.

We were not aware of the significant differences in pharmacy preparations between the EU and the US.

Other oversight and approaches may indeed be preferable in other settings.

In the US, the FDA IND individual case-by-case assessment can be a time-consuming process and will not be ideal for this type of treatment especially in cases of dire emergency. Even though there is an emergency IND that can speed up this process, the FDA does not encourage single patient INDs. As stated in their website, the FDA wants to see phages put through classical clinical trials.

Minor comments

Line 45 Small point but I'd suggest inserting the word "perceived" since commercial sustainability is in the eyes of the beholder. Big Pharma is not interested in drugs that are merely "commercially sustainable" but rather blockbusters that yield blockbuster returns. Thus, I hate to let them off the hook by saying that antibiotics are not commercially sustainable and would suggest qualifying this comment.

Response: Thank you. We made this addition.

46 The AMR Action fund hopes to bring 2 – 4 antibiotics to the market by 2030. The way the sentence is worded ("is expected") is imprecise. Those donating the money may expect for this fund to do just this but it is not a foregone conclusion that this would happen. It would be more accurate to say that the fund was created with the hope that 2 – 4 new antibiotics could be brought to market.

Response: Thank you. We made this addition.

171 Can the authors identify experimental data that define an EOP>0.1 as being "therapeutically acceptable"?

Response: We base our assessment criteria on the expertise from the George Eliava Institute of Bacteriophages as stated in the manuscript. For clarification we changed this statement to say "We considered an EOP \geq 0.1 on the patient's bacterial strain as therapeutically acceptable based on the expertise from the Eliava Institute".

337 and Figure 1c Recognizing all of the complexities of retrospectively analyzing a case series in which patients were selected for bacteriophage therapy based on the perception of a multidisciplinary team that phages might be of benefit clinically and in which clinical outcomes were determined by the treating physician without a formal definition, ***can the authors provide treatment "success" rates for infections of the various types depicted in Figure 1c?*** Such data might help others make more informed decisions about selecting candidates for phage therapy.

Response: Thank you, we included information of clinical improvement and eradication of the targeted bacteria in Supplementary tables 1 and 2. We agree that some of this data should also be at the forefront. To address reviewer 3 as well, we have included treatment outcomes in

59Figure 1c.

We thank reviewer 2 for the comments that will help us improve this manuscript.REVIEWER #3

The manuscript by Pirnay et al., describes a retrospective study of the first one hundred personalised phage therapy cases treated by the Belgium consortium, which included groups across Europe, and targeted a broad range of difficult-to-treat infections with antimicrobial-resistance determinants. This is an incredibly important study and I want to upfront thank the authors for their work, progress, and impact they have made on the field. I especially enjoyed the juxtaposition between phage therapy approaches, mainly the disappointing results that have been seen from RCT using pre-defined phage cocktails compared with the personalised, phage-for-service model proposed here.

With regards to the manuscript, I believe this will be a seminal paper in the field that will likely define much of what the field prescribes for personalised phage therapy moving forward. It is for this reason that I have been especially picky and nuanced in my comments provided here. I have no major concerns with regards to the results, statistics and data presented in the paper, and much of what is presented should be considered state-of-the-art with respect to personalised phage therapy. However, in my opinion, the authors have not done a good job of presenting their data in the most impactful way, nor have they effectively communicated the work in an accessible way, particularly for a broader readership in Nature Microbiology. As such, much of my comments and suggestions will be focused on these two criticisms.

The impact of this paper, in my opinion, is on the retrospective treatment of 100 patients with personalised phage therapy. I encourage the authors to keep emphasis this front and centre when revising the manuscript in order to maintain focus, while significantly shortening their writing. I have two major summary points for the authors to address. The first is revising the overall structure of the manuscript, which is not traditional, which is fine, but as a result there are many sections and paragraphs that are not well placed, breaking the flow and narrative of the paper. Secondly, the paper is unnecessarily long and at times unfocused. An example of this is found in the discussion, which is nearly five solid pages of text, it has little organisation in its writing, and has many sections that are better suited to a literature review rather than being included within a retrospective analysis that is presented here. In brief, the authors are doing too much, and in my opinion, this has negatively impacted the paper and its message.

Please see below line-by-line comments and suggestions:

RESPONSE TO REVIEWER 3:

Thank you for your kind words. We believe this manuscript to be important to the field as well and we hope we have improved it in order to effectively communicate its importance. Please see below for a response to comments and questions raised.

Lines 92-96: 'Quality & safety were verified by Sciensano' - I think it would be of benefit to the field (and future regulators) to briefly state ***how these preparations were verified. i.e., sterility testing? Endotoxin? Sequencing? In vitro screens?***

61Response: Thank you. The details of quality and safety tests are explained in the manuscript pg. 18 lines 826-857. However, to address this question and the nextquestions raised, we included a short description in the Introduction and more specifics in the Methods section.

Lines 101-102: As noted in the abstract, not all cases employed these assays - could the authors include a short note on this point?

Response: Yes, the assays were offered to the physicians, but were only reported when sufficient and adequate consecutive bacterial samples and/or serum samples were provided to allow for a meaningful analysis. We included a note on this (in the Results section).

Lines 101-108: Is this best suited for the introduction? This writing reads more like a method and I personally think it breaks the flow between impact of Eliava and the summary of this article.

Response: Yes, this was moved to Methods.

Line 151: Can you provide *further details on the methodology used for your bioburden assays? Were these USP71 sterility tests? If so, how much volume or fraction of the batch was tested? Or was this a plating and CFU count performed by Sciensano?* This is important to define as a 0 CFU count can be a false positive depending on your volume tested compared to batch production. This detail also needs to be clearly defined in the methods, but a brief summary statement in main text is also justified due to the importance of these end-point metrics. I also note the associated methods section was very limited on these important points (lines 852-854). Readers should not have to go to other documentation/papers to find these key criteria.

Response: As stated in lines 853-854, the bioburden (total viable aerobic count) of each lot was assessed as described in European Pharmacopoeia (Ph. Eur.) chapter 2.6.12. The European pharmacopoeia provides guidance on test methods for microbial contamination of pharmaceutical preparations, and these are followed by Sciensano. The volume assessed was 4 mL (1.6-2.6% of 150-250 mL production batches).

We agree that the usage of less than 10% of the produced batch can result in false negatives. However, similar amounts are also used to assess sterility for antibiotic liquids. Moreover, although indirectly, the presence of any bacterial or fungal contaminant is also assessed by verifying the occurrence and amount of residual DNA present in the sample (purity scored by the % of phage reads) and the presence of endotoxins. We understand that more clarity is needed on these assays, so we have included more clarification in the manuscript.

Line 152: *Please clarify - were the median EU measurements of your phages 5 EU/mL of product? Or median of 5 EU/mL per patient kg?* I believe it was the former, but this should be very clear in the writing. Again, associated methods were poorly described.

Response: Correct, this was the product concentration. Additional clarifications were introduced into the revised manuscript.

Lines 154-157: *Do these requirements hold for the Eliava phage cocktails?* I note these were administered via topical, nebulised, rectal, and intralesional. From my understanding these products are not produced within safe/FDA limits for endotoxins

Response: The Eliava bacteriophage cocktails were not quality-controlled by Sciensano. Due to

their likely higher endotoxin content, these products were never administered intravenously. This information was added to the revised manuscript.Lines 159-160: Same comment as above for Eliava phage cocktails. From my understanding these cocktails are not genetically defined. In your cases using these products, were these sequenced? If so, how did you confirm <5% bacterial reads if host genomes were not available/known?

Response: Indeed, these cocktails were not genetically identified by Sciensano. We could thus not confirm < 5% bacterial reads. We included a statement about these exceptions in the manuscript.

Lines 176-199: In my opinion, this section is an example of where the authors have tried to include too much detail and depth in the manuscript, and I think it detracts from its focus.

You state that a detailed genetic comparison of adapted phages falls out of scope of this manuscript, and I agree. But you then go on and provide a detailed genomic comparison of two published adapted phages. The writing should be tightened in aspects of the manuscript, I would suggest removing this content and keeping your focus on the therapy cases. You cannot report everything in this paper, so I would suggest focusing on the clinical impact.

Response: Thank you for the suggestion. We have significantly shortened this section. We include some new information on ISP (previously unpublished) thus we would like to preserve some of the text.

Figure 1: This is a suggestion, but I feel like Figure 1 is missing an opportunity to present a simple yet impactful visualisation of the success and eradication rates of the 100 treated patients - this is likely the impactful point most readers and citations will be looking for - yet it is not graphically represented.

Response: We agree and, also based on reviewer 2 comments, we included a graph to show clinical improvement and bacterial eradication in Figure 1.

Lines 231-232: How was colicin I identified as a phage receptor? In this study or a previous one? If no knock out work was completed to validate this receptor then this should be listed as a putative receptor. Same comment with T4P in subsequent lines, either provide citation for mechanistic work, or use putative.

Response: Thank you. A citation was added for T4P that describes further mechanistic work (Ceyssens et al., 2011). For colicin I, this was corrected to say "putative receptor".

Lines 238-243: When discussing patients 54, 30, and 91 please be clear that these patients were all infected with P.a.

Response: We clarified this.

Lines 264-276: Please add a concluding statement for the Galleria model section and define the initial goals of this assay. I would also ask the authors to consider its relevance within this article. The impact here is patients treated, and I'm not certain the animal work fits well, and in some ways, it detracts from the study.

Response: A sentence was added to the manuscript to define goals. Also, a concluding sentence was added.

We believe this is an important experiment needed to show that virulence trade-offs are seen in

phage-resistant isolates. The manuscript not only focuses on patients treated, but goes above and beyond to show follow-up data to guide future phage therapy development.Figure 3: I'd encourage authors to revise their data presentation in Figure 3 - this looks more like a supplemental figure. There is a lot of data here and it is not obvious the point you are trying to make with this figure. Further the axis fonts are almost unreadable, there are no error bars presented, and I doubt non-phage experts would be able to easily decipher this figure. Why not represent this as AUC? Or another condensed metric to emphasize the phage vs phage+Abx comparison? Then include these curves as supplemental figures.

Response: Thank you for the suggestion. While AUCs are great to quantify the growth of a bacterial strain over a certain time period, some information is lost (e.g. after how many hours does the growth of resistant mutants occur). Representing the data as AUCs is surely warranted for large data sets, for statistical analysis, to compare groups, etc.

When limited to only 12 experiments, we feel that the growth curves are visible representations that we think readers of Nature Microbiology could readily understand and interpret. This is why we left this analysis as a growth curve. It is also impactful to show that these studies were performed in 72-hour assays which is not common in the field. We do agree that the axis fonts are unreadable and revised the figure to make the axis more readable.

Lines 282-287: It is worth to briefly mention how synergy and additive effects were calculated/distinguished.

Response: Our synergy definition is indeed based on the duration of suppression. We speak of synergy when the growth suppression period produced by the addition of both phage and antibiotic is clearly longer than the sum of the suppression periods induced by the antibiotic and the phage separately. The definition was added to the revised manuscript.

Lines 278-293: It was unclear from this section, ***was phage-antibiotic synergy only tested in vitro? Did these results impact or influence the patient treatment? Were these synergistic/additive combinations used clinically? How did this data inform the clinical treatment of the 100 patients?***

Response: These tests are performed *in vitro* as stated in the main text and in the Methods section. When most of these tests were performed, bacteriophage therapy had already started and test results did not influence patient treatment. However, today, based on these test results, and the overall observation that pathogen eradication is more likely when phages are applied in combination with antibiotics, we strongly advise physicians to perform these tests before treatment, when time permits. This information was added to the revised manuscript.

Figure 4: This figure can be improved by overlaying additional treatment information, such as the duration of phage therapy, clinical improvement and bacterial eradication (if these occurred), and potentially co-administration of antibiotic treatment.

Response: We added the suggested additional information to Figure 4.

Lines 318-321: Please rephrase as it currently reads as if five patients died of septic shock, when this was only two.

Response: This was clarified in the revised manuscript.

Discussion: The discussion was close to five pages of solid text. It at times, lost focus and read more like a literature review or thesis, rather than a critical discussion on the 100-

cases treated. Further, the structure of the discussion seemed erratic, jumping from topic to topic without much flow or coherency.

After finishing this section, I found I had missed/forgotten the impact of this work, which in my opinion is the treatment of 100 BT cases, and was more distracted by the writing, historical musings, and summary of prior studies.

I strongly encourage the authors to review this discussion with the goal to focus their writing on critical points, to reduce the text by over half, and to maintain a logical flow in the discussion that summarises the major findings across their study.

Response: Thank you for the suggestions. Since this manuscript is an accumulation of data from 100 patients, it is important to have a longer discussion to make several points that have been raised in this manuscript. For brevity we have tightened the discussion.

Line 356: Please remove the dashes here, it breaks sentence flow

Response: We removed the dashes.

Lines 386-388: This definition is an idyllic goal. More specifically, I note that most (arguably all) of your biolog growth curves (Fig3) show the emergence of phage resistance at low MOIs within as short as seven hours and stable lysis for 24-48 hours was not evident. This is to be expected for in vitro growth curve assays. Yet by your own definitions, these phages do not constitute an efficient therapeutic phage, yet they were used with promising success rates. So how do you reconcile these discrepancies and adhere to your own definition?

Response: These standards that we have described are not an idyllic goal but a standard when adapting phages for treatment as described in lines 164-168. A 24-48 h suppression of growth in liquid medium is indeed the goal. We admit that in some rare occasions this goal was not fully achieved or necessitated the pre-adaptation of the phages to the patient's strains. The growth curves (Fig. 3), however, were obtained using a sub-optimal ratio of phages to bacteria (= multiplicity of infection (MOI)), and antibiotics were added a sub-MIC concentrations. These sub-optimal conditions are necessary to enable the analysis of possible synergy or antagonism. If either phages or antibiotics would be applied in optimal concentrations, leading to the efficient killing of the bacterial strain by either antibiotics or phages, it would be impossible to demonstrate synergy or antagonism. This information was added to the revised manuscript.

Lines 418-430: This section reads more like a literature review than a discussion. What approach for dosing did you take in this study? As far as I could tell, dosing information from your own study was completely absent here. Keep the description of the prior work focus on your study, particularly within the discussion.

Response: We report dosing with available information in Supplementary Table 1. Many of these cases have been extensively reported with dosing details as well (see reference in Table 2).

Lines 438-443: Recommend this text be removed.

Response: The text has been shortened.

Lines 451-464: Again, a literature review style of writing that inflates your writing and retracts for

this papers message. Recommend this text be removed.Response: We think this is an important point to make about bacterial resistance and fitness costs. This text has been shortened to be reader friendly.

511-527: Recommend this text be removed.

Response: This has been significantly shortened.

We thank review 3 for the constructive comments which improve our manuscript.REVIEWER #4

Remarks to manuscript from the viewpoint of statistics

The authors correctly claim in lines 345f that “the majority of patients were unique cases” and that “no inferential statistics could be applied”. In this sense, the reporting of figures like odds ratios can be seen as hints to possible connections without fully well- grounded statistical evidence. However, reporting confidence intervals (CIs) is part of inferential statistics. I do not say that CIs should not be mentioned. But it should be noted that they must be interpreted with caution. Fisher’s exact test (line 322) is also part of inferential statistics.

Response: We would like to thank the reviewer for the comments. We were cautious to not make inferences, hence our specific wording as rightly pointed out in line 345. However, we have decided to make this further clear by prefacing the lines 37 (abstract) and line 327 (results) with “In our dataset of 100 consecutive cases, ...” to emphasize that we are not inferring about the general population of patients.

From line 119 on can conclude that some patients suffered from coinfections. This should be explicitly mentioned. It seems that a coinfection was treated as two independent infections concerning the bacteriophage therapy. My medical is not sufficient to judge whether this is justified.

Response: Thank you. We mention the fact that patients had coinfections (we decided to call them second-site infections) in line 120 “including 14 second-site infections”. Each patient case being unique, we considered clinical improvement and eradication for patients with second-site infections in a similar way as cases with single infections only. Infections and second-site infections were not analyzed as independent infections, as they are most often linked (e.g. sepsis as a consequence of a bone infection).

In line 309 it is mentioned that “in eight patients, no adequate post BT bacteriological data was available”. What is the reason for the missing data? If an adverse event like death was the reason, this is should be taken into account in the analysis.

Response: The reason for seven out of eight of these cases is that the treatment center did not collect this bacteriological data as part of their routine follow up of patients, and was not allowed to collect the data prospectively.

For the remaining case, it is not clear why the bacteriological data was not available. The treatment center either did not collect this data, failed to extract this data from the medical files, or was not able or willing to transfer this data to us.

Adverse events were not the reason for the missing data. In two cases adverse events were indeed observed, but they had no impact on the collection of bacteriological data. The reason was that the healthcare personnel was not allowed to collect bacteriological data prospectively.

More explanations were added to the revised manuscript.

Maybe this question is naïve. But is the antibiotic resistance profile (line 323) a dichotomous variable? Otherwise it is not clear to me how Fisher’s exact test was applied.

Response: That is correct, we considered whether the bacterial strain was MDR or not and this was encoded as a dichotomous variable. We have made this statement clearer in the manuscript where lines 323-325 now mention: Fisher exact test for count datashowed univariate significant effects on eradication for the following categorical variables: concomitant use of antibiotics (yes or no), antibiotic resistance profile of the targeted bacteria (MDR or not) and the clinical setting (Ambulatory or Hospitalized).

In lines 324f it is mentioned that “No effects of age or gender on eradication were found using a logistic regression analysis.” This was univariate logistic regression considering solely age and gender, respectively?

Response: Yes. This was specified in the revised manuscript.

Does “stepwise logistic regression analysis” in line 325 refer to forward selection (or backward elimination)?

Response: Thank you, the stepwise logistic regression analysis refers to forward selection. This has now been clarified in line 325 of the results (A stepwise, forward selection, logistic regression analysis...) and in the methods line 1047 (We used a stepwise, forward selection procedure...).

Logistic regression can easily show overfitting effects when it is tested on the same data with which the variables were selected and the coefficients were determined. The confusion matrix mentioned in line 329 should be computed based on leave-one-out cross-validation and not based on the training data.

Response: Thank you for this suggestion. We used a classical statistics approach here, with the logistic regression that assumes a binomial distribution of the errors. However, given the nature of the dataset, with a hundred unique cases, we are not in a position to test this assumption. Using leave-one-out cross-validation on the dataset is tempting, but we do not think that we could have overfit in this case as we only have one degree of freedom and do not have a dataset where a specific point would have a lot of leverage and pull the data in a certain direction (The variable “ABCONCOM” is binary, with 67 “yes” values and 33 “no” values).

Line 330 states that the logistic regression model was “right in 65% ... of the time”. What is the proportion of the majority class? If it is 65% or more, 65% would be very poor. I do not understand the connection between the confusion matrix mentioned in line 329 and the logistic regression. It seems that the confusion matrix provides the correlation between eradication and AB treatment. Where do the predictions of the logistic regression enter the confusion matrix?

Response: We thank the reviewer for raising this point. To answer the first part of the question, the majority class of the variable ERADICATION is “Yes”, observed 53 times. Importantly, this remark from the reviewer helped us realize that the confusion matrix presented in the Extended figure 4 was incorrectly labelled with the variable names instead of the predicted classes vs. true classes. This caused confusion regarding the interpretation of the matrix. We trust the corrected version of this figure clarifies the points raised here.

$$\text{Logit model: } \pi = \frac{1}{1 + e^{-(b_0 + b_1 x)}}$$

$$b_0: \text{ Intercept} = 0.74 \text{ (} p < 0.01 \text{)}$$

$$b_1: \text{ No concomitant AB} = -1.151 \text{ (} p < 0.01 \text{)}$$

$$\text{Odds-ratio: } \frac{\pi}{1 - \pi} = 0.309$$

π = probability that eradication is "Yes"

$1 - \pi$ = probability of "No" eradication

Confusion matrix (n=92)

		True class	
		No	Yes
Predicted class	No	20	13
	Yes	19	40

We thank review 4 for the constructive comments, which improve our manuscript.

Final Decision Letter:

Message 19th April 2024

Dear Jean-Paul,

I am pleased to accept your Article "Personalized bacteriophage therapy outcomes for 100 consecutive cases: a multi-centre, multi-national, retrospective, observational study" for publication in Nature Microbiology. Thank you for having chosen to submit your work to us and many congratulations.

1After the grant of rights is completed, you will receive a link to your electronic proof via email with a request to make any corrections within 48 hours. If, when you receive your proof, you cannot meet this deadline, please inform us at rjsproduction@springernature.com immediately. You will not receive your proofs until the publishing agreement has been received through our system

Please note that *Nature Microbiology* is a Transformative Journal (TJ). Authors may publish their research with us through the traditional subscription access route or make their paper immediately open access through payment of an article-processing charge (APC). Authors will not be required to make a final decision about access to their article until it has been accepted. Find out more about Transformative Journals

Congratulations once again and I look forward to seeing the article published.

Best wishes,